# Sample-Efficient Geometry Reconstruction from Euclidean Distances using Non-Convex Optimization

**Ipsita Ghosh**
Department of Computer Science
University of North Carolina at Charlotte
ighosh2@charlotte.edu

**Abiy Tasissa**
Department of Mathematics
Tufts University
Abiy.Tasissa@tufts.edu

**Christian Kümmerle**
Department of Computer Science
University of North Carolina at Charlotte
kuemmerle@charlotte.edu

## Abstract

The problem of finding suitable point embedding or geometric configurations given only Euclidean distance information of point pairs arises both as a core task and as a sub-problem in a variety of machine learning applications. In this paper, we aim to solve this problem given a minimal number of distance samples. To this end, we leverage continuous and non-convex rank minimization formulations of the problem and establish a local convergence guarantee for a variant of iteratively reweighted least squares (IRLS), which applies if a minimal random set of observed distances is provided. As a technical tool, we establish a restricted isometry property (RIP) restricted to a tangent space of the manifold of symmetric rank-$r$ matrices given random Euclidean distance measurements, which might be of independent interest for the analysis of other non-convex approaches. Furthermore, we assess data efficiency, scalability and generalizability of different reconstruction algorithms through numerical experiments with simulated data as well as real-world data, demonstrating the proposed algorithm's ability to identify the underlying geometry from fewer distance samples compared to the state-of-the-art.

The Matlab code can be found at github_EDG-IRLS

## 1 Introduction

Euclidean Distance Geometry (EDG) problems have applications spanning diverse domains, from comprehending protein structures through molecular conformations [MMLL19, MMWL22], prediction of molecular conformations in computational chemistry [DPRV15, SWBB97] and aiding dimensional reduction in machine learning [TSL00] to facilitating localization in sensor networks [FQX19], coupled with its role in solving partial differential equations on manifolds [LL17]. This emphasizes its broad impact in computational sciences. [LLMM14] proposes embedding entities based on Distance Geometry Problems (DGP), where object positions are determined based on a subset of pairwise distances or inner products which significantly reduces computational complexity compared to traditional word embedding methods.

*Problem* 1. Mathematically, consider a collection of $n$ points $\mathbf{p}_i$ in an $r$-dimensional Euclidean space with coordinates $\mathbf{P} = [\mathbf{p}_1, \mathbf{p}_2, ..., \mathbf{p}_n] \in \mathbb{R}^{r \times n}$ whose pairwise squared Euclidean distances are given by $d_{ij}^2 = ||\mathbf{p}_i - \mathbf{p}_j||^2$ for each $1 \leq i \neq j \leq n$ where $|| \cdot ||$ denotes the Euclidean norm. Given only partial information of the $\{d_{ij}\}$ such that only a subset of cardinality $m < n(n-1)/2$ is known, the goal is to reconstruct the geometry of the points, that is to recover the point coordinates $\mathbf{P}$.

38th Conference on Neural Information Processing Systems (NeurIPS 2024).

If all the distances between the points are provided, the problem is known as multidimensional scaling [LT24, GCKG23], and closed solution formula exists. However, this is not the case for the incomplete setup which is the focus of this work, and in which only partial information of the pairwise distances is available.

For instance, AlphaFold [SEJ+20, JEP+21] has shown effectiveness in predicting the three-dimensional structure of protein given as input its amino acid sequence. AlphaFold2, which uses an attention mechanism-based transformer architecture [VSP+17], is trained on known sequences and structures from the Protein Data Bank [HMBFGG+00] and determines the distances between the $C_\alpha$ atoms of all residue pairs in a protein. Subsequently, as a subproblem, this distance information is used to predict the protein's structure by identifying the *foldings* within the protein molecule. In the context of this subproblem, where the input is the predicted pairwise distances, a linear mapping can be used to predict geometric coordinates. However, many of the predicted distances might not be accurate, which is why low confidence regions of the resulting structures, as indicated by the predicted local-distance difference test (pLDDT) [JEP+21], could be masked, and a new structure could subsequently be re-computed based on the distance entries of the medium-to-high confidence regions using high-accuracy solution algorithms for Problem 1.

In the context of the Problem 1, we have access to a subset of entries of this $\mathbf{D} = [d_{ij}^2]$ matrix, defined by the index set $\Omega \subset \{1, \ldots, n\}^2$. Now, we formulate the Gram matrix $\mathbf{X}^0 = \mathbf{P}^\top \mathbf{P}$, residing in the set of real-valued symmetric matrices $S_n$. This matrix has a lower rank $r$ compared to the symmetric distance matrix $\mathbf{D} \in S_n = \{\mathbf{X} \in \mathbb{R}^{n \times n} : \mathbf{X} = \mathbf{X}^\top\}$, which has a rank of $r + 2$ [DPRV15].

To make the solution translation invariant, the centroid of the points must be the origin, i.e., $\sum_{\ell=1}^n p_\ell = 0$. To ensure this, the Gram matrix should satisfy also satisfy $\mathbf{X}^0 \cdot \mathbf{1} = 0$, where $\mathbf{1} \in \mathbb{R}^n$ is the vector of all ones. Based on these observations, we can frame Problem 1 as an instance of the computationally challenging NP-hard *rank minimization* [Rec11] problem defined by

$$\min_{\mathbf{X} \in S_n, \mathbf{X} \cdot \mathbf{1} = 0, \mathbf{X} \succeq \mathbf{0}} \operatorname{rank}(\mathbf{X}) \quad \text{subject to } \mathbf{X}_{i,i} + \mathbf{X}_{j,j} - 2\mathbf{X}_{i,j} = \mathbf{D}_{i,j} \quad \text{for all } (i,j) \in \Omega, \quad (1)$$

incorporating also the positive semi-definiteness constraint $\mathbf{X} \succeq \mathbf{0}$, which comes from the observation that $\mathbf{X}^0 = \mathbf{P}^\top \mathbf{P}$ is also PSD.

From an optimization perspective, this rank-minimization problem is highly complex due to its non-convexity and non-smoothness. Being an NP-hard problem [TL18, Hen95, MW97], it is extremely difficult to solve directly. Consequently, a significant amount of existing research [Rec11, DR16, CR09, CW18, Gro11, TL18] has focused on minimizing the convex envelope of the rank function, known as the nuclear norm, instead. However, it was observed that for related low-rank matrix completion problem, such convex relaxations are not as data-efficient as other formulations [ALMT14, TW13], while posing also computational limitations [CC14].

In this paper, we study the question of, given *incomplete* distance information Problem 1, how many samples are necessary to ensure accurate recovery? [TL18] established convergence with $O(\nu r n \log^2 n)$ uniformly random samples for solving (1) using a nuclear norm surrogate. While there are numerous results for non-convex methods in other low-rank recovery problems [KV21, CLC19, Van13], there are currently only few non-convex algorithm specifically designed for EDG problems available [TL18, SLCT23, ZCZ22, NKKS19, LS24]. To the best of our knowledge, only [NKKS19, LS24] provide convergence guarantees in the non-convex setting. Unlike generic low-rank matrix recovery problems [RFP10], EDG problems share the complexities of low-rank matrix completion problems, such as the absence of a direct uniform null space property [RXH11] or a restricted isometry property [RFP10]. Additionally, the underlying measurement basis in EDG problems is not orthonormal, making it impossible to directly use the analysis of standard matrix completion to provide guarantees for techniques using the Gram matrix-based low-rank modelling (1) of Problem 1.

## 1.1 Our Contribution

In this paper, we propose and analyze an algorithm for the EDG problem based on the iteratively reweighted least squares framework (MatrixIRLS, see Section 3), for which we show that $m = \Omega(\nu r n \log(n))$ (where $\nu$ is the coherence factor) randomly sampled distances are sufficient to guarantee local convergence to the ground truth, $\mathbf{X}^0$ with a quadratic rate as shown in Theorem 4.3, from which the geometry of the points $\mathbf{P}$ is trivially recovered. The sample complexity assumption

of Theorem 4.3 matches the lower bound of low-rank matrix completion problems as established in [CT10]. In Section 4, we construct a dual basis (Lemma 4.4) that spans $S_n$ which enables us to show the restricted isometry property (RIP) restricted to a tangent space of the manifold of symmetric rank-$r$ in Theorem 4.5 for our approach, which can be of independent interest for the analysis of other nonconvex algorithms.

While the convergence statement of Theorem 4.3 only applies in a local neighborhood of a ground truth, the indicated data-efficiency of the proposed method is numerically validated through different experiments in Section 5 on synthetic and real data in comparison to the state-of-the-art methods. Furthermore, we demonstrate that MatrixIRLS method is robust to ill-conditioned data, further highlighting its flexibility. In the Supplementary material, we discuss the limitations in Appendix B. Additionally, we provide proofs related to the the theoretical results of Section 4 in Appendices B and C. We further discuss the numerical considerations of our experiments in Appendix E. We discuss about the computational complexity of the algorithm in detail in Appendix F.

## 2   Related Work

Early research on EDG problems focused on establishing its mathematical properties like defining the conditions under which a distance matrix can be represented in Euclidean space[Gow85, YH38, YH38, Alf05].A comprehensive overview of EDG applications, including molecular conformations, wireless sensor networks, robotics, and manifold learning, is available in [Lib20]. Motivated by the molecular conformation problem, [BJ95] relates this Euclidean distance matrix completion to a graph realization problem by showing that such matrices can be completed if the graph is chordal. Other than graph theoretic approaches, as a technical tool, various optimization strategies [Hen95, MW97, Tro00] have been deployed to solve EDG problems. [AKW99] proposes a primal-dual interior point algorithm that solves an equivalent semi-definite programming problem. However, none these works provide theoretical reconstruction guarantees in the incomplete setup of Problem 1.

While providing an accurate modelling of Problem 1, the rank minimization formulation (1) poses challenges due to its non-convex and non-smooth nature. There is a mature existing literature [DR16, CR09, CW18, Gro11] around replacing the rank function, $\text{rank}(\mathbf{X})$, with the sums of its singular values $\sigma_i(\mathbf{X})$ (also known as the nuclear norm). Building on the rank minimization formulation (1) of the EDG problem, [TL18] minimizes the convex nuclear norm surrogate of the inferred Gram matrix. They propose a dual basis approach that enables a theoretical guarantee for this type of nuclear norm minimization formulation of the EDG problem.

From a practical point of view, it is well-known that the convex approach is computationally intensive, as the arithmetic complexity is cubic in $n$, the dimension of $\mathbf{X}^0$ [CC14]. It also tends to demand more data samples than non-convex alternatives, making it less efficient in terms of data [ALMT14, TW13].

To mitigate these issues, recent studies have shifted focus to non-convex methods such as matrix factorization [SL16, MWCC20, ZCZ22]. These methods optimize a function involving a data-fit objective regularized by squared Frobenius norms of the factor matrices, which are computationally more feasible and data-efficient. Few "non-convex" algorithms are based on matrix factorization like the work in [BM03].

Based on a similar formulation, ScaledSGD [ZCZ22] is a preconditioned stochastic gradient descent method aimed at robustness for ill-conditioned problems. Additionally, some of the most effective techniques for low-rank matrix completion involve minimizing smooth objectives on the Riemannian manifold of fixed-rank matrices, providing scalability and the potential to reconstruct the matrix with fewer samples, although they lack strong performance guarantees [Van13, WCCL20, BA15, BNZ21]. ReiEDG [SLCT23] is a Riemannian-based gradient descent strategy utilizing the sampling operator on $\Omega$, which is non-convex approach to solving the EDG problem. However, it does not provide convergence guarantees. To the best of our knowledge, [NKKS19, LS24] are the non-convex approaches for solving the EDG problem in the Gram-matrix-based low-rank modeling (1) of Problem 1. The work in [NKKS19] proposes an algorithm based on Riemannian optimization over a manifold and provides convergence guarantees. These guarantees are derived from extended Wolfe conditions. However, it is not explicitly detailed how these convergence guarantees depend on problem parameters such as the sampling model and the number of samples (see Remark III.8 in [NKKS19]). Similarly, the study in [LS24] employs a Riemannian framework and provides local convergence guarantees under Bernoulli sampling. Nonetheless, [LS24] does not clarify whether

the proposed algorithm achieves local linear convergence. In contrast to these studies, the algorithm proposed in this paper achieves local quadratic convergence under uniform sampling of the distances.

To handle the non-smoothness and non-convexity of rank minimization problems of the type (1), iteratively reweighted least squares (IRLS) algorithms take a different route than methods based on [BM03] or Riemannian methods by minimizing a sequence of quadratic majorizing functions of smoothed rank surrogates [DDFG10, FRW11, MF10, KS18, KV21]. IRLS algorithms have been studied extensively over the years, as indicated by [Law61, GR97, Bur12]. In the context of low-rank matrix completion, IRLS algorithms are known to be among the most data-efficient methods available, while being amenable to a rigorous convergence analysis [FRW11, MF10, KS18, KV21]. Most recently, [KV21] provided an improvement on previous instantiations of the IRLS framework [FRW11, MF10, KS18] for low-rank optimization problems by providing an improved reweighting strategy, for which the authors show a local convergence guarantee that is applicable for low-rank matrix completion, given random entrywise samples of minimal sample complexity. The algorithm we propose in Section 3 is similar to [KV21, Algorithm 1] and Theorem 4.3 follows partially the proof strategy of a related result in [KV21]. However, the setup of Problem 1 does not allow a direct adaptation of both the implementation and analysis of [KV21] due to the non-orthogonality of the measurement basis.

## 3  MatrixIRLS for Euclidean Distance Geometry

In this section, we provide a detailed outline and description of the iteratively reweighted least squares method in the context of the EDG reconstruction problem. To this end, we define preliminaries for stating the algorithm. MatrixIRLS, defined in Algorithm 1 below, can be interpreted as a hybrid of a smoothing method [Che12] and a *Majorization-Minimization* algorithm [SBP17]. In particular, the proposed algorithm minimizes *smoothed log-det objectives* defined as $F_\epsilon(\mathbf{X}) := \sum_{i=1}^n f_\epsilon(\sigma_i(\mathbf{X}))$

$$f_\epsilon(\sigma) = \begin{cases} \log|\sigma|, & \text{if } \sigma \geq \epsilon, \\ \log(\epsilon) + \frac{1}{2}\left(\frac{\sigma^2}{\epsilon^2} - 1\right), & \text{if } \sigma < \epsilon, \end{cases} \tag{2}$$

which is a continuously differentiable function with $\epsilon^{-2}$-Lipschitz gradient [KV21].

We can decompose $\mathbf{X}$ by $\mathbf{X} = \mathbf{U}\,\mathrm{dg}(\gamma\sigma)\mathbf{U}^\top$, where $\gamma \in \{+1, -1\}$, It is clear that the optimization landscape of $F_\epsilon$ crucially depends on the smoothing parameter $\epsilon$. Instead of minimizing $F_\epsilon$ directly, our method minimizes, for $k \in \mathbb{N}$, $\epsilon_k > 0$ and $\mathbf{X}^{(k)}$ a *quadratic model* that is related to the second-order Taylor expansion of the function $f_\epsilon$ at the current iterate and its information is encoded in a weight operator [KV21] defined below in Definition 3.1.

**Definition 3.1** ([KV21]). $\mathbf{X} \in S_n$ be a matrix with singular value decomposition $\mathbf{X}^{(k)} = \mathbf{U}\,\mathrm{dg}(\gamma\sigma)\mathbf{U}^\top$, where $\gamma \in \{+1, -1\}$ i.e., $\mathbf{U} \in S_n$ are orthonormal matrices. We define the *weight operator core matrix* $\mathbf{H}_{\mathbf{X},\varepsilon} \in S_n$ of $\mathbf{X}$ for smoothing parameter $\varepsilon > 0$ such that

$(\mathbf{H}_{\mathbf{X},\varepsilon})_{ij} := \Big(\max(\sigma_i, \epsilon)\max(\sigma_j, \epsilon)\Big)^{-1}$ for each $i, j \in \{1, \dots, n\}$ and the *weight operator* $W_{\mathbf{X},\varepsilon} : S_n \to S_n$, which maps any $\mathbf{Z} \in S_n$ to

$$W_{\mathbf{X},\varepsilon}(\mathbf{Z}) := \mathbf{U}\left[\mathbf{H}_{\mathbf{X},\varepsilon} \circ (\mathbf{U}^\top \mathbf{Z} \mathbf{U})\right]\mathbf{U}^\top, \tag{3}$$

where $\circ$ denotes the entrywise ore Hadamard product of two matrices.

Our method is designed to provide iterates that satisfy the constraints of the formulation (1) at each iteration. They can be encoded using the following definition.

**Definition 3.2.** Given $n \in \mathbb{N}$, let $\mathbb{I} = \{\alpha = (i, j) \mid 1 \leq i < j \leq n\}$ be the index set of upper triangular indices.

We define the operator basis $\{\mathbf{w}_\alpha\}_{\alpha \in \mathbb{I} \cup \{(i,i):i \in \{1,\dots,n\}\}}$ where

$$\mathbf{w}_\alpha = \begin{cases} \mathbf{e}_i\mathbf{e}_i^\top + \mathbf{e}_j\mathbf{e}_j^\top - \mathbf{e}_i\mathbf{e}_j^\top - \mathbf{e}_j\mathbf{e}_i^\top, & \text{if } \alpha = (i, j) \in \mathbb{I}, \\ \frac{1}{2}(\mathbf{e}_i\mathbf{1}^\top + \mathbf{1}\mathbf{e}_i^\top), & \text{if } \alpha = (i, i) \text{ for some } i \in \{1, \dots, n\}, \end{cases} \tag{4}$$

where $\mathbf{e}_i \in \mathbb{R}^n$ is the $i$-th standard basis vector.

Given a set (or multiset) of indices $\Omega = \{\alpha_1, \ldots, \alpha_m\} \subset \mathbb{I}$ of cardinality $m = |\Omega|$, we define the *measurement operator* $\mathcal{A} = \mathcal{A}_\Omega : S_n \to \mathbb{R}^{m+n}$ which maps $\mathbf{X} \in S_n$ to $\mathcal{A}(\mathbf{X})$ whose $\ell$-th coordinate is defined as $\mathcal{A}(\mathbf{X})_\ell = \langle \mathbf{w}_{\alpha_\ell}, \mathbf{X} \rangle$ for $\ell \leq m$ and as $\mathcal{A}(\mathbf{X})_\ell = \langle \mathbf{w}_{(\ell-m,\ell-m)}, \mathbf{X} \rangle$ for $\ell > m$.

The basis $\{\mathbf{w}_\alpha\}_\alpha$ of Definition 3.2 can be considered as an extended definition of the primal basis used in [TL18, LT24], but additionally is able to encode the constraint $\mathbf{X} \cdot \mathbf{1} = \mathbf{0}$ which guarantees that the Gram matrix corresponds to points whose centroid is located at the origin. Accordingly, we can define the constraint set corresponding to Gram matrices of points that are centered and simultaneously satisfy the pairwise distance constraints of Problem 1 as $\{\mathbf{X} \in S_n : \mathcal{A}(\mathbf{X}) = [\mathbf{D}_\Omega; \mathbf{0}]\}$. The algorithm below minimizes the quadratic, majorizing model of the objective $F_{\epsilon_k}(\cdot)$ given $k$, while satisfying the measurement operator based on the sampled distances. Equivalently, we can define the main computational step of the method as

$$\mathbf{X}^{(k+1)} = \underset{\text{s.t. } \mathcal{A}(\mathbf{X}) = [\mathbf{D}_\Omega; \mathbf{0}]}{\arg\min} \langle \mathbf{X}, W_k(\mathbf{X}) \rangle, \tag{5}$$

where $W_k : S_n \mapsto S_n$ is defined as $W_k := W_{\mathbf{X}^{(k)}, \epsilon_k}$ with the definition of the weight operator Definition 3.1 above. With this preparation, we provide an outline of MatrixIRLS in Algorithm 1 below. Equation (7) provides suitable update rule for the smoothing parameter sequence $(\epsilon_k)_{k \in \mathbb{N}}$ that enables the computation of only $r = O(\widetilde{r})$ singular triplets of each algorithmic iterate [KV21].

---

**Algorithm 1** `MatrixIRLS` for Euclidean Distance Geometry Problems

---

**Input:** Index pairs $\Omega \subset \mathbf{1}$, distances $\mathbf{D}_\Omega = (d_{ij})_{(i,j) \in \Omega}$, rank estimate $\widetilde{r}$. **Output:** $\mathbf{X}^{(k)}$ after suitable stopping condition.
Initialize $k = 0$, $\epsilon_0 = \infty$ and $W_0 = \mathrm{Id}$.
**for** $k = 1, 2, \ldots,$ **do**
  **Solve weighted least squares:** Solve (5) by

$$\mathbf{X}^{(k)} = W_{k-1}^{-1} \mathcal{A}^* \left( \mathcal{A} W_{k-1}^{-1} \mathcal{A}^* \right)^{-1} \mathbf{y} \tag{6}$$

  **Update smoothing:** Compute $\widetilde{r} + 1$-th singular value of $\mathbf{X}^{(k)}$ to update

$$\epsilon_k = \min \left( \epsilon_{k-1}, \sigma_{\widetilde{r}+1}(\mathbf{X}^{(k)}) \right). \tag{7}$$

  **Update weight operator:** For $r_k := |\{i \in [n] : \sigma_i(\mathbf{X}^{(k)}) > \epsilon_k\}|$, compute the first $r_k$ singular values $\sigma_i^{(k)} := \sigma_i(\mathbf{X}^{(k)})$ and matrices $\mathbf{U}^{(k)} \in \mathbb{R}^{n \times r_k}$ with leading $r_k$ left singular vectors of $\mathbf{X}^{(k)}$ to update $W_k = W_{\mathbf{X}^{(k)}, \epsilon_k}$ defined in Definition 3.1.
**end for**

---

### 3.1 Computational Considerations

The computational complexity of this above algorithm can be computed from the steps (6) and (7). In our numerical implementation, we largely follow the tangent space formulation of the weighted least squares step (6), cf. [KV21, Section 3], which involves the solution of an order $O(nr_k) = O(nr)$ linear system if $\widetilde{r} = r$ is chosen as the ground truth rank. An additional difficulty we overcame in the provided reference implementation arises from the fact that $\mathcal{A}\mathcal{A}^*$ is not the identity. The per-iteration time complexity of our method is dominated by $O((mr + r^2n)\mathrm{N}_{inner}^0)$, where $\mathrm{N}_{inner}^0$ is the number of inner iteration bound of the iterative linear system solver. Detailed FLOPs calculation is shown in Appendix F.

## 4 Theoretical Analysis

In this section, we discuss about the local convergence of MatrixIRLS in Section 4.1 and establish RIP restricted to the tangent space of the manifold of symmetric rank-$r$ matrices at the ground truth Gram matrix $\mathbf{X}^0$ in Section 4.2.

### 4.1 Local Convergence Analysis of Algorithm 1

It is well-known in the literature on low-rank matrix completion that for an entrywise measurement basis, recovery from generic measurements is more difficult if most information of the low-rank

matrix is concentrated in few entries, and this observation is typically captured by the notion of incoherence [CR09, Rec11, Che15]. For the purpose of the EDG problem of interest, we use the following coherence notion, which has appeared in similar form in [TL18].

**Definition 4.1** (Coherence for Gram matrices in the EDG problem, [TL18]). Let $\mathbf{X} \in S_n$ be of rank $r$. Let $T = T_{\mathbf{X}} = \{\mathbf{Z}\mathbf{X} + \mathbf{X}\mathbf{Z}^\top : \mathbf{Z} \in \mathbb{R}^{n \times n}\}$ be the tangent space onto the rank-r manifold $\mathcal{M}_r = \{\mathbf{Z} \in S_n : \operatorname{rank}(\mathbf{Z}) = r\}$ at $\mathbf{X}$. We say that $\mathbf{X}$ has coherence $\nu$ with respect to the basis $\{\mathbf{w}_\alpha\}_{\alpha \in \mathbb{I}}$ of the subspace $\{\mathbf{X} \in S_n : \mathbf{X}\mathbf{1} = \mathbf{0}\}$ if

$$\max_{\alpha \in \mathbb{I}} \sum_{\beta \in \mathbb{I}} \langle \mathcal{P}_T \mathbf{w}_\alpha, \mathbf{w}_\beta \rangle^2 \leq 2\nu \frac{r}{n} \quad \text{and} \quad \max_{\alpha \in \mathbb{I}} \sum_{\beta \in \mathbb{I}} \langle \mathcal{P}_T \mathbf{v}_\alpha, \mathbf{w}_\beta \rangle^2 \leq 4\nu \frac{r}{n},$$

where $\mathcal{P}_T : S_n \to S_n$ denotes the projection operator onto $T$ and $\{\mathbf{v}\}_{\alpha \in \mathbf{I}}$ is a *dual basis* of $\{\mathbf{w}_\alpha\}_{\alpha \in \mathbb{I}}$, which means that $\langle \mathbf{w}_\alpha, \mathbf{v}_\beta \rangle = \delta_{\alpha,\beta}$ for each $\alpha, \beta \in \mathbb{I}$ ($\delta_{\alpha,\beta} = 1$ for $\alpha = \beta$ and equal to 0 otherwise).

*Remark* 4.2. In [TL18, Definition 1], the coherence constant $\nu$ was required to satisfy a third condition (see [TL18, (Ineq. 14)]). However, this condition is not needed for our proofs, which is why we can use the weaker definition of Definition 4.1. Similar improvements for the standard basis incoherence notion were achieved in [Che15]. In [TL18, Lemma 21], it was shown that up constants, the definition above is equivalent to a coherence condition with respect to the standard basis [Rec11, Che15].

Following the conventional sampling approach in the existing literature [Rec11, Che15, CWB08], the index set is $\Omega = (i_\ell, j_\ell)_{\ell=1}^m \subset \mathbf{I}$ contains $m$ samples drawn uniformly at random without replacement.

**Theorem 4.3** (Local convergence of MatrixIRLS for EDG with Quadratic Rate). *Let $\mathbf{X}^0 \in S_n$ be a matrix of rank $r$ that is $\nu$-incoherent, and let $\mathcal{A} : S_n \to \mathbb{R}^{m+n}$ be the measurement operator corresponding to an index set $\Omega \subset \mathbb{I}$ of size $m = |\Omega|$ that is drawn uniformly without replacement. There exist constants $C^*$, $\widetilde{C}$ and $C$ such that the following holds. (a) If the sample complexity fulfills $m \geq C\nu rn \log n$, and if (b) the output matrix $\mathbf{X}^{(k)} \in S_n$ of the k-th iteration of MatrixIRLS for EDG with inputs $\mathbf{y} = \mathcal{A}(\mathbf{X}^0)$ and $\widetilde{r} = r$ updates the smoothing parameter in (7) such that $\epsilon_k = \sigma_{r+1}(\mathbf{X}^{(k)})$ and fulfills $\|\mathbf{X}^{(k)} - \mathbf{X}^0\|_{S_\infty} \lesssim \frac{m^{\frac{3}{2}}}{C^* \kappa L^2 (\log n)^{\frac{3}{2}} \sqrt{n-r}} \sigma_r(\mathbf{X}^0)$ where $\kappa = \sigma_1(\mathbf{X}^0)/\sigma_r(\mathbf{X}^0)$ is the condition number of $\mathbf{X}^0$, then the **local convergence rate is quadratic** in the sense that $\|\mathbf{X}^{(k+1)} - \mathbf{X}^0\|_{S_\infty} \leq \min(\mu\|\mathbf{X}^{(k)} - \mathbf{X}^0\|_{S_\infty}^2, \|\mathbf{X}^{(k)} - \mathbf{X}^0\|_{S_\infty})$ with $\mu = \left(\frac{m}{\widetilde{C}^2 L \log n}\right) \frac{1}{4(1+6\kappa)\sigma_r(\mathbf{X}^0)}$ and furthermore $\mathbf{X}^{(k+\ell)} \xrightarrow{\ell \to \infty} \mathbf{X}^0$ with high probability.*

*(The values of the constants $C^* = 10^5, \widetilde{C} = 21\sqrt{5}, C = 4900$ are explicitly derived in the Supplementary material.)*

In other words, Theorem 4.3 indicates Algorithm 1 converges to the ground truth with high probability with a sample complexity of $\Omega(\nu rn \log n)$, if initialized close to the ground truth Gram matrix. We refer to Appendix D for its proof.

Our theorem's sample complexity requirement aligns with the lower bound for generic low-rank matrix completion problems, as established in [CT10]. We note that this theorem only provides a local convergence guarantee for Algorithm 1. This is in line with the strongest known results for IRLS algorithms optimizing non-convex objectives [For10, KV21, PKV22]. We provide numerical evidence in Section 5 that the minimal sample complexity assumption (a) of Theorem 4.3 indeed captures the generic reconstruction ability of the method.To the best of our knowledge, Theorem 4.3 represents the first convergence guarantee for any algorithm for Problem 1 that applies at the optimal order $\Omega(\nu rn \log n)$ of provided pairwise distances, and furthermore, the first theoretical guarantee for any non-convex optimization framework for Problem 1.

## 4.2 Dual Basis Construction and Local Restricted Isometry Property on Tangent Spaces

While a convergence result similar to Theorem 4.3 had been previously obtained for an IRLS algorithm for low-rank matrix completion [KV21], an adaptation of the proof of [KV21] to the EDG setting is not possible due to non-orthogonality of the basis $\{\mathbf{w}_\alpha\}_\alpha$ of Definition 3.2.

In order to prove Theorem 4.3, we establish a restricted isometry property of a suitably defined sampling operator (see (9)) with respect to the tangent space of the manifold of symmetric rank-$r$ matrices at the ground truth Gram matrix $\mathbf{X}^0 = \mathbf{P}^\top \mathbf{P}$. To formulate this sampling operator and the respective RIP condition, we construct a *dual* basis to the measurement basis $\{\mathbf{w}_\alpha\}_\alpha$ of Definition 3.2.

**Lemma 4.4** (Dual Basis Construction). *Let $n \in \mathbb{N}$, $\mathbb{I} = \{(i,j) \mid 1 \leq i < j \leq n\}$ be the index set and $\{\mathbf{w}_\alpha\}_{\alpha \in \mathbb{I} \cup \mathbb{I}_D}$ be the primal basis of Definition 3.2 with $\mathbb{I}_D = \{(i,i) : i \in \{1, \ldots, n\}\}$. If*

$$\mathbf{v}_\alpha = \begin{cases} -\frac{1}{2}(\mathbf{a}_i \mathbf{a}_j^\top + \mathbf{a}_j \mathbf{a}_i^\top), & \text{if } \alpha = (i,j) \in \mathbb{I}, \\ \mathbf{e}_i \mathbf{e}_i^\top - \mathbf{a}_i \mathbf{a}_i^\top, & \text{if } \alpha = (i,i) \in \mathbb{I}_D, \end{cases} \tag{8}$$

*where $\mathbf{a}_i = \mathbf{e}_i - \frac{1}{n}\mathbf{1}$ for $i \in \{1, \ldots, n\}$, then $\{\mathbf{v}_\alpha\}_{\alpha \in \mathbb{I} \cup \mathbb{I}_D}$ is a dual basis with respect to $\{\mathbf{w}_\alpha\}_{\alpha \in \mathbb{I} \cup \mathbb{I}_D}$, i.e., $\{\mathbf{v}_\alpha\}_\alpha$ and $\{\mathbf{w}_\alpha\}_\alpha$ are bi-orthogonal.*

Lemma 4.4 extends the dual basis construction of [TL18, LT24], in which the duality of $\{\mathbf{v}_\alpha\}_{\alpha \in \mathbb{I}}$ with respect to $\{\mathbf{w}_\alpha\}_{\alpha \in \mathbb{I}}$ was shown. The proof of Lemma 4.4 is detailed in Appendix B.1.

Unlike the basis pair of [TL18, LT24], our bases span the entire space of symmetric matrices $S_n$ (see Appendix B.2), which enables us to show the following restricted isometry property.

**Theorem 4.5** (Restricted Isometry Property for Sampling Operator $\mathcal{Q}_\Omega$). *Let $L = n(n-1)/2$, $0 < \epsilon \leq \frac{1}{2}$, and $\Omega \subset \mathbb{I}$ be a multiset of size $m$ sampled independently with replacement. Define the sampling operator $\mathcal{Q}_\Omega : S_n \to S_n$ such that*

$$\mathcal{Q}_\Omega(\mathbf{X}) := \frac{L}{m} \sum_{\alpha \in \Omega \cup (i,i)_{i=1}^n} \langle \mathbf{X}, \mathbf{w}_\alpha \rangle \mathbf{v}_\alpha \tag{9}$$

*where $\mathbf{w}_\alpha$ and $\mathbf{v}_\alpha$ as in (4) and (8), respectively. Let $\mathbf{X}^0 \in S_n$ be a $\nu$-incoherent matrix whose tangent space onto the manifold $\mathcal{M}_r$ of symmetric rank-$r$ matrices is denoted as $T_0 = T_{\mathbf{X}^0}$. Let $\mathcal{P}_{T_0} : S_n \to S_n$ be the projection operator associated to $T_0$. Then $\left\| \mathcal{P}_{T_0} \mathcal{Q}_\Omega^* \mathcal{P}_{T_0} - \mathcal{P}_{T_0} \right\|_{S_\infty} \leq \epsilon$ holds with probability at least $1 - \frac{2}{n}$ provided that*

$$m \geq (49/\epsilon^2)\nu n r \log n.$$

The dual basis as discussed in Section 3 along with the extension for the diagonal entries together spans the space of $n \times n$ symmetric matrices. This construction has been crucial in proving the RIP restricted to the tangent space $T_{\mathbf{X}}$ of rank constrained smooth manifold $\mathcal{M}_r$. This construction could also be valuable for analyzing other non-convex algorithms. A detailed proof of Theorem 4.5 is provided in Appendix C This proof is achieved by using concentration inequalities like the Matrix Bernstein inequality theorem C.1 in multiple lemmas stated and proved in detail in the supplemental material.

Since the existing literature for nonconvex approaches for solving Problem 1 lacks this property, this approach of establishing RIP by restricting it to the tangent space can be useful for the analysis of other nonconvex methods. The RIP condition, originally introduced in the context of compressed sensing [Can08], is a fundamental assumption in the literature on low-rank matrix recovery ([ZL12, Che15, KV21, CZ13]). A tangent-space restricted RIP has been has been useful for analyzing the convergence and performance properties of other non-convex methods [GLM16, LZ23, LHLZ20, ZLTW18] in a non-EDG setting.

## 5 Numerical Experiments

We evaluate the performance of MatrixIRLS, Algorithm 1, for instances of the EDG reconstruction Problem 1 in terms of data efficiency across multiple datasets in comparison to other state-of-the-art methods in the literature. We compare the performance of MatrixIRLS with three other algorithms: (a) ALM, an augmented Lagrangian method that minimizes the non-convex formulation defined by $\min_{\mathbf{P} \in \mathbb{R}^{r \times n}} \text{Tr}(\mathbf{P}^\top \mathbf{P})$ subject to $\mathcal{R}_\Omega(\mathbf{P}^\top \mathbf{P}) = \mathcal{R}_\Omega(\mathbf{X}^0)$,

with $\mathcal{R}_\Omega$ being equal to the sampling operator $\mathcal{Q}_\Omega$ restricted to $\Omega$, which is based on a Burer-Monteiro factorization of the Gram matrix, and which has been studied in the numerical experiments of [TL18], (b) ScaledSGD [ZCZ22] which is a preconditioned stochastic gradient descent method designed to be robust with respect to ill-conditioned problems, (c) RieEDG [SLCT23] which is a Riemannian-based gradient descent approach based on the sampling operator (9) restricted to $\Omega$. The choice of these algorithms is based on their robustness to noise as claimed in the respective papers.

### 5.1 Synthetic Data

We first consider a synthetic data, where we select $n = 500$ points $\mathbf{P}^0 = [\mathbf{p}_1, \ldots, \mathbf{p}_n] \in \mathbb{R}^{r \times n}$ from a standard Gaussian distribution at random such that $(\mathbf{p}_i)_j \overset{i.i.d.}{\sim} \mathcal{N}(0,1)$ for all $i, j$, which

defines the ground truth Gram matrix $\mathbf{X}^0 = \mathbf{P}^{0\top}\mathbf{P}^0$. We are provided with $m = |\Omega|$ Euclidean distances, where the point index pair set $\Omega \subset \{(i,j) \in [n] \times [n], i < j\}$ is sampled uniformly at random. This is parametrized by the oversampling factor $\rho = \frac{m}{nr - r(r-1)/2}$ where the denominator is the degrees of freedom (discussed in Appendix E.4). To understand the efficiency of the algorithm over a range of ranks $r$ and across different oversampling factors $\rho$, we conduct phase transition experiments for all the above mentioned algorithms. We define a successful recovery as the case of the relative Procrustes distance $d_{\text{Procrustes}}(\mathbf{P}_{rec}, \mathbf{P}^0)$ (discussed in Appendix E.3) between the recovered matrix $\mathbf{P}_{rec}$ and ground truth coordinate matrix $\mathbf{P}^0$ does not exceed a tolerance threshold $\text{tol}_{\text{rec}} = 10^{-3}$. We chose the Procrustes distance at it is a shape preserving distance that accounts also for differences in scaling and alignment between the reconstructed geometries. We observe the performance of the algorithms as shown in the Figure 1 for ranks between 2 to 5 and a oversampling factor ranging from 1 to 4 over 24 instances.

In Figure 1, each entry on the figure represents the probability of success of an algorithm for the given rank ground truth rank $r$ and oversampling factor $\rho$ over 24 instances.[1] In terms of the recovery of the ground truth, we notice a comparable performance for the MatrixIRLS and ALM. However, the other two algorithms RieEDG and ScaledSGD, for the given success tolerance, the recovery of a ground truth is only possible if more samples corresponding to a larger oversampling factor are provided, for any of the ranks $r$ considered. This emphasizes that MatrixIRLS is able to achieve state-of-the-art data efficiency.

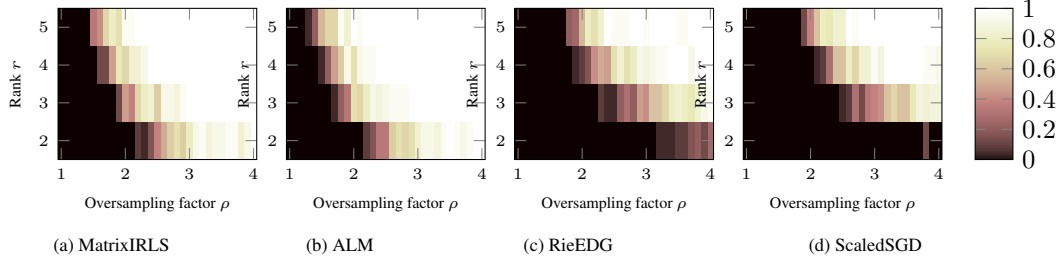

Figure 1: Success probabilities for recovery for different algorithms, given Gaussian ground truths $\mathbf{X}^0$ of different ranks, computed across 24 instances.

In order to understand the scalability of the proposed algorithm, we have provided the time of completion of MatrixIRLS in Table 1. The first two rows shows that for fixed oversampling factor $\rho = 3$ , we are able to recover the points in the magnitude of $10^4$, is just 13.7 minutes. So to further test the scalability, we also looked at the time to completion when less number of samples are provided ($\rho = 2.5$). In that setup $n = 10000$ takes 57.5 minutes to recover with high precision.

## 5.2 Ill-conditioned Data

We now assess the performance of the different EDG methods on ill-conditioned data. Ill-conditioning in the point matrix in particular arises in situations where, for example, $r$-dimensional points are approximately following an $r-1$-dimensional geometry, a situation which often arises when the available distance information if affected by outliers [ZCZ22].

Similar to the setup above, we construct Gram matrices $\mathbf{X}^0 \in S_n$ of rank $r = 5$ with condition number $\kappa = \sigma_1(\mathbf{X}^0)/\sigma_r(\mathbf{X}^0) = 10^5$ corresponding to $n = 400$ random data points $\mathbf{P}^0 \in \mathbb{R}^{r \times n}$ generated with random orthogonal singular vectors and singular values $\sigma_i(\mathbf{X}^0)$ interpolated between $\kappa$ and 1 with decay of order $O(i^{-2})$. Figure 2 shows the success probability of different algorithms for ill-conditioned ground truths.

To have a deeper insight into the algorithm's performance, Figure 3 shows the per-iterate relative reconstructed error of the four algorithms on this ill-conditioned data at an oversampling factor of $\rho = 2$ for a representative problem instance. This reconstructed error refers to the Procrustes distance between the ground truth $\mathbf{P}^0$ and the recovered $\mathbf{P}^k$ at each iteration $k$. We observe that for ill-conditioned data, MatrixIRLS is the only method that can recover the ground truth up to a

---

[1]For example, for a rank $r$ and oversampling factor $\rho$, if out of the 24 instances, in 12 such cases, the Procrustes distance based error is less than the threshold, then the probability of success is $\frac{1}{2}$.

reasonable precision, achieving a relative error of $\approx 10^{-8}$ after around 35 iterations (top of Figure 3), whereas the other methods do not achieve errors below $10^{-3}$ even after 100000 iterations. While one iteration of MatrixIRLS typically has a longer runtime than an iteration of any of the other algorithms due to its second-order nature, we also provide a visualization of the observed runtimes of the methods in the bottom of Figure 3. It can be seen that MatrixIRLS is able reconstruct the geometry of the challenging problem in around 200 seconds up to a high precision. Additionally, we run a study on the algorithms' behavior across different oversampling factors between $\rho = 1$ and $\rho = 4$ for 24 random problem instances, visualized in Figure 4. The box plots indicate visualize median relative Procrustes error together with the relevant 25% and 75% quantiles of the observed error distribution. We observe that MatrixIRLS has consistent convergence for ill-conditioned data even for lower oversampling rates as long as $\rho \geq 1.5$.

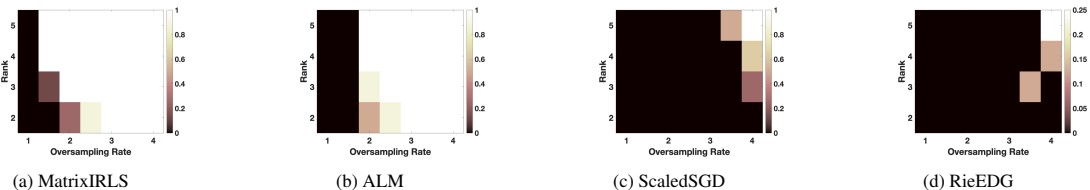

(a) MatrixIRLS        (b) ALM        (c) ScaledSGD        (d) RieEDG

Figure 2: Empirical success probabilities for recovery for different algorithms, given ill-conditioned ground truths $\mathbf{X}^0$ of different ranks, computed across 8 instances.

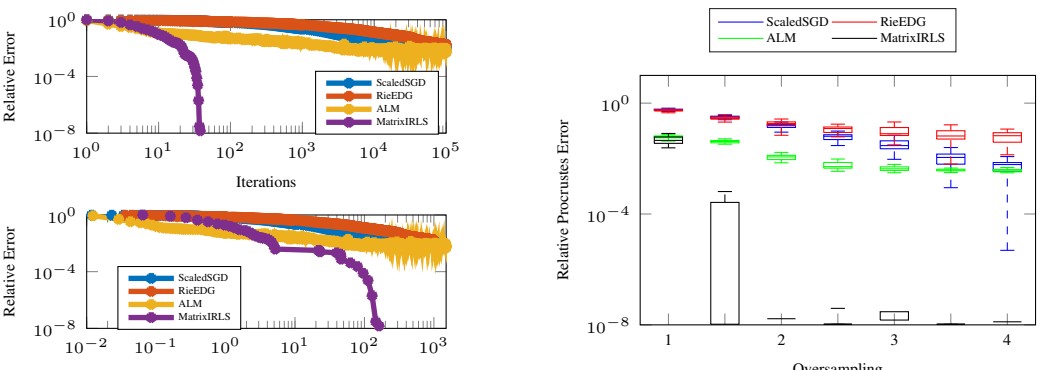

Figure 3: Top: Relative error plot. Bottom: Runtime across iterates for one instance.

Figure 4: Performance of EDG completion algorithms over 24 instances on ill-conditioned data.

## 5.3 Real Data

To assess the performance of Algorithm 1 in realistic setups, we consider the task of molecular conformation, i.e., we aim to reconstruct the 3D structure of a molecule from partial information about the distances of its atoms. For our experiments, we determine the structures of a protein molecule (1BPM) from the Protein Data Bank [HMBFGG+00], which is collected using X-ray diffraction experiments or nuclear magnetic resonance (NMR) spectroscopy.

The goal of this experiment is to reconstruct the point matrix $\mathbf{P}^0$ from $|\Omega|$ distance samples as defined in Problem 1. For the 1BPM protein data the rank of the input matrix is 3. We conduct the experiments corresponding to oversampling factors $\rho$ between 1 and 4. Like the previous setup, here successful recovery refers to the case where the relative Procrustes distance, that is the metric $d_{\text{Procrustes}}(\mathbf{P}_{rec}, \mathbf{P}^0)$ between the recovered matrix $\mathbf{P}_{rec}$ and ground truth coordinate matrix $\mathbf{P}^0$ does not exceed a tolerance threshold $\text{tol}_{\text{rec}}$ across 24 independent realizations. The error analysis for this experiment is shown in Figures 7a and 7b in Appendix E.

It is evident that MatrixIRLS and ALM have comparable results in recovering the geometry of the data given lesser samples, where as algorithms like ScaledSGD or RieEDG are unable to reconstruct geometries even when the oversampling rate is as high as $\rho = 4$. In the fig. 5, we see a convergence

with high precision for MatrixIRLS from $\rho \sim 2.5$ for Protein which means that it recovered the ground truth with around $0.5\%$ samples.

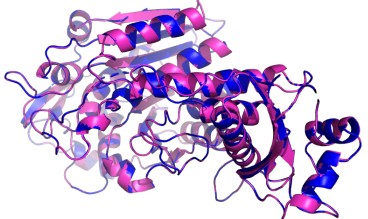

Figure 5: Reconstruction of Protein 1BPM molecule by MatrixIRLS with $0.5\%$ samples

Figure 6: Reconstruction of Protein 1BPM molecule by MatrixIRLS with $0.6\%$ samples

In the Figure 6 the reconstructed protein in blue is aligned with the ground truth structure of the protein in pink. We can see that when the oversampling is 3.5, which is when there is $0.6\%$ samples available, the reconstruction exactly matches with the ground truth.

In order to understand the runtime of the different algorithms Table 2, reports the reconstruction times for each of the four algorithms applied to the 1BPM Protein data (with datapoints $n = 3672$) in a low-data regime with oversampling factor of $\rho = 3$. . It can be seen that MatrixIRLS is with 7.08 minutes around 3 times faster than ALM, which needed 23.04 minutes until convergence. While these times certainly depend on the precise choice of stopping criteria for each algorithm (we used the ones indicated in Appendix E ), this shows that the proposed method is competitive in terms of clock time.

| Datapoints $n$ | Relative Error | Time (mins) |
|---|---|---|
| 500 | $1.04 \times 10^{-7}$ | 0.03 |
| 1000 | $1.19 \times 10^{-7}$ | 0.07 |
| 3000 | $1.37 \times 10^{-12}$ | 0.8 |
| 5000 | $8.56 \times 10^{-13}$ | 5.6 |
| 7000 | $2.26 \times 10^{-12}$ | 8.5 |
| 10000 | $4.06 \times 10^{-11}$ | 13.7 |

Table 1: Execution time of MatrixIRLS vs problem size $n$ for Gaussian data with over-sampling factor $\rho = 3$

| Algorithm | Relative Error | Time until Convergence (mins) |
|---|---|---|
| ALM | $5.6 \times 10^{-6}$ | 23.4 |
| RieEDG | $5.6 \times 10^{-2}$ | 116.6 |
| ScaledSGD | $2.4 \times 10^{-2}$ | 20.1 |
| **MatrixIRLS** | $2.7 \times 10^{-12}$ | **7.08** |

Table 2: Runtime comparison for different geometry reconstruction algorithms from partially known pairwise distances for 1BPM Protein data ($n = 3672, r = 3$), oversampling factor $\rho = 3$

For the implementation of the other algorithms, we use the authors' code for the respective approaches (discussed in Appendix E). We include another set of experiments on US cities data [UU20], in the Appendix E.

## 6   Conclusion

In this paper, we address the challenge of reconstructing suitable geometric configurations using minimal Euclidean distance samples. By leveraging continuous and non-convex rank minimization formulations, we develop a variant of the iteratively reweighted least squares (IRLS) algorithm and establish a local convergence guarantee under the condition of a minimal random set of observed distances. Our contribution also includes the proof of a restricted isometry property (RIP) restricted to the tangent space of the manifold of symmetric rank-$r$ matrices, a result which might be of independent interest for the analysis of other non-convex methods. As future work further analysis on the global convergence can be established. Through numerical validation we conclude that the algorithm is able to achieve state-of-the-art data efficiency.

## Acknowledgement

Abiy Tasissa acknowledges partial support from the National Science Foundation through grant DMS-2208392.

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

# Supplementary Material

## A   Limitations

For our method, the limited convergence radius leads to the convergence guarantee locally. Additionally, the runtime per iteration is larger compared to other non-convex methods [ZCZ22, TL18, SLCT23] but our method achieves convergence in less than a tenth of the number of iterations of the other methods. For some datasets like the Protein (1BPM) [HMBFGG$^+$00], ALM has a superior performance for oversampling factor ($\rho$) between 1 and 1.5.

## B   Dual Basis Construction

In this section, we provide the proof the bi-orthogonality of the basis defined in Lemma 4.4 with respect to the measurement basis Definition 3.2, and furthermore establish properties of the basis pair in Appendix B.2 that will be useful later on.

### B.1   Proof of Lemma 4.4

*Proof of Lemma 4.4.* For simplicity of the proof, we denote the two cases of Definition 3.2, separately. This helps us to divide the proof into multiple subcases.

With $L = \frac{n(n-1)}{2}$, let $\mathbf{V} = \begin{bmatrix} v_{(1,2)}, & \cdots, & v_{(n-1,n)} \end{bmatrix} \in \mathbb{R}^{n^2 \times L}$ be a matrix whose columns enumerate the vectorized dual basis elements $\mathbf{v}_\alpha$ where $\alpha = (i,j) \in \mathbb{I}$, and let $\mathbf{V}_E \in \mathbb{R}^{n^2 \times n}$ consist of vectorized dual basis vectors $\mathbf{v}_{(i,i)}$ with $i \in \{1, \ldots, n\}$. Similarly, $\mathbf{W} \in \mathbb{R}^{n^2 \times L}$ and $\mathbf{W}_E \in \mathbb{R}^{n^2 \times n}$ are defined using vectorized basis elements $\{\mathbf{w}_\alpha\}_{\alpha \in \mathbb{I} \cup \mathbb{I}_D}$. As establishing duality between the two basis sets is equivalent to bi-orthogonality of the basis elements, it remains to show that the extended basis matrices $\tilde{\mathbf{V}} = [\mathbf{V}|\mathbf{V}_E] \in \mathbb{R}^{n^2 \times L+n}$ and $\tilde{\mathbf{W}} = [\mathbf{W}|\mathbf{W}_E] \in \mathbb{R}^{n^2 \times L+n}$ satisfy the relationship

$$\tilde{\mathbf{W}}^\top \tilde{\mathbf{V}} = \begin{bmatrix} \mathbf{W}^\top \mathbf{V} & \mathbf{W}^\top \mathbf{V}_E \\ \mathbf{W}_E^\top \mathbf{V} & \mathbf{W}_E^\top \mathbf{V}_E \end{bmatrix} = \mathrm{Id},$$

where Id is the identity matrix.

In coordinate-wise notation, this means that for $\alpha = (i,j)$ and $\beta = (k,l)$, with $1 \leq i \leq j \leq n$ and $1 \leq k \leq l \leq n$, we must show that $\langle \tilde{\mathbf{w}}_\alpha, \tilde{\mathbf{v}}_\beta \rangle = \delta_{\alpha,\beta}$ where $\tilde{\mathbf{w}}_\alpha$ and $\tilde{\mathbf{v}}_\beta$ are columns of $\tilde{\mathbf{W}}$ and $\tilde{\mathbf{V}}$ respectively.

From Claim 3.1 of [LT24], it is clear that $\mathbf{V}$ is the dual of $\mathbf{W}$. We require to show the biorthogonality for the extended parts.

First we show that, for $\alpha$ of the form $(i,i)$,

$$
\begin{aligned}
\langle \mathbf{w}_\alpha, \mathbf{v}_\alpha \rangle &= \langle \mathbf{w}_{i,i}, \mathbf{v}_{i,i} \rangle \\
&= \frac{1}{2} \langle \mathbf{e}_i \cdot 1^\top + 1 \cdot \mathbf{e}_i^\top, \mathbf{e}_i \mathbf{e}_i^\top - \mathbf{a}_i \mathbf{a}_i^\top \rangle \\
&= \frac{1}{2} (\langle \mathbf{e}_i \cdot 1^\top, \mathbf{e}_i \mathbf{e}_i^\top \rangle + \langle 1 \cdot \mathbf{e}_i^\top, \mathbf{e}_i \mathbf{e}_i^\top \rangle - \langle \mathbf{e}_i \cdot 1^\top, \mathbf{a}_i \mathbf{a}_i^\top \rangle - \langle 1 \cdot \mathbf{e}_i^\top, \mathbf{a}_i \mathbf{a}_i^\top \rangle
\end{aligned}
\tag{10}
$$

Let us decompose the summand $\langle \mathbf{e}_i \cdot 1^\top, \mathbf{a}_i \mathbf{a}_i^\top \rangle$ in detail, so that we can derive the remaining calculations in a similar manner,

$$
\begin{aligned}
\langle \mathbf{e}_i \cdot 1^\top, \mathbf{a}_i \mathbf{a}_i^\top \rangle &= \langle \mathbf{e}_i \cdot 1^\top, \mathbf{e}_i \mathbf{e}_i^\top \rangle - \langle \mathbf{e}_i \cdot 1^\top, \frac{1}{n} 1 \mathbf{e}_i^\top \rangle - \langle \mathbf{e}_i \cdot 1^\top, \frac{1}{n} \mathbf{e}_i 1^\top \rangle + \langle \mathbf{e}_i \cdot 1^\top, \frac{1}{n^2} 1 \cdot 1^\top \rangle \\
&= 1 - \frac{1}{n} - 1 + \frac{1}{n} \\
&= 0.
\end{aligned}
$$

Putting this value back in (10), we can complete the rest of the calculation in a similar manner and finally arrive at the following,

$$
\begin{aligned}
\langle \mathbf{w}_\alpha, \mathbf{v}_\alpha \rangle &= \langle \mathbf{w}_{i,i}, \mathbf{v}_{i,i} \rangle \\
&= \frac{1}{2}(\langle \mathbf{e}_i \cdot 1^\top, \mathbf{e}_i \mathbf{e}_i^\top \rangle + \langle 1 \cdot \mathbf{e}_i^\top, \mathbf{e}_i \mathbf{e}_i^\top \rangle - \langle \mathbf{e}_i \cdot 1^\top, \mathbf{a}_i \mathbf{a}_i^\top \rangle + \langle 1 \cdot \mathbf{e}_i^\top, \mathbf{a}_i \mathbf{a}_i^\top \rangle \\
&= \frac{1}{2}(1 + 1 - 0 - 0) \\
&= 1.
\end{aligned}
$$

Now, let us look at the case where $\alpha = (i,i)$, and $\beta = (j,j)$ and $i \neq j$. Using a similar calculation we can show that,

$$
\begin{aligned}
\langle \mathbf{w}_\alpha, \mathbf{v}_\beta \rangle &= \langle \mathbf{w}_{i,i}, \mathbf{v}_{j,j} \rangle \\
&= \frac{1}{2} \langle \mathbf{e}_i \cdot 1^\top - 1 \cdot \mathbf{e}_i^\top, \mathbf{e}_j \mathbf{e}_j^\top - \mathbf{a}_j \mathbf{a}_j^\top \rangle \\
&= \frac{1}{2}(\langle \mathbf{e}_i \cdot 1^\top, \mathbf{e}_j \mathbf{e}_j^\top \rangle - \langle 1 \cdot \mathbf{e}_i^\top, \mathbf{e}_j \mathbf{e}_j^\top \rangle - \langle \mathbf{e}_i \cdot 1^\top, \mathbf{a}_j \mathbf{a}_j^\top \rangle + \langle 1 \cdot \mathbf{e}_i^\top, \mathbf{a}_j \mathbf{a}_j^\top \rangle \\
&= \frac{1}{2}(0 - 0 - 0 + 0) \\
&= 0.
\end{aligned}
$$

Also, between the $\mathbf{w}_\alpha$ and $\mathbf{v}_\beta$ where $\alpha = (i,j)$ with $1 \leq i < j \leq n$ and $\beta = (k,k)$ with $1 \leq k \leq n$, we can show a similar relation by

$$
\begin{aligned}
\langle \mathbf{w}_\alpha, \mathbf{v}_\beta \rangle &= \langle \mathbf{w}_{i,j}, \mathbf{v}_{k,k} \rangle \\
&= \frac{1}{2} \langle \mathbf{e}_{i,i} + \mathbf{e}_{j,j} - \mathbf{e}_{i,j} - \mathbf{e}_{j,i}, \mathbf{e}_k \mathbf{e}_k^\top - \mathbf{a}_k \mathbf{e}_k^\top \rangle \\
&= \frac{1}{2}\left( \delta_{i,k} + \delta_{j,k} - 0 - 0 - (\delta_{i,k} - \frac{1}{n})(\delta_{i,k} - \frac{1}{n}) - (\delta_{j,k} - \frac{1}{n})(\delta_{j,k} - \frac{1}{n}) \right) \\
&\quad + \frac{1}{2}\left( (\delta_{i,k} - \frac{1}{n})(\delta_{j,k} - \frac{1}{n}) \right) \\
&= \frac{1}{2}\left( \delta_{i,k} + \delta_{j,k} - \delta_{i,k}\delta_{i,k} + \frac{1}{n}\delta_{i,k} + \frac{1}{n}\delta_{j,k} - \frac{1}{n^2} - \delta_{j,k}\delta_{j,k} + \frac{1}{n}\delta_{j,k} + \frac{1}{n}\delta_{j,k} \right) \\
&\quad + \frac{1}{2}\left( -\frac{1}{n^2} + 2\delta_{i,k}\delta_{j,k} - 2\frac{1}{n}\delta_{j,k} - 2\delta_{i,k} + \frac{2}{n^2} \right) \\
&= \frac{1}{2}(2\delta_{i,k}\delta_{j,k}) \\
&= \delta_{i,j} \\
&= 0.
\end{aligned}
$$

Finally, we need to show for $\mathbf{w}_\alpha$ with $\alpha = (i,i)$ for $1 \leq i \leq n$ and $\mathbf{v}_\beta$ where $\beta = (k,l)$ with $1 \leq k < l \leq n$ that the respective dot product is 0. This is verified by the calculation

$$
\begin{aligned}
\langle \mathbf{w}_\alpha, \mathbf{v}_\beta \rangle &= -\frac{1}{4} \langle \mathbf{e}_i \cdot 1^\top + 1 \cdot \mathbf{e}_i^\top, \mathbf{e}_k \mathbf{e}_l^\top + \mathbf{e}_l \mathbf{e}_k^\top \rangle \\
&= -\frac{1}{4}(\langle \mathbf{e}_i \cdot 1^\top, \mathbf{e}_k \mathbf{e}_l^\top \rangle + \langle \mathbf{e}_i \cdot 1^\top, \mathbf{e}_l \mathbf{e}_k^\top \rangle + \langle 1 \cdot \mathbf{e}_i^\top, \mathbf{e}_k \mathbf{e}_l^\top \rangle + \langle 1 \cdot \mathbf{e}_i^\top, \mathbf{e}_l \mathbf{e}_k^\top \rangle) \\
&= -\frac{1}{4}\left( \delta_{ik}\delta_{il} + \delta_{ik}\delta_{il} + \delta_{ik}\delta_{il} + \delta_{ik}\delta_{il} \right) \\
&= -\frac{1}{4}\left( \delta_{ik}\delta_{il} + \delta_{ik}\delta_{il} + \delta_{ik}\delta_{il} + \delta_{ik}\delta_{il} \right) \\
&= -\frac{1}{4}\left( 4\delta_{ik}\delta_{il} \right) \\
&= -\delta_{ik}\delta_{il} \\
&= -\delta_{kl} \\
&= 0
\end{aligned}
$$

Hence combining the above parts we have proved that $\tilde{\mathbf{V}}$ is the dual of $\tilde{\mathbf{W}}$. □

## B.2 Properties of Dual Basis

**Lemma B.1.** $\tilde{\mathbf{W}}\tilde{\mathbf{V}}^\top$ *maps* $n \times n$ *symmetric matrices to itself.*

*Proof.* Let $\mathbf{x}$ be a vectorized representation of a symmetric matrix $\mathbf{X}$. Then, using the dual basis expansion, $\mathbf{x}$ can be represented as: $\mathbf{x} = \tilde{\mathbf{W}}\tilde{\mathbf{V}}^T\mathbf{x}$. We now apply the the operator $\tilde{\mathbf{W}}\tilde{\mathbf{V}}^T$ to $\mathbf{x}$.

$$(\tilde{\mathbf{W}}\tilde{\mathbf{V}}^T)\tilde{\mathbf{W}}\tilde{\mathbf{V}}^T\mathbf{x} = \tilde{\mathbf{W}}\tilde{\mathbf{V}}^{\mathbf{T}}\mathbf{x}.$$

where the last equality follows from the fact that $\tilde{\mathbf{V}}$ is dual to $\tilde{\mathbf{W}}$. □

**Lemma B.2.** *For orthonormal basis* $\{\mathbf{w}_\alpha\}$ *and* $\{\mathbf{v}_\alpha\}$ *as defined in Definition 3.2 , the spectral norm of* $\tilde{\mathbf{W}}\tilde{\mathbf{V}}^\top$ *is 1, that is*

$$\left\|\tilde{\mathbf{W}}\tilde{\mathbf{V}}^\top\right\|_{S_\infty} = 1. \tag{11}$$

*Proof.* By definition, $\|\tilde{\mathbf{W}}\tilde{\mathbf{V}}^T\|_{S_\infty} = \max_{\|x\|_2=1} \|\tilde{\mathbf{W}}\tilde{\mathbf{V}}^T\mathbf{x}\|_2$. Note that

$$\|\tilde{\mathbf{W}}\tilde{\mathbf{V}}^T\mathbf{x}\|_2^2 = \mathbf{x}^T\tilde{\mathbf{V}}(\tilde{\mathbf{W}}^T\tilde{\mathbf{W}})\tilde{\mathbf{V}}^T\mathbf{x}.$$

By construction of the dual basis, $(\tilde{\mathbf{W}}^T\tilde{\mathbf{W}}) = (\tilde{\mathbf{V}}^T\tilde{\mathbf{V}})^{-1}$. Therefore, $\tilde{\mathbf{V}}(\tilde{\mathbf{W}}^T\tilde{\mathbf{W}})\tilde{\mathbf{V}}^T$ is an orthogonal projection operator onto the column space of $\tilde{\mathbf{V}}$. If $\mathbf{x}$ is a vectorized representation of any symmetric matrix, using Lemma B.1 the projection operator is an identity operator. Hence, its operator norm is 1.

□

## C Sampling Operator Bounds and Proof of Tangent Space Restricted Isometry Properties

In this section, we first state the matrix Bernstein [Tro11] concentration inequality for rectangular matrices, which is repeatedly used in the proofs further in the section. We then provide a proof for the bound of our sampling operator $\mathcal{Q}_\Omega$ and then we conclude this section by proving the RIP restricted to the tangent space of manifold of symmetric rank-$r$ matrices at the ground truth $\mathbf{X}^0$.

**Theorem C.1** (Matrix Bernstein for rectangular matrices [Tro11, Theorem 1.6])**.** *Consider the finite sequence* $\{\mathbf{Z_k}\}$ *of independent, random matrices with dimension* $d_1 \times d_2$. *Assume that each random matrix satisfies*

$$\mathbb{E}[\mathbf{Z_k}] = 0 \text{ and } \|\mathbf{Z_k}\| \leq R \text{ almost surely,}$$

*Define*

$$\sigma^2 := \max\left\{\left\|\sum_k \mathbb{E}[\mathbf{Z_k}\mathbf{Z_k^*}]\right\|, \left\|\sum_k \mathbb{E}[\mathbf{Z_k^*}\mathbf{Z_k}]\right\|\right\}$$

*Then, for all* $t \geq 0$,

$$P\left\{\left\|\sum_k \mathbf{Z_k}\right\| \geq t\right\} \leq (d_1 + d_2) \cdot \exp\left(\frac{-t^2/2}{\sigma^2 + Rt/3}\right) \tag{12}$$

**Lemma C.2** (Spectral Norm Bound for Sampling Operator $\mathcal{Q}_\Omega$)**.** *Let $n$ be the dimension of the input matrix and let* $\Omega = (i_\ell, j_\ell)_{\ell=1}^m \subset \mathbf{I}$ *be a multiset of double indices fulfilling* $m < L = n(n-1)/2$ *that are sampled independently with replacement. Consequently, we have that with probability of at least* $1 - \frac{2}{n^2}$, *the sampling operator* $\mathcal{Q}_\Omega : S_n \to S_n$ *of* (9)

*fulfills*

$$\|\mathcal{Q}_\Omega\|_{S_\infty} \leq 20L\sqrt{\frac{\log n}{m}} + 1.$$

*Proof.* We define,

$$\mathcal{R}_\Omega' = \mathbf{V}\mathbf{S}_\Omega\mathbf{W}^\top$$
$$= \sum_{\alpha \in \Omega} \mathbf{v}_\alpha \mathbf{w}_\alpha^\top \tag{13}$$

So from (13),

$$\mathcal{R}_\Omega = \frac{L}{m}\mathcal{R}_\Omega'$$
$$\mathcal{Q}_\Omega = \mathcal{R}_\Omega + \mathbf{V}_E\mathbf{W}_E^\top$$
$$\mathcal{Q}_\Omega' = \mathcal{R}_\Omega' + \frac{L}{m}\mathbf{V}_E\mathbf{W}_E^\top$$

In order to provide a bound for $\mathcal{Q}_\Omega$, we first evaluate a bound for $\mathcal{R}_\Omega$. We would use concentration inequality of Theorem C.1, for computing the bound of $\mathcal{Q}_\Omega$

Let us define

$$\mathbf{z}_k = \frac{L}{m}\mathbf{v}_{\alpha_k}\mathbf{w}_{\alpha_k}^\top - \frac{1}{m}\mathbf{V}\mathbf{W}^\top \tag{14}$$

We compute the expectation of $\mathbf{z}_k$ by,

$$\mathbb{E}[\mathbf{z}_k] = \mathbb{E}[\frac{L}{m}\mathbf{v}_{\alpha_k}\mathbf{w}_{\alpha_k}^\top - \frac{1}{m}\mathbf{V}\mathbf{W}^\top]$$
$$= \frac{1}{L}\sum_{k=1}^{L}\frac{L}{m}\mathbf{v}_{\alpha_k}\mathbf{w}_{\alpha_k}^\top - \frac{1}{m}\mathbf{V}\mathbf{W}^\top \tag{15}$$
$$= \frac{1}{m}\mathbf{V}\mathbf{W}^\top - \frac{1}{m}\mathbf{V}\mathbf{W}^\top$$
$$= 0$$

Now let us compute the bound for $\mathbf{z}_k$

$$\|\mathbf{z}_k\|_{S_\infty} = \left\|\frac{L}{m}\mathbf{v}_{\alpha_k}\mathbf{w}_{\alpha_k}^\top - \frac{1}{m}\mathbf{V}\mathbf{W}^\top\right\|_{S_\infty}$$
$$\leq \left\|\frac{L}{m}\mathbf{v}_{\alpha_k}\mathbf{w}_{\alpha_k}^\top\right\|_{S_\infty} + \frac{1}{m}\left\|\mathbf{V}\mathbf{W}^\top\right\|_{S_\infty} \tag{16}$$
$$\leq \left\|\frac{L}{m}\mathbf{v}_{\alpha_k}\mathbf{w}_{\alpha_k}^\top\right\|_{S_\infty} + \frac{1}{m}$$

We use triangle inequality in the first inequality and since the spectral norm of $\mathbf{V}\mathbf{W}^\top$ is 1, we arrive at the second inequality.

Now, we can further bound the spectral norm of $\mathbf{v}_{\alpha_k}\mathbf{w}_{\alpha_k}^\top$ by,

$$\|\mathbf{v}_{\alpha_k}\mathbf{w}_{\alpha_k}^\top\|_{S_\infty} = \max \frac{\langle \mathbf{V}, \mathbf{v}_{\alpha_k}\mathbf{w}_{\alpha_k}^\top\mathbf{W}\rangle}{\|\mathbf{V}\|_2\|\mathbf{W}\|_2}$$
$$\leq \frac{\|\mathbf{v}_{\alpha_k}\|_2^2\|\mathbf{w}_{\alpha_k}\|_2^2}{\|\mathbf{v}_{\alpha_k}\|_2\|\mathbf{w}_{\alpha_k}\|_2} \tag{17}$$
$$\leq 1 \cdot 4 = 4$$

So from (16), we get

$$\|\mathbf{z}_k\|_{S_\infty} \leq \frac{4L+1}{m} \tag{18}$$

We take the product of $\mathbf{z}_k$ and $\mathbf{z}_k^*$ as,

$$
\begin{aligned}
\mathbf{z}_k^* \mathbf{z}_k &= (\frac{L}{m} \mathbf{v}_{\alpha_k} \mathbf{w}_{\alpha_k}^\top - \frac{1}{m} \mathbf{V} \mathbf{W}^\top)^\top (\frac{L}{m} \mathbf{v}_{\alpha_k} \mathbf{w}_{\alpha_k}^\top - \frac{1}{m} \mathbf{V} \mathbf{W}^\top) \\
&= (\frac{L}{m} \mathbf{w}_{\alpha_k} \mathbf{v}_{\alpha_k}^\top - \frac{1}{m} \mathbf{W} \mathbf{V}^\top)(\frac{L}{m} \mathbf{v}_{\alpha_k} \mathbf{w}_{\alpha_k}^\top - \frac{1}{m} \mathbf{V} \mathbf{W}^\top) \\
&= \frac{L^2}{m^2} (\mathbf{w}_{\alpha_k} \mathbf{v}_{\alpha_k}^\top \mathbf{v}_{\alpha_k} \mathbf{w}_{\alpha_k}^\top) - \frac{L}{m^2} (\mathbf{w}_{\alpha_k} \mathbf{v}_{\alpha_k}^\top \mathbf{V} \mathbf{W}^\top) - \frac{L}{m^2} (\mathbf{W} \mathbf{V}^\top \mathbf{v}_{\alpha_k} \mathbf{w}_{\alpha_k}^\top) \\
&\quad + \frac{1}{m^2} (\mathbf{W} \mathbf{V}^\top \mathbf{V} \mathbf{W}^\top)
\end{aligned}
$$

Now taking expectation on both side of the above relation we get;

$$
\begin{aligned}
||\mathbb{E}(\mathbf{z}_k^* \mathbf{z}_k)||_{S_\infty} &= \left\| \mathbb{E}\left[\frac{L^2}{m^2}(\mathbf{w}_{\alpha_k} \mathbf{v}_{\alpha_k}^\top \mathbf{v}_{\alpha_k} \mathbf{w}_{\alpha_k}^\top)\right] + \mathbb{E}\left[\frac{L}{m^2}(\mathbf{w}_{\alpha_k} \mathbf{v}_{\alpha_k}^\top \mathbf{V} \mathbf{W}^\top)\right] \right. \\
&\quad \left. + \mathbb{E}\left[\frac{L}{m^2}(\mathbf{W} \mathbf{V}^\top \mathbf{v}_{\alpha_k} \mathbf{w}_{\alpha_k}^\top)\right] + \mathbb{E}\left[\frac{1}{m^2}(\mathbf{W} \mathbf{V}^\top \mathbf{V} \mathbf{W}^\top)\right] \right\|_{S_\infty} \\
&\leq \left\| \mathbb{E}\left[\frac{L^2}{m^2}(\mathbf{w}_{\alpha_k} \mathbf{v}_{\alpha_k}^\top \mathbf{v}_{\alpha_k} \mathbf{w}_{\alpha_k}^\top)\right] \right\|_{S_\infty} + \left\| \mathbb{E}\left[\frac{L}{m^2}(\mathbf{w}_{\alpha_k} \mathbf{v}_{\alpha_k}^\top \mathbf{V} \mathbf{W}^\top)\right] \right\|_{S_\infty} \\
&\quad + \left\| \mathbb{E}\left[\frac{L}{m^2}(\mathbf{W} \mathbf{V}^\top \mathbf{v}_{\alpha_k} \mathbf{w}_{\alpha_k}^\top)\right] \right\|_{S_\infty} + \left\| \mathbb{E}\left[\frac{1}{m^2}(\mathbf{W} \mathbf{V}^\top \mathbf{V} \mathbf{W}^\top)\right] \right\|_{S_\infty} \\
&= \left\| \frac{L^2}{m^2}\left[\frac{1}{L}\sum_{k=1}^L (\mathbf{w}_{\alpha_k} \mathbf{v}_{\alpha_k}^\top \mathbf{v}_{\alpha_k} \mathbf{w}_{\alpha_k}^\top)\right] \right\|_{S_\infty} + \left\| \frac{1}{L}\left[\sum_{k=1}^L \frac{L}{m^2}(\mathbf{w}_{\alpha_k} \mathbf{v}_{\alpha_k}^\top \mathbf{V} \mathbf{W}^\top)\right] \right\|_{S_\infty} \\
&\quad + \left\| \frac{1}{L}\left[\frac{L}{m^2}(\mathbf{w}_{\alpha_k} \mathbf{v}_{\alpha_k}^\top \mathbf{V} \mathbf{W}^\top)\right] \right\|_{S_\infty} + \frac{1}{m^2} \\
&\leq \frac{L^2}{m^2} \max ||\mathbf{w}_{\alpha_k} \mathbf{v}_{\alpha_k}^\top \mathbf{v}_{\alpha_k} \mathbf{w}_{\alpha_k}^\top||_{S_\infty} + \left\| \frac{L}{m^2} \mathbf{W} \mathbf{V}^\top \mathbf{V} \mathbf{W}^\top \right\|_{S_\infty} \\
&\quad + \left\| \frac{L}{m^2} \mathbf{W} \mathbf{V}^\top \mathbf{V} \mathbf{W}^\top \right\|_{S_\infty} + \frac{1}{m^2} \\
&\leq \frac{16L^2 + 2L + 1}{m^2}
\end{aligned}
$$

$$(19)$$

Since expectation is a linear operator we can preserve the equality in the first equality, then we use triangle inequality in the first inequality. We further use that the spectral norm of $\mathbf{W} \mathbf{V}^\top$ is 1 and arrive at the second inequality. Further using (17) we bound the expression in the last inequality. As per Theorem C.1 we now compute the product of $\mathbf{z}_k$ and $\mathbf{z}_k^*$

$$
\begin{aligned}
\mathbf{z}_k \mathbf{z}_k^* &= (\frac{L}{m} \mathbf{v}_{\alpha_k} \mathbf{w}_{\alpha_k}^\top - \frac{1}{m} \mathbf{V} \mathbf{W}^\top)(\frac{L}{m} \mathbf{v}_{\alpha_k} \mathbf{w}_{\alpha_k}^\top - \frac{1}{m} \mathbf{V} \mathbf{W}^\top)^\top \\
&= (\frac{L}{m} \mathbf{v}_{\alpha_k} \mathbf{w}_{\alpha_k}^\top - \frac{1}{m} \mathbf{V} \mathbf{W}^\top)(\frac{L}{m} \mathbf{w}_{\alpha_k} \mathbf{v}_{\alpha_k}^\top - \frac{1}{m} \mathbf{W} \mathbf{V}^\top) \\
&= \frac{L^2}{m^2} (\mathbf{v}_{\alpha_k} \mathbf{w}_{\alpha_k}^\top \mathbf{w}_{\alpha_k} \mathbf{v}_{\alpha_k}^\top) - \frac{L}{m^2} (\mathbf{v}_{\alpha_k} \mathbf{w}_{\alpha_k}^\top \mathbf{W} \mathbf{V}^\top) \\
&\quad - \frac{L}{m^2} (\mathbf{V} \mathbf{W}^\top \mathbf{w}_{\alpha_k} \mathbf{v}_{\alpha_k}^\top) + \frac{1}{m^2} (\mathbf{V} \mathbf{W}^\top \mathbf{W} \mathbf{V}^\top)
\end{aligned}
$$

Now taking expectation on both side of the above relation we get;

$$
\begin{aligned}
|| \, \mathbb{E}(\mathbf{z}_k \mathbf{z}_k^*)||_{S_\infty} &= \left\| \frac{L^2}{m^2} [ \frac{1}{L} \sum_{k=1}^{L} (\mathbf{v}_{\alpha_k} \mathbf{w}_{\alpha_k}^\top \mathbf{w}_{\alpha_k} \mathbf{v}_{\alpha_k}^\top)] \right\|_{S_\infty} \\
&+ \left\| \frac{1}{L} \sum_{k=1}^{L} \frac{L}{m^2} (\mathbf{v}_{\alpha_k} \mathbf{w}_{\alpha_k}^\top \mathbf{W} \mathbf{V}^\top)] \right\|_{S_\infty} \\
&+ \left\| \frac{1}{L} \frac{L}{m^2} (\mathbf{V} \mathbf{W}^\top \mathbf{w}_{\alpha_k} \mathbf{v}_{\alpha_k}^\top)] \right\|_{S_\infty} + \frac{1}{m^2} \\
&\leq \frac{L^2}{m^2} \max \left\| \mathbf{v}_{\alpha_k} \mathbf{w}_{\alpha_k}^\top \mathbf{w}_{\alpha_k} \mathbf{v}_{\alpha_k}^\top \right\|_{S_\infty} + \left\| \frac{L}{m^2} \mathbf{V} \mathbf{W}^\top \mathbf{W} \mathbf{V}^\top \right\|_{S_\infty} \\
&+ \left\| \frac{L}{m^2} \mathbf{V} \mathbf{W}^\top \mathbf{W} \mathbf{V}^\top \right\|_{S_\infty} + \frac{1}{m^2} \\
&\leq \frac{16L^2 + 2L + 1}{m^2}
\end{aligned}
\tag{20}
$$

So, now by Matrix Bernstein Inequality as stated Theorem C.1,

$$
\mathbb{E}(\mathbf{z}_k) = 0
$$

The value of the bound of $\mathbf{z}_k$ is denoted by R in Theorem C.1. So R in this case is

$$
||\mathbf{z}_k||_{S_\infty} \leq R = \frac{4L + 1}{m}
$$

Further $\sigma$ is defined by $\max[\sum_k \mathbb{E}(\mathbf{z}_k \mathbf{z}_k^*), \sum_k \mathbb{E}(\mathbf{z}_k^* \mathbf{z}_k)]$ as per Theorem C.1. We have,

$$
\sigma^2 = m \cdot \frac{16L^2 + 2L + 1}{m^2} = \frac{16L^2 + 2L + 1}{m}
$$

So as per Theorem C.1,

$$
\forall t > 0, P(|| \sum_{k=1}^{L} \mathbf{z}_k||_{S_\infty} \geq t) \leq 2n \exp(\frac{\frac{-t^2}{2}}{\sigma^2 + \frac{Rt}{3}})
\tag{21}
$$

Recalling the definition of $\mathbf{z}_k$ from (14)

$$
|| \sum_{k=1}^{L} \mathbf{z}_k||_{S_\infty} \leq t \implies ||\frac{L}{m} \mathcal{Q}_\Omega{}' - \mathbf{V} \mathbf{W}^\top||_{s_\infty} \leq t
$$

Therefore,

$$
\begin{aligned}
|| \, \mathcal{Q}_\Omega \, ||_{s_\infty} &= ||\frac{L}{m} \mathcal{Q}_\Omega{}' \, ||_{s_\infty} \\
&\leq ||\frac{L}{m} \mathcal{Q}_\Omega{}' - \mathbf{V} \mathbf{W}^\top||_{s_\infty} + ||\mathbf{V} \mathbf{W}^\top||_{s_\infty} \\
&\leq t + 1
\end{aligned}
\tag{22}
$$

with a probability $1 - 2n \exp(\frac{\frac{-t^2}{2}}{\sigma^2 + \frac{Rt}{3}})$. We have arrived at the above inequality from the definition of $\mathcal{Q}_\Omega$ and that the spectral norm of the extended basis of the dual pair, $\mathbf{V}_E \mathbf{W}_E^\top$ is 1.

Since $L = \frac{n(n-1)}{2}$ so, $L^2 \leq \frac{n^2 L}{2}$. We can further simplify the denominator by the following,

$$
\begin{aligned}
\left( \sigma^2 + \frac{Rt}{3} \right) &= \frac{16L^2 + 2L + 1}{m} + \frac{4L + 1}{3m} t \\
&\leq \frac{16L^2}{m} + \frac{2L}{m} + \frac{L}{m} + \frac{2Lt}{m} \\
&\leq \frac{L}{m} (16L + 3 + 2t)
\end{aligned}
\tag{23}
$$

We arrive at the first inequality by using $\frac{1}{m} \leq \frac{L}{m}$ and $\frac{4L+1}{3m} \leq \frac{6L}{3m}$

So using this relation, let us assume that $t = 20L\sqrt{\frac{\log n}{m}}$.

Further let us simply Equation (23), by using this value of t. Since by Theorem 4.3, $m \geq Kn \log n$, where $K = C\nu r$, so $\frac{\log n}{m} \leq \frac{1}{Kn}$, hence we can say simplify Equation (23), further by the following,

$$
\begin{aligned}
\left(\sigma^2 + \frac{Rt}{3}\right) &\leq \frac{L}{m}(16L + 3 + 2t) \\
&\leq \frac{L}{m}\left(16L + 3 + 2 \cdot 20L\sqrt{\frac{\log n}{m}}\right) \\
&\leq \frac{L}{m}\left(16L + 3 + 2 \cdot 20L\sqrt{\frac{1}{Kn}}\right) \\
&\leq \frac{L}{m}(60L)
\end{aligned}
\tag{24}
$$

We arrive at the third inequality bu using the expression for L and using $\frac{logn}{m} \leq \frac{1}{Kn}$.

Then, we can deduce the following from Equation (24),

$$
\begin{aligned}
\exp\left(\frac{-\frac{t^2}{2}}{\sigma^2 + \frac{Rt}{3}}\right) &= \exp\left(\frac{-\frac{20^2 L^2 \log n}{2m}}{\frac{60L^2}{m}}\right) \\
&\leq \exp\left(-\frac{400}{120}\log n\right) \\
&= \exp(3.33\log(\frac{1}{n})) \\
&\leq \exp(\log(\frac{1}{n^3})) \\
&= \frac{1}{n^3}
\end{aligned}
$$

So the probability in (21) becomes $2n \cdot \frac{1}{n^3}$ If we use (22), then we can conclude that that the spectral norm of $\mathcal{Q}_\Omega$ is bounded by $\left(20L\sqrt{\frac{\log n}{m}} + 1\right)$. Hence, we can say that the spectral norm of $\mathcal{Q}_\Omega$ is bounded by $\left(20L\sqrt{\frac{\log n}{m}} + 1\right)$ with a probability of $1 - \frac{2}{n^2}$. $\qquad\square$

Now we prove the restricted isometry property of the sampling operator $\mathcal{Q}_\Omega$ with respect to the tangent space of the rank-$r$ matrix manifold at the ground truth, as stated in Theorem 4.5.

*Proof of Theorem 4.5.* Define $z_k = \frac{L}{m}\mathcal{P}_{T_0}\mathbf{w}_{\alpha_k}\mathbf{v}_{\alpha_k}^\top\mathcal{P}_{T_0} - \frac{1}{m}\mathcal{P}_{T_0}\mathbf{W}\mathbf{V}^\top\mathcal{P}_{T_0}$.

Then,

$$
\begin{aligned}
\mathbb{E}(\mathbf{z}_k) &= \mathbb{E}\left(\frac{L}{m}\mathcal{P}_{T_0}\mathbf{w}_{\alpha_k}\mathbf{v}_{\alpha_k}^\top\mathcal{P}_{T_0} - \frac{1}{m}\mathcal{P}_{T_0}\mathbf{W}\mathbf{V}^\top\mathcal{P}_{T_0}\right) \\
&= \frac{1}{L}\sum_{\ell=1}^{L}\left(\frac{L}{m}\mathcal{P}_{T_0}\mathbf{w}_\ell\mathbf{v}_\ell^\top\mathcal{P}_{T_0} - \frac{1}{m}\mathcal{P}_{T_0}\mathbf{W}\mathbf{V}^\top\mathcal{P}_{T_0}\right) \\
&= \frac{1}{m}\mathcal{P}_{T_0}\mathbf{W}\mathbf{V}^\top\mathcal{P}_{T_0} - \frac{1}{m}\mathcal{P}_{T_0}\mathbf{W}\mathbf{V}^\top\mathcal{P}_{T_0} \\
&= 0
\end{aligned}
\tag{25}
$$

Now we calculate the spectral norm of $\mathbf{z}_k$

$$\begin{aligned}
||\mathbf{z}_k||_{S_\infty} &= \left\| \frac{L}{m} \mathcal{P}_{T_0} \mathbf{w}_{\alpha_k} \mathbf{v}_{\alpha_k}^\top \mathcal{P}_{T_0} - \frac{1}{m} \mathcal{P}_{T_0} \mathbf{W} \mathbf{V}^\top \mathcal{P}_{T_0} \right\|_{S_\infty} \\
&\leq \left\| \frac{L}{m} \mathcal{P}_{T_0} \mathbf{w}_{\alpha_k} \mathbf{v}_{\alpha_k}^\top \mathcal{P}_{T_0} \right\|_{S_\infty} + \frac{1}{m} \left\| \mathcal{P}_{T_0} \mathbf{W} \mathbf{V}^\top \mathcal{P}_{T_0} \right\|_{S_\infty} \\
&\leq \frac{L}{m} || \mathcal{P}_{T_0} \mathbf{w}_{\alpha_k} ||_F || \mathcal{P}_{T_0} \mathbf{v}_{\alpha_k} ||_F + \frac{1}{m} \\
&\leq \frac{2L}{m} \sqrt{\frac{2\nu r}{n}} \sqrt{\frac{2\nu r}{n}} + \frac{1}{m} \\
&\leq \frac{4\nu r L}{mn} + \frac{1}{m}
\end{aligned}$$
(26)

The third inequality follows from the coherence bounds as in Definition 4.1 .

$$\begin{aligned}
\mathbf{z}_k^* \mathbf{z}_k &= (\frac{L}{m} \mathcal{P}_{T_0} \mathbf{w}_{\alpha_k} \mathbf{v}_{\alpha_k}^\top \mathcal{P}_{T_0} - \frac{1}{m} \mathcal{P}_{T_0} \mathbf{W} \mathbf{V}^\top \mathcal{P}_{T_0})^\top (\frac{L}{m} \mathcal{P}_{T_0} \mathbf{w}_{\alpha_k} \mathbf{v}_{\alpha_k}^\top \mathcal{P}_{T_0} - \frac{1}{m} \mathcal{P}_{T_0} \mathbf{W} \mathbf{V}^\top \mathcal{P}_{T_0}) \\
&= (\frac{L^2}{m^2} \mathcal{P}_{T_0} \mathbf{v}_{\alpha_k} \mathbf{w}_{\alpha_k}^\top \mathcal{P}_{T_0} \mathbf{w}_{\alpha_k} \mathbf{v}_{\alpha_k}^\top \mathcal{P}_{T_0}) - (\frac{L}{m^2} \mathcal{P}_{T_0} \mathbf{v}_{\alpha_k} \mathbf{w}_{\alpha_k}^\top \mathcal{P}_{T_0} \mathcal{P}_{T_0} \mathbf{W} \mathbf{V}^\top \mathcal{P}_{T_0}) \\
&\quad - (\frac{L}{m^2} \mathcal{P}_{T_0} \mathbf{V} \mathbf{W}^\top \mathcal{P}_{T_0} \mathcal{P}_{T_0} \mathbf{w}_{\alpha_k} \mathbf{v}_{\alpha_k}^\top \mathcal{P}_{T_0}) + \frac{1}{m^2} \mathcal{P}_{T_0} \mathbf{V} \mathbf{W}^\top \mathcal{P}_{T_0} \mathcal{P}_{T_0} \mathbf{W} \mathbf{V}^\top \mathcal{P}_{T_0}
\end{aligned}$$

We use that $\mathcal{P}_{T_0}^2 = \mathcal{P}_{T_0}$ in the second equality to further calculate the expectation of the above equality,

$$\begin{aligned}
\mathbb{E}(\mathbf{z}_k^* \mathbf{z}_k) &= \mathbb{E}(\frac{L^2}{m^2} \mathcal{P}_{T_0} \mathbf{v}_{\alpha_k} \mathbf{w}_{\alpha_k}^\top \mathcal{P}_{T_0} \mathbf{w}_{\alpha_k} \mathbf{v}_{\alpha_k}^\top \mathcal{P}_{T_0}) - \mathbb{E}((\frac{L}{m^2} \mathcal{P}_{T_0} \mathbf{v}_{\alpha_k} \mathbf{w}_{\alpha_k}^\top \mathcal{P}_{T_0} \mathcal{P}_{T_0} \mathbf{W} \mathbf{V}^\top \mathcal{P}_{T_0}) \\
&\quad - \mathbb{E}((\frac{L}{m^2} \mathcal{P}_{T_0} \mathbf{V} \mathbf{W}^\top \mathcal{P}_{T_0} \mathcal{P}_{T_0} \mathbf{w}_{\alpha_k} \mathbf{v}_{\alpha_k}^\top \mathcal{P}_{T_0})) + \mathbb{E}(\frac{1}{m^2} \mathcal{P}_{T_0} \mathbf{V} \mathbf{W}^\top \mathcal{P}_{T_0} \mathcal{P}_{T_0} \mathbf{W} \mathbf{V}^\top \mathcal{P}_{T_0}) \\
&= \mathbb{E}(\frac{L^2}{m^2} \mathcal{P}_{T_0} \mathbf{v}_{\alpha_k} \mathbf{w}_{\alpha_k}^\top \mathcal{P}_{T_0} \mathbf{w}_{\alpha_k} \mathbf{v}_{\alpha_k}^\top \mathcal{P}_{T_0}) - \frac{1}{m^2} \mathcal{P}_{T_0} \mathbf{V} \mathbf{W}^\top \mathcal{P}_{T_0} \mathbf{W} \mathbf{V}^\top \mathcal{P}_{T_0} \\
&\quad - \frac{1}{m^2} \mathcal{P}_{T_0} \mathbf{V} \mathbf{W}^\top \mathcal{P}_{T_0} \mathcal{P}_{T_0} \mathbf{W} \mathbf{V}^\top \mathcal{P}_{T_0} + \frac{1}{m^2} \mathcal{P}_{T_0} \mathbf{V} \mathbf{W}^\top \mathcal{P}_{T_0} \mathcal{P}_{T_0} \mathbf{W} \mathbf{V}^\top \mathcal{P}_{T_0} \\
&= \mathbb{E}(\frac{L^2}{m^2} \mathcal{P}_{T_0} \mathbf{v}_{\alpha_k} \mathbf{w}_{\alpha_k}^\top \mathcal{P}_{T_0} \mathbf{w}_{\alpha_k} \mathbf{v}_{\alpha_k}^\top \mathcal{P}_{T_0}) - \frac{1}{m^2} \mathcal{P}_{T_0} \mathbf{V} \mathbf{W}^\top \mathcal{P}_{T_0} \mathcal{P}_{T_0} \mathbf{W} \mathbf{V}^\top \mathcal{P}_{T_0}
\end{aligned}$$

since $\tilde{\mathbf{W}} \tilde{\mathbf{V}}^\top = Id$, we can adjust summand at the seocond equality.
Defining the random operator $\widetilde{\mathbf{z}}_k := \frac{L}{m} \mathcal{P}_{T_0} \mathbf{w}_{\alpha_k} \mathbf{v}_{\alpha_k}^\top \mathcal{P}_{T_0}$, we see that $\mathbf{z}_k$ from above satisfies $\mathbf{z}_k = \widetilde{\mathbf{z}}_k - \frac{1}{m} \mathcal{P}_{T_0} \mathbf{W} \mathbf{V}^\top \mathcal{P}_{T_0}$ and therefore

$$\mathbb{E}[\widetilde{\mathbf{z}}_k] = \frac{1}{m} \mathcal{P}_{T_0} \mathbf{W} \mathbf{V}^\top \mathcal{P}_{T_0}$$

With this definition, and for the fact that positive semi-definite matrices satisfies

$\|A - B\|_{S_\infty} \leq \max(\|A\|_{S_\infty}, \|B\|_{S_\infty})$, we can bound the spectral norm of the expectation in the following way

$$\begin{aligned}
\| \mathbb{E}(\mathbf{z}_k^* \mathbf{z}_k) \|_{S_\infty} &= \left\| \mathbb{E}[\widetilde{\mathbf{z}}_k^* \widetilde{\mathbf{z}}_k] - \frac{1}{m^2} \mathcal{P}_{T_0} \mathbf{W} \mathbf{V}^\top \mathcal{P}_{T_0} \mathcal{P}_{T_0} \mathbf{V} \mathbf{W}^\top \mathcal{P}_{T_0} \right\|_{S_\infty} \\
&\leq \max \left( \left\| \mathbb{E}[\widetilde{\mathbf{z}}_k^* \widetilde{\mathbf{z}}_k] \right\|_{S_\infty}, \frac{1}{m^2} \right)
\end{aligned}$$
(27)

For any $\mathbf{M} \in \mathbb{R}^{n \times n}$, it follows from the definition of $\widetilde{\mathbf{z}_k}$

$$\widetilde{z}_k(\mathbf{M}) = \frac{L}{m} \langle \mathbf{v}_{\alpha_k}, \mathcal{P}_{T_0}(\mathbf{M}) \rangle \mathcal{P}_{T_0}(\mathbf{w}_{\alpha_k}) = \frac{L}{m} \langle \mathcal{P}_{T_0}(\mathbf{v}_{\alpha_k}), \mathbf{M} \rangle \mathcal{P}_{T_0}(\mathbf{w}_{\alpha_k}),$$

$$\widetilde{z}_k^*(\mathbf{M}) = \frac{L}{m}\langle \mathbf{w}_{\alpha_k}, \mathcal{P}_{T_0}(\mathbf{M})\rangle \mathcal{P}_{T_0}(\mathbf{v}_{\alpha_k}) = \frac{L}{m}\langle \mathcal{P}_{T_0}(\mathbf{w}_{\alpha_k}), \mathbf{M}\rangle \mathcal{P}_{T_0}(\mathbf{v}_{\alpha_k})$$

and thus

$$
\begin{aligned}
\widetilde{\mathbf{z}}_k^* \widetilde{\mathbf{z}}_k(\mathbf{M}) &= \frac{L}{m}\langle \mathcal{P}_{T_0}(\mathbf{w}_{\alpha_k}), \widetilde{\mathbf{z}}_k(\mathbf{M})\rangle \mathcal{P}_{T_0}(\mathbf{v}_{\alpha_k}) \\
&= \frac{L^2}{m^2}\langle \mathcal{P}_{T_0}(\mathbf{w}_{\alpha_k}), \langle \mathcal{P}_{T_0}(\mathbf{v}_{\alpha_k}), \mathbf{M}\rangle \mathcal{P}_{T_0}(\mathbf{w}_{\alpha_k})\rangle \mathcal{P}_{T_0}(\mathbf{v}_{\alpha_k}) \\
&= \frac{L^2}{m^2}\|\mathcal{P}_{T_0}(\mathbf{w}_{\alpha_k})\|_F^2 \langle \mathcal{P}_{T_0}(\mathbf{v}_{\alpha_k}), \mathbf{M}\rangle \mathcal{P}_{T_0}(\mathbf{v}_{\alpha_k})
\end{aligned}
$$

Now using the incoherence condition such as Definition 4.1 and [TL18, Lemma 22]. we can argue that

$$
\begin{aligned}
\left\|\mathbb{E}\left[\widetilde{\mathbf{z}}_k^* \widetilde{\mathbf{z}}_k\right]\right\|_{S_\infty} &\leq \frac{L^2}{m^2} \max_{\ell=1}^{L} \|\mathcal{P}_{T_0}(\mathbf{w}_\ell)\|_F^2 \left\|\mathbb{E}\left[\mathcal{P}_{T_0}\mathbf{v}_{\alpha_k}\mathbf{v}_{\alpha_k}^\top \mathcal{P}_{T_0}\right]\right\|_{S_\infty} \\
&= \frac{L^2}{m^2} \max_{\ell=1}^{L} \|\mathcal{P}_{T_0}(\mathbf{w}_\ell)\|_F^2 \left\|\frac{1}{L}\sum_{\ell=1}^{L}\mathcal{P}_{T_0}\mathbf{v}_\ell\mathbf{v}_\ell^\top \mathcal{P}_{T_0}\right\|_{S_\infty} \\
&\leq \frac{L^2}{m^2} \max_{\ell=1}^{L} \|\mathcal{P}_{T_0}(\mathbf{w}_\ell)\|_F^2 \max_{\ell=1}^{L} \|\mathcal{P}_{T_0}(\mathbf{v}_{\alpha_k})\|_{s_\infty}^2 \\
&\leq \frac{L^2}{m^2}(2\nu\frac{r}{n})\frac{1}{L}(8\nu\frac{r}{n}) \\
&= \frac{L}{m^2}16\frac{\nu^2 r^2}{n^2}.
\end{aligned}
$$

Now we get from (27)

$$\|\mathbb{E}(\mathbf{z}_k^*\mathbf{z}_k)\|_{S_\infty} \leq \max\left(\frac{L}{m^2}(16\frac{\nu^2 r^2}{n^2}), \frac{1}{m^2}\right) \tag{28}$$

So by Triangle Inequality,

$$\left\|\sum_{k=1}^{m}\mathbb{E}(\mathbf{z}_k^*\mathbf{z}_k)\right\|_{S_\infty} \leq \sum_{k=1}^{m}\|\mathbb{E}(\mathbf{z}_k^*\mathbf{z}_k)\|_{S_\infty} \leq m\cdot\frac{L}{m^2}\left(16\frac{\nu^2 r^2}{n^2}\right) = \frac{L}{m}\left(16\frac{\nu^2 r^2}{n^2}\right) \tag{29}$$

Now we need to compute similar bounds for $\mathbb{E}[\mathbf{z}_k\mathbf{z}_k^*]$.

$$
\begin{aligned}
\|\mathbb{E}[\widetilde{\mathbf{z}}_k\widetilde{\mathbf{z}}_k^*]\|_{s_\infty} &= \left\|\mathbb{E}\left[\frac{L^2}{m^2}\mathcal{P}_{T_0}\mathbf{w}_{\alpha_k}\mathbf{v}_{\alpha_k}^\top \mathcal{P}_{T_0}\mathbf{v}_{\alpha_k}\mathbf{w}_{\alpha_k}^\top \mathcal{P}_{T_0}\right]\right\|_{s_\infty} \\
&= \left\|\frac{L^2}{m^2}\sum_{\ell=1}^{L}\frac{1}{L}\mathcal{P}_{T_0}\mathbf{w}_\ell\mathbf{v}_\ell^\top \mathcal{P}_{T_0}\mathbf{v}_\ell\mathbf{w}_\ell^\top \mathcal{P}_{T_0}\right\|_{s_\infty} \\
&\leq \frac{L}{m}\max_\ell\|\mathcal{P}_{T_0}\mathbf{w}_\ell\mathbf{v}_\ell^\top \mathcal{P}_{T_0}\mathbf{v}_\ell\mathbf{w}_\ell^\top \mathcal{P}_{T_0}\|_{s_\infty} \\
&\leq \frac{L}{m}\max_\ell\|\mathcal{P}_{T_0}\mathbf{w}_{\alpha_k}\|_{s_\infty}^2 \max_\ell\|\mathcal{P}_{T_0}\mathbf{v}_{\alpha_k}\|_{s_\infty}^2 \\
&\leq \frac{L}{m^2}(2\nu\frac{r}{n})(8\nu\frac{r}{n}) \\
&= \frac{L}{m^2}16\frac{\nu^2 r^2}{n^2}
\end{aligned}
\tag{30}
$$

So similar to (29) we apply Triangle Inequality on (30) and get,

$$\left\|\sum_{k=1}^{m}\mathbb{E}(\mathbf{z}_k\mathbf{z}_k^*)\right\|_{S_\infty} \leq \sum_{k=1}^{m}\|\mathbb{E}(\mathbf{z}_k\mathbf{z}_k^*)\|_{S_\infty} \leq m\cdot\frac{L}{m^2}\left(16\frac{\nu^2 r^2}{n^2}\right) = \frac{L}{m}\left(16\frac{\nu^2 r^2}{n^2}\right) \tag{31}$$

Now, we can approximate this bound in the following way,

Since $L = \frac{n(n-1)}{2}$, so $L \leq \frac{n^2}{2}$ this bound can be written as

$$\left\| \sum_{k=1}^{m} \mathbb{E}(\mathbf{z}_k \mathbf{z}_k^*) \right\|_{S_\infty} \leq \frac{L}{m} \left( 16 \frac{\nu^2 r^2}{n^2} \right) \leq 8 \frac{\nu^2 r^2}{m} \tag{32}$$

Comparing the equations (31) and (32), we can argue that the bound on (31) is larger than that of (32) as $\frac{L}{2}$ is larger than $\nu r$. So

$$\left\| \sum_{k=1}^{m} \mathbb{E}(\mathbf{z}_k \mathbf{z}_k^*) \right\|_{S_\infty} \leq 8 \frac{\nu^2 r^2}{m} \leq 4 \frac{\nu r L}{m} \tag{33}$$

So, now by Matrix Bernstein Inequality as stated Theorem C.1,

$$\mathbb{E}(\mathbf{z}_k) = 0$$

The value of the bound of $\mathbf{z}_k$ is denoted by R in Theorem C.1. So R in this case is

$$\|\mathbf{z}_k\|_{S_\infty} \leq R = \frac{4\nu r L}{mn} + \frac{1}{m}$$

Further $\sigma$ is defined by $\max \mathbb{E}(\mathbf{z}_k \mathbf{z}_k^*), \mathbb{E}(\mathbf{z}_k^* \mathbf{z}_k)]$ as per Theorem C.1. We have,

$$\sigma^2 = \max \left[ \left\| \sum_{k=1}^{m} \mathbb{E}(\mathbf{z}_k \mathbf{z}_k^*) \right\|_{S_\infty}, \left\| \sum_{k=1}^{m} \mathbb{E}(\mathbf{z}_k^* \mathbf{z}_k) \right\|_{S_\infty} \right]$$

$$\sigma^2 = \frac{L}{m} \left( \frac{4\nu r}{n} \right)$$

So as per Theorem C.1,

$$\forall t > 0, P(\| \sum_{k=1}^{L} \mathbf{z}_k \|_{S_\infty} \geq t) \leq 2n \exp(\frac{\frac{-t^2}{2}}{\sigma^2 + \frac{Rt}{3}}) \tag{34}$$

Now we would calculate the bound for the above probability, In this scenario, since $L = \frac{n(n-1)}{2}$, we can bound $(\sigma^2 + \frac{R\epsilon}{3})$ by the following inequality in the second,

$$(\sigma^2 + \frac{R\epsilon}{3}) = \frac{L}{m} \left( \frac{4\nu r}{n} \right) + \left( \frac{4\nu r L}{mn} + \frac{1}{m} \right) \frac{\epsilon}{3}$$

$$= \left( \frac{4\nu r L}{nm} \right) \left( 1 + \frac{\epsilon}{3} \right) + \frac{\epsilon}{3m}$$

$$\leq \left( \frac{8\nu r L}{nm} \right) + \frac{\epsilon}{3m}$$

$$\leq \left( \frac{9\nu r L}{nm} \right)$$

We use $\frac{\epsilon}{3}$ is less than 1 in the first inequality and the the second summand is less than $\frac{\nu r L}{nm}$ in the second inequality.

Here further we used that $\epsilon \leq \frac{1}{2}$ and $\nu \geq 1, r \geq 1$ and bound the probability as follows,

$$2n \exp \left( \frac{\frac{-\epsilon^2}{2}}{\sigma^2 + \frac{R\epsilon}{3}} \right) \leq 2n \exp \left( \frac{\frac{-\epsilon^2}{2}}{\frac{9\nu r L}{nm}} \right)$$

$$= 2n \exp \left( \frac{-\epsilon^2 nm}{18\nu r L} \right)$$

$$\leq 2n \exp \left( \frac{-49\nu r n^2 \log n}{18\nu r L} \right)$$

$$\leq 2n \exp \left( \frac{-98 \log n}{18} \right)$$

So, given that $m \geq \frac{49\nu nr}{\epsilon^2} \log n$ holds true, we can arrive at the first inequality above, further, since $L = \frac{n(n-1)}{2}$ so we can say, $L \leq n^2/2$ hence we get the second inequality. We can bound this further by,

$$2n \exp \left( \frac{\frac{-\epsilon^2}{2}}{\sigma^2 + \frac{R\epsilon}{3}} \right) \leq 2n \exp \left( \frac{-98 \log n}{18} \right)$$

$$\leq 2n \exp \left( -4 \log n \right)$$

$$\leq 2n \exp \left( \log \left( \frac{1}{n} \right)^2 \right)$$

$$= \frac{2}{n^3}$$

Hence we can conclude from here that $\| \sum_{k=1}^{m} \mathbf{z}_k \|_{S_\infty} \leq \epsilon$) holds with a probability of $1 - \frac{2}{n^3}$. Now we can derive the bound $\left\| \mathcal{P}_{T_0} \mathcal{Q}_\Omega^* \mathcal{P}_{T_0} - \mathcal{P}_{T_0} \right\|_{S_\infty} \leq \epsilon$ in the statement of Theorem 4.5 by the following argument,

$$\left\| \sum_{k=1}^{m} \mathbf{z}_k \right\|_{S_\infty} = \left\| \sum_{\ell=1}^{L} \left( \frac{L}{m} \mathcal{P}_{T_0} \mathbf{w}_{\alpha_\ell} \mathbf{v}_{\alpha_\ell}^\top \mathcal{P}_{T_0} - \frac{1}{m} \mathcal{P}_{T_0} \mathbf{W} \mathbf{V}^\top \mathcal{P}_{T_0} \right) \right\|_{S_\infty}$$

$$= \left\| \mathcal{P}_{T_0} \mathcal{R}_\Omega^* \mathcal{P}_{T_0} - \mathcal{P}_{T_0} \mathbf{W} \mathbf{V}^\top \mathcal{P}_{T_0} \right\|_{S_\infty}$$

$$= \left\| \mathcal{P}_{T_0} \mathcal{R}_\Omega^* \mathcal{P}_{T_0} + \mathcal{P}_{T_0} \mathbf{w}_E \mathbf{v}_E^\top \mathcal{P}_{T_0} - (\mathcal{P}_{T_0} \mathbf{W} \mathbf{V}^\top \mathcal{P}_{T_0} + \mathcal{P}_{T_0} \mathbf{w}_E \mathbf{v}_E^\top \mathcal{P}_{T_0}) \right\|_{S_\infty}$$

$$= \left\| \mathcal{P}_{T_0} \mathcal{Q}_\Omega^* \mathcal{P}_{T_0} - \mathcal{P}_{T_0} \right\|_{S_\infty} \leq \epsilon$$

We get the get implication by adjusting the equation with addition and subtraction of $\mathcal{P}_{T_0} \mathbf{w}_E \mathbf{v}_E^\top \mathcal{P}_{T_0}$ because $\mathcal{P}_{T_0} \mathbf{W} \mathbf{V}^\top \mathcal{P}_{T_0} + \mathcal{P}_{T_0} \mathbf{w}_E \mathbf{v}_E^\top \mathcal{P}_{T_0} = \mathcal{P}_{T_0} \tilde{\mathbf{W}} \tilde{\mathbf{V}}^\top \mathcal{P}_{T_0}$ and $\tilde{\mathbf{W}} \tilde{\mathbf{V}}^\top \mathcal{P}_{T_0} = \mathcal{P}_{T_0}$
So, we have proved the assertion of Theorem 4.5 is true with a probability of $1 - \frac{2}{n^3}$. $\qquad\square$

Following Proposition 3.1 in [RXH11], if we change the sampling model to sampling without replacement we should have the same bound with same failure probability.

## D  Proof of Local Quadratic Convergence

In this section we provide the proof of the local convergence theorem as stated in Theorem 4.3.

**Lemma D.1.** *Let* $0 < \epsilon \leq \frac{1}{2}$, *let* $\mathbf{X}^0 \in S_n$ *be a* $\nu$*-incoherent matrix. Let* $\mathcal{P}_{T_0} : S_n \to S_n$ *be the projection operator associated to* $T_0$. *Then assume that the following three conditions hold:*

*(a) For* $\mathcal{Q}_\Omega : S_n \to S_n$ *be defined as in (13) from* $m$ *independent uniformly sampled locations, we have :*

$$\| \mathcal{Q}_\Omega \|_{S_\infty} \leq \left( 20L \sqrt{\frac{\log n}{m}} + 1 \right).$$

*(b) The tangent space* $T_0 = T_{\mathbf{X}^0}$ *onto the rank-$r$ manifold* $T_0 = T_{\mathbf{X}^0} \mathcal{M}_r$ *fulfills :*

$$\left\| \mathcal{P}_{T_0} \mathcal{Q}_\Omega^* \mathcal{P}_{T_0} - \mathcal{P}_{T_0} \right\|_{S_\infty} \leq \varepsilon \tag{35}$$

*(c) The spectral norm distance between* $\mathbf{X}$ *and* $\mathbf{X}^0$ *fulfills:*

$$\left\| \mathbf{X} - \mathbf{X}^0 \right\|_{S_\infty} \leq \frac{\epsilon}{\left( 20L \sqrt{\frac{\log n}{m}} + 1 \right)} \sigma_r(\mathbf{X}^0) \tag{36}$$

*Then the tangent space $T = T_{\mathbf{X}}$ onto the rank-r manifold at $\mathbf{X}$ fulfills:*

$$\left\| \mathcal{P}_T \, \mathcal{Q}_\Omega^* \, \mathcal{P}_T - \mathcal{P}_T \right\|_{S_\infty} \leq 4\varepsilon \tag{37}$$

Lemma D.1 is a literal adaptation of [KV21, Lemma B.3] to the operator $\mathcal{Q}_\Omega^*$, which is why we omit its proof. Similarly, we can use Lemmas B.8 and B.9 of [KV21] in the same way for our proofs.

**Lemma D.2.** *Let $\mathbf{X}^0 \in S_n$ be a matrix of rank $r$ that is $\nu$-incoherent, and let $\Omega = (i_\ell, j_\ell)_{\ell=1}^m$ be a random index set of cardinality $|\Omega| = m$ that is sampled uniformly without replacement, or, alternatively, sampled independently with replacement. There exists constants $C, \widetilde{C}, C_1$ such that if*

$$m \geq C\nu rn \log n \tag{38}$$

*then, with probability at least $1 - \frac{2}{n^2}$, the following holds: For each matrix $\mathbf{X}^{(k)} \in S_n$ fulfilling*

$$\|\mathbf{X}^{(k)} - \mathbf{X}^0\|_{S_\infty} \leq C_1 \frac{\sqrt{m}}{L\sqrt{\log n}} \sigma_r(\mathbf{X}^0), \tag{39}$$

*it follows that the projection $\mathcal{P}_{T_k} : S_n \to S_n$ onto the tangent space $T_k := T_{\mathcal{T}_r(\mathbf{X}^{(k)})}\mathcal{M}_r$ satisfies*

$$\left\| \mathcal{P}_{T_k} \, \mathcal{Q}_\Omega^* \, \mathcal{P}_{T_k} - \mathcal{P}_{T_k} \right\|_{S_\infty} \leq \frac{2}{5},$$

*and furthermore,*

$$\|\eta\|_F \leq \widetilde{C} L \sqrt{\frac{\log n}{m}} \|\mathcal{P}_{T_k^\perp}(\eta)\|_F.$$

*for each matrix $\eta \in \ker \mathcal{Q}_\Omega$ in the null space of the operator $\mathcal{Q}_\Omega : S_n \to S_n$.*

*Proof of Lemma D.2.* Assume that there are $m$ locations $\Omega = (i_\ell, j_\ell)_{\ell=1}^m$ in $n \times n$ sampled independently uniformly *with replacement*, where $m$ fulfills (38) with $C := 49/\varepsilon^2$ and $\varepsilon = 0.1$. By Lemma C.2, it follows that the corresponding operator $\mathcal{Q}_\Omega$ from Lemma C.2 fulfills

$$\| \mathcal{Q}_\Omega \|_{S_\infty} \leq \left( 20L\sqrt{\frac{\log n}{m}} + 1 \right). \tag{40}$$

on an event called $\mathbf{e}_\Omega$, which occurs with a probability of at least $1 - \frac{2}{n}$, and by Theorem 4.5, the tangent space $T_0 = T_{\mathbf{X}^0}\mathcal{M}_r$ corresponding to the $\mu_0$-incoherent rank-$r$ matrix $\mathbf{X}^0$ fulfills

$$\left\| \mathcal{P}_{T_0} \mathcal{Q}_\Omega^* \mathcal{P}_{T_0} - \mathcal{P}_{T_0} \right\|_{S_\infty} \leq \varepsilon$$

on an event called $\mathbf{e}_{\Omega, T_0}$, which occurs with a probability of at least $1 - n^{-2}$. Let $\tilde{\epsilon} = \frac{1}{10}$. If $\mathbf{X}^{(k)} \in \mathbb{R}^{n \times n}$ is such that $\|\mathbf{X}^{(k)} - \mathbf{X}^0\|_{S_\infty} \leq \widetilde{\xi}\sigma_r(\mathbf{X}^0)$ with

$$\widetilde{\xi} = \frac{\tilde{\epsilon}}{\left( 20L\sqrt{\frac{\log n}{m}} + 1 \right)} \leq \frac{1}{\left( 20 \cdot 10L\sqrt{\frac{\log n}{m}} \right)} \tag{41}$$

it follows by Lemma D.1 that on the event $E_\Omega \cap E_{\Omega, T_0}$, the tangent space $T_k := \mathbf{X}^{(k)}$ onto the rank-$r$ manifold at $\mathbf{X}^{(k)}$ fulfills

$$\left\| \mathcal{P}_{T_k} \mathcal{Q}_\Omega^* \mathcal{P}_{T_k} - \mathcal{P}_{T_k} \right\|_{S_\infty} \leq 4\tilde{\epsilon} = \frac{2}{5}. \tag{42}$$

Next, we claim that on the event $E_\Omega \cap E_{\Omega, T_0}$,

$$\|\eta\|_F \leq \widetilde{C} L \sqrt{\frac{\log n}{m}} \|\mathcal{P}_{T_k^\perp}(\eta)\|_F. \tag{43}$$

for any for each matrix $\eta \in \ker \mathcal{Q}_\Omega$ in the null space of the operator $\mathcal{Q}_\Omega : S_n \to S_n$.

Let $\eta \in \ker \mathcal{Q}_\Omega$.

Then

$$\|\mathcal{P}_{T_k}(\eta)\|_F^2 = \langle \mathcal{P}_{T_k}(\eta), \mathcal{P}_{T_k}(\eta) \rangle$$

$$= \left\langle \mathcal{P}_{T_k}(\eta), \mathcal{P}_{T_k} \, \mathcal{Q}_\Omega^* \, \mathcal{P}_{T_k}(\eta) \right\rangle + \left\langle \mathcal{P}_{T_k}(\eta), \mathcal{P}_{T_k}(\eta) - \mathcal{P}_{T_k} \, \mathcal{Q}_\Omega^* \, \mathcal{P}_{T_k}(\eta) \right\rangle$$

$$\leq \left\langle \mathcal{P}_{T_k}(\eta), \mathcal{P}_{T_k} \, \mathcal{Q}_\Omega^* \, \mathcal{P}_{T_k}(\eta) \right\rangle + \|\mathcal{P}_{T_k}(\eta)\|_F \left\| \mathcal{P}_{T_k} - \mathcal{P}_{T_k} \, \mathcal{Q}_\Omega^* \, \mathcal{P}_{T_k} \right\|_{S_\infty} \|\mathcal{P}_{T_k}(\eta)\|_F$$

$$\leq \left\langle \mathcal{P}_{T_k}(\eta), \mathcal{P}_{T_k} \, \mathcal{Q}_\Omega^* \, \mathcal{P}_{T_k}(\eta) \right\rangle + 4\epsilon \, \|\mathcal{P}_{T_k}(\eta)\|_F^2$$

Using (42) in the last inequality, implies that

$$
\begin{aligned}
\|\mathcal{P}_{T_k}(\eta)\|_F^2 &\leq \frac{1}{1-5\epsilon} \langle \mathcal{P}_{T_k}(\eta), \mathcal{P}_{T_k} \, \mathcal{Q}_\Omega^* \, \mathcal{P}_{T_k}(\eta) \rangle \\
&\leq \frac{1}{1-5\epsilon} \langle \mathcal{P}_{T_k}(\eta), \mathcal{P}_{T_k} \, \mathcal{Q}_\Omega^* \, \mathcal{P}_{T_k}(\eta) \rangle \\
&\leq \frac{1}{1-4\epsilon} \langle \mathcal{P}_{T_k}(\eta)^2, \mathcal{Q}_\Omega^* \, \mathcal{P}_{T_k}(\eta) \rangle \\
&\leq \frac{1}{1-4\epsilon} \langle \mathcal{Q}_\Omega \, \mathcal{P}_{T_k}(\eta), \mathcal{P}_{T_k}(\eta)^2) \rangle \\
&\leq \frac{1}{1-4\epsilon} \langle \mathcal{Q}_\Omega \, \mathcal{P}_{T_k}(\eta), \mathcal{P}_{T_k} \mathcal{P}_{T_k}(\eta) \rangle \\
&\leq \frac{1}{1-4\epsilon} \langle \mathcal{P}_{T_k} \, \mathcal{Q}_\Omega \, \mathcal{P}_{T_k}(\eta), \mathcal{P}_{T_k}(\eta) \rangle \\
&\leq \frac{1}{1-4\epsilon} \langle \mathcal{P}_{T_k}(\eta), \mathcal{P}_{T_k} \, \mathcal{Q}_\Omega \, \mathcal{P}_{T_k}(\eta) \rangle \\
&\leq \frac{1}{1-4\epsilon} \|\mathcal{P}_{T_k}(\eta)\|_F \|\mathcal{P}_{T_k}\|_{S_\infty} \| \mathcal{Q}_\Omega \, \mathcal{P}_{T_k}(\eta)\|_F
\end{aligned}
\tag{44}
$$

Dividing by $\|\mathcal{P}_{T_k}(\eta)\|_F$ on both sides we get,

$$
\begin{aligned}
\|\mathcal{P}_{T_k}(\eta)\|_F &\leq \frac{1}{1-4\epsilon} \|\mathcal{P}_{T_k}\|_{S_\infty} \| \mathcal{Q}_\Omega \, \mathcal{P}_{T_k}(\eta)\|_F \\
&\leq 2\| \mathcal{Q}_\Omega \, \mathcal{P}_{T_k}(\eta)\|_F
\end{aligned}
$$

Since spectral norm of $\mathcal{P}_{T_k}$. is bounded by 1. Furthermore, we used that $\epsilon \leq \frac{1}{10}$ in the last inequality. Since $\eta \in \ker \mathcal{Q}_\Omega$, it holds that

$$0 = \| \mathcal{Q}_\Omega(\eta)\|_F = \left\| \mathcal{Q}_\Omega \left( \mathcal{P}_{T_k}(\eta) + \mathcal{P}_{T_k^\perp}(\eta) \right) \right\|_F \geq \| \mathcal{Q}_\Omega \, \mathcal{P}_{T_k}(\eta)\|_F - \| \mathcal{Q}_\Omega \, \mathcal{P}_{T_k^\perp}(\eta)\|_F$$

so that

$$\| \mathcal{Q}_\Omega \, \mathcal{P}_{T_k}(\eta)\|_F \leq \| \mathcal{Q}_\Omega \, \mathcal{P}_{T_k^\perp}(\eta)\|_F \leq \left( 20L\sqrt{\frac{\log n}{m}} + 1 \right) \|\mathcal{P}_{T_k^\perp}(\eta)\|_F,$$

where we used (40) in the last inequality. Inserting this above, we obtain

$$\|\eta\|_F^2 = \|\mathcal{P}_{T_k}(\eta)\|_F^2 + \|\mathcal{P}_{T_k^\perp}(\eta)\|_F^2 \leq \left( 4\left( 20L\sqrt{\frac{\log n}{m}} + 1 \right)^2 + 1 \right) \|\mathcal{P}_{T_k^\perp}(\eta)\|_F^2$$

$$\leq 5\left( 21L\sqrt{\frac{\log n}{m}} \right)^2 \|\mathcal{P}_{T_k^\perp}(\eta)\|_F^2$$

Since $\frac{L}{m} \log n > 1$ So we get

$$\|\eta\|_F \leq \widetilde{C}L\sqrt{\frac{\log n}{m}} \|\mathcal{P}_{T_k^\perp}(\eta)\|_F. \tag{45}$$

for the constant $\widetilde{C}$ defined by,

$$\widetilde{C} := \sqrt{5} \cdot 21$$

Moreover, we observe that for $C_1 := \frac{1}{20\widetilde{C}}$ where $C_1$ is the constant of (39), it holds that

$$\widetilde{\xi} \leq \frac{1}{\left(10 \cdot 20L \frac{\log n}{m}\right)} = \frac{C_1 \sqrt{m}}{L\sqrt{\log n}}$$

implying that the two statements of Lemma D.2 are satisfied on the event $E_\Omega \cap E_{\Omega,T_0}$ if (39) holds. By the above mentioned probability bounds and a union bound, $E_\Omega \cap E_{\Omega,T_0}$ occurs with a probability of at least $1 - 2n^{-2}$, finishing the proof for the sampling with replacement model. By the argument of Proposition 3 of [Rec11], the result extends to the model of sampling locations drawn uniformly at random without replacement, with the same probability bound. This concludes the proof of Lemma D.2. $\qquad\square$

The following lemma will also play a role in the proof of Theorem 4.3.

**Lemma D.3.** *Let $C, \widetilde{C}, C_1$ be the constants of Lemma D.2 and $\mu_0$ be the incoherence factor of a rank-$r$ matrix $\mathbf{X}^0$. If*

$$m \geq C\nu r n \log n$$

*and if $\eta^{(k)} = \mathbf{X}^{(k)} - \mathbf{X}^0$ fulfills*

$$\|\eta^{(k)}\|_{S_\infty} \leq \xi\sigma_r(\mathbf{X}^0),$$

*with*

$$\xi := \min\left(\frac{C_1\sqrt{m}}{L\sqrt{\log n}}, \frac{10C_1\sqrt{m}}{8L\sqrt{r}\kappa\sqrt{\log n}}\right)$$

*then, on the event of Lemma D.2, it holds that*

$$\|\eta^{(k)}\|_{S_\infty} \leq 2\widetilde{C}L\sqrt{\frac{\log n}{m}}(\sqrt{n-r})\sigma_{r+1}(\mathbf{X}^{(k)}) \tag{46}$$

*Proof.* First, we compute that

$$\|\mathcal{P}_{T_k^\perp}(\eta^{(k)})\|_F \leq \|\mathcal{P}_{T_k^\perp}(\mathbf{X}^{(k)})\|_F + \|\mathcal{P}_{T_k^\perp}(\mathbf{X}^0)\|_F$$

$$\leq \sqrt{\sum_{i=r+1}^d \sigma_i^2(\mathbf{X}^{(k)})} + \left\|\mathbf{U}_\perp^{(k)}\mathbf{U}_\perp^{(k)*}\mathbf{X}^0\mathbf{V}_\perp^{(k)}\mathbf{V}_\perp^{(k)*}\right\|_F$$

$$\leq \sqrt{n-r}\sigma_{r+1}(\mathbf{X}^{(k)}) + \|\mathbf{U}_\perp^{(k)*}\mathbf{U}_0\|_{S_\infty}\|\mathbf{\Sigma}_0\|_F\|\mathbf{V}_0^*\mathbf{V}_\perp^{(k)}\|_{S_\infty}$$

$$\leq \sqrt{n-r}\sigma_{r+1}(\mathbf{X}^{(k)}) + \frac{2\|\eta^{(k)}\|_{S_\infty}^2}{(1-\zeta)^2\sigma_r^2(\mathbf{X}^0)}\sqrt{r}\sigma_1(\mathbf{X}^0)$$

$$= \sqrt{n-r}\sigma_{r+1}(\mathbf{X}^{(k)}) + \frac{2\|\eta^{(k)}\|_{S_\infty}^2}{(1-\zeta)^2\sigma_r(\mathbf{X}^0)}\sqrt{r}\kappa,$$

where $0 < \zeta < 1$ such that $\|\mathbf{X}^{(k)} - \mathbf{X}^0\|_{S_\infty} \leq \zeta\sigma_r(\mathbf{X}^0)$, using Lemma D.4 twice in the fourth inequality and $\|\mathbf{AB}\|_F \leq \|\mathbf{A}\|_{S_\infty}\|\mathbf{B}\|_F$ all matrices $\mathbf{A}$ and $\mathbf{B}$, referring to the notations of lemma $B.8$ in [KV21](see below) for $\mathbf{U}_0, \mathbf{\Sigma}_0, \mathbf{V}_0, \mathbf{U}_\perp^{(k)}$ and $\mathbf{V}_\perp^{(k)}$.

Using Lemma D.2 for $\eta^{(k)} = \mathbf{X}^{(k)} - \mathbf{X}^0$, we obtain on the event on which the statement of Lemma D.2 holds that

$$\|\eta^{(k)}\|_{S_\infty} \leq \|\eta^{(k)}\|_F \leq \widetilde{C}L\sqrt{\frac{\log n}{m}}\|\mathcal{P}_{T_k^\perp}(\eta^{(k)})\|_F$$

$$\leq \widetilde{C}L\sqrt{\frac{\log n}{m}}\left(\sqrt{n-r}\sigma_{r+1}(\mathbf{X}^{(k)}) + \frac{8\sqrt{r}\kappa\|\eta^{(k)}\|_{S_\infty}^2}{\sigma_r(\mathbf{X}^0)}\right)$$

$$\leq \widetilde{C}L\sqrt{\frac{\log n}{m}}\left(\sqrt{n-r}\sigma_{r+1}(\mathbf{X}^{(k)}) + \frac{8 \cdot 10C_1\sqrt{m}\sqrt{r}\kappa}{8L\sqrt{\log n}\sqrt{r}\kappa}\|\eta^{(k)}\|_{S_\infty}\right)$$

since $C_1 = \frac{1}{20\widetilde{C}}$, after rearranging we get,

$$\left(1 - \frac{1}{2}\right)\|\eta^{(k)}\|_{S_\infty} \le \widetilde{C}L\sqrt{\frac{\log n}{m}}(\sqrt{n-r})\sigma_{r+1}(\mathbf{X}^{(k)})$$

which implies the statement of this lemma. $\qquad\square$

**Lemma D.4** (Wedin's bound [Ste06]). *Let $\mathbf{X}$ and $\widehat{\mathbf{X}}$ be two matrices of the same size and their singular value decompositions*

$$\mathbf{X} = (\mathbf{U} \quad \mathbf{U}_\perp)\begin{pmatrix}\mathbf{\Sigma} & 0 \\ 0 & \mathbf{\Sigma}_\perp\end{pmatrix}\begin{pmatrix}\mathbf{V}^* \\ \mathbf{V}_\perp^*\end{pmatrix} \quad and \quad \widehat{\mathbf{X}} = (\widehat{\mathbf{U}} \quad \widehat{\mathbf{U}}_\perp)\begin{pmatrix}\widehat{\mathbf{\Sigma}} & 0 \\ 0 & \widehat{\mathbf{\Sigma}}_\perp\end{pmatrix}\begin{pmatrix}\widehat{\mathbf{V}}^* \\ \widehat{\mathbf{V}}_\perp^*\end{pmatrix},$$

*where the submatrices have the sizes of corresponding dimensions. Suppose that $\delta, \alpha$ satisfying $0 < \delta \le \alpha$ are such that $\alpha \le \sigma_{\min}(\Sigma)$ and $\sigma_{\max}(\widehat{\Sigma}_\perp) < \alpha - \delta$. Then*

$$\|\widehat{\mathbf{U}}_\perp^* \mathbf{U}\|_{S_\infty} \le \sqrt{2}\frac{\|\mathbf{X} - \widehat{\mathbf{X}}\|_{S_\infty}}{\delta} \text{ and } \|\widehat{\mathbf{V}}_\perp^* \mathbf{V}\|_{S_\infty} \le \sqrt{2}\frac{\|\mathbf{X} - \widehat{\mathbf{X}}\|_{S_\infty}}{\delta}. \tag{47}$$

Now using the above lemma let us conclude the proof for the theorem,

*Proof of Theorem 4.3.* Let $k = k_0$ and $\mathbf{X}^{(k)}$ be the $k$-th iterate of `MatrixIRLS fro EDG` with the parameters stated in Theorem 4.3. Under the sampling model of Theorem 4.3, if the number of samples $m$ fulfills $m \ge C\nu rn \log n$, where $C$ is the constant of Lemma D.2, we know from Lemma D.2

if furthermore $\eta^{(k)} := \mathbf{X}^{(k)} - \mathbf{X}^0$ fulfills

$$\|\eta^{(k)}\|_{S_\infty} \le \xi\sigma_r(\mathbf{X}^0) \tag{48}$$

with

$$\xi \le \frac{C_1\sqrt{m}}{L\sqrt{\log n}}, \tag{49}$$

holds with a probability of at least $1 - 2n^{-2}$. Then, by lemma $B.9$ of [KV21],

$$\|\mathbf{X}^{(k+1)} - \mathbf{X}^0\|_{S_\infty} \le \left(\frac{\widetilde{C}^2 L \log n}{m}\right)\epsilon_k^2\|W^{(k)}(\mathbf{X}^0)\|_{S_1}. \tag{50}$$

We denote the event that this is fulfilled by $E$. Furthermore, on this event, if $\xi \le 1/2$ in (48) and denoting the condition number by $\kappa = \sigma_1(\mathbf{X}^0)/\sigma_r(\mathbf{X}^0)$, it follows from ,lemma $B.8$ of [KV21], that

$$\|\mathbf{X}^{(k+1)} - \mathbf{X}^0\|_{S_\infty} \le \left(\frac{\widetilde{C}^2 L \log n}{m}\right)4\sigma_r(\mathbf{X}^0)^{-1}\left(\epsilon_k^2 + 4\epsilon_k\|\eta^{(k)}\|_{S_\infty}\kappa + 2\|\eta^{(k)}\|_{S_\infty}^2\kappa\right)$$

Furthermore, if $\mathbf{X}_r^{(k)} \in \mathbb{R}^{n\times n}$ denotes the best rank-$r$ approximation of $\mathbf{X}^{(k)}$ in any unitarily invariant norm, we estimate that

$$\epsilon_k \le \sigma_{r+1}(\mathbf{X}^{(k)}) = \|\mathbf{X}^{(k)} - \mathbf{X}_r^{(k)}\|_{S_\infty} \le \|\mathbf{X}^{(k)} - \mathbf{X}^0\|_{S_\infty} = \|\eta^{(k)}\|_{S_\infty},$$

Inserting these two bounds into (50), we obtain

$$\|\eta^{(k+1)}\|_{S_\infty} = \|\mathbf{X}^{(k+1)} - \mathbf{X}^0\|_{S_\infty} = \left(\frac{\widetilde{C}^2 L \log n}{m}\right)4\sigma_r(\mathbf{X}^0)^{-1}(1 + 6\kappa)\|\eta^{(k)}\|_{S_\infty}^2.$$

Finally, if, additionally, (48) is satisfied for

$$\xi \le \left(\frac{m}{\widetilde{C}^2 L \log n}\right)\frac{1}{4(1 + 6\kappa)}, \tag{51}$$

we conclude that

$$\|\eta^{(k+1)}\|_{S_\infty} < \|\eta^{(k)}\|_{S_\infty}$$

and also, we observe a quadratic decay in the spectral error such that
$$\|\eta^{(k+1)}\|_{S_\infty} \leq \mu\|\eta^{(k)}\|_{S_\infty}^2$$
with a constant $\mu = \left(\frac{m}{\widetilde{C}^2 L \log n}\right)\frac{1}{4(1+6\kappa)\sigma_r(\mathbf{X}^0)}$. This shows condition $b$ of Theorem 4.3.

To show the remaining statement, we can use Lemma D.3 to show that if $\mathbf{X}^{(k)}$ is close enough to $\mathbf{X}^0$, we can ensure that the $(r+1)$-st singular value $\sigma_{r+1}(\mathbf{X}^{(k)})$ of the current iterate is strictly decreasing. More precisely, assume now the stricter assumption of

$$\|\eta^{(k)}\|_{S_\infty} \leq \frac{\sqrt{m}\sqrt{r}}{12\widetilde{C}L(\log n)\sqrt{n-r}}\xi\sigma_r(\mathbf{X}^0) \tag{52}$$

In fact, if $\xi$ fulfills (49) and (51), we can conclude that on the event $E$,

$$\begin{aligned}
\sigma_{r+1}(\mathbf{X}^{(k+1)}) \leq \|\eta^{(k+1)}\|_{S_\infty} &\leq \|\eta^{(k+1)}\|_{S_\infty} \\
&\leq \left(\frac{\widetilde{C}^2 L \log n}{m}\right) 4\sigma_r(\mathbf{X}^0)^{-1}(1+6\kappa)\|\eta^{(k)}\|_{S_\infty}\cdot\|\eta^{(k)}\|_{S_\infty} \\
&< \left(\frac{\widetilde{C}^2 L \log n}{m}\right) 4\sigma_r(\mathbf{X}^0)^{-1}(1+6\kappa)\frac{\sqrt{m}\sqrt{r}}{12\widetilde{C}L(\log n)\sqrt{n-r}} \\
&\quad \cdot \xi\sigma_r(\mathbf{X}^0)2\widetilde{C}L\sqrt{\frac{\log n}{m}}(\sqrt{n-r})\sigma_{r+1}(\mathbf{X}^{(k)}) \\
&\leq \sigma_{r+1}(\mathbf{X}^{(k)})
\end{aligned}$$

using Lemma D.3 for one factor $\|\eta^{(k)}\|_{S_\infty}$ and (52) for the other factor $\|\eta^{(k)}\|_{S_\infty}$ in the third inequality, and (51) in the last inequality. Taking the update rule (7) for the smoothing parameter into account, this implies that $\epsilon_{k+1} = \sigma_{r+1}(\mathbf{X}^{(k+1)})$, which ensures that the first statement of Theorem 4.3 is fulfilled likewise for iteration $k+1$. By induction, this implies that $\mathbf{X}^{(k+\ell)} \xrightarrow{\ell\to\infty} \mathbf{X}^0$, which finishes the proof of Theorem 4.3.

$\square$

# E  Numerical Considerations

## E.1  Experiments on more real data

As a continuation of Section 5, we provide the error analysis for the 1BPM protein data whose corresponding recovery visualizations are found in Figures 7a and 7b. The figure Figure 7a shows the Procrustes distance between the recovered matrix from the samples provided and the ground-truth for both datasets. Similar to the phase transition diagram of the Gaussian data, we observe the probability of success observed over these 24 instances in fig. 7b.

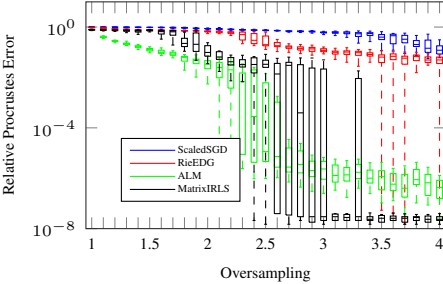
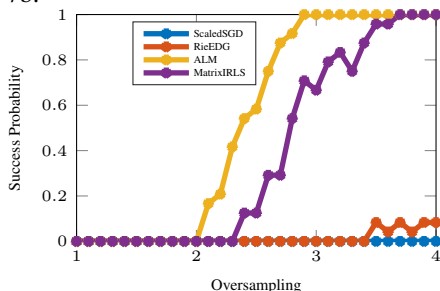

(a) The relative Procrustes error for all the algorithms for different oversampling rates for 1BPM protein data

(b) The probability of success for all the algorithms for different oversampling rates for 1BPM protein data.

Figure 7

Next, we evaluate the performance of MatrixIRLS (Algorithm 1), on the US cities dataset [UU20] in comparison to the aforementioned algorithms. In this setup, we are given $m = |\Omega|$ Euclidean

distances $\sqrt{(\lambda_i - \lambda_j)^2 + (\phi_i - \phi_j)^2}$ between vectors containing longitude and latitude values $\lambda_i, \lambda_j$ and $\phi_i, \phi_j$ of $n = 2920$ cities in the United States, whose squares can be arranged in an incomplete distance matrix $\mathbf{D} \in S_n$ that serves as an input for the reconstruction algorithms. Like the previous setup of Section 5, the set $\Omega \subset \mathbb{I}$ of point index pairs is sampled uniformly at random, and we consider different choices of $m$ parameterized by the oversampling factor $\rho$. Here, for the US cities the rank of the input matrix is 2. In Figure 8a we observe a similar pattern to that of Figure 7a, for the US cities data. The success probability Figure 8b also shows a similar pattern to that Figure 4.

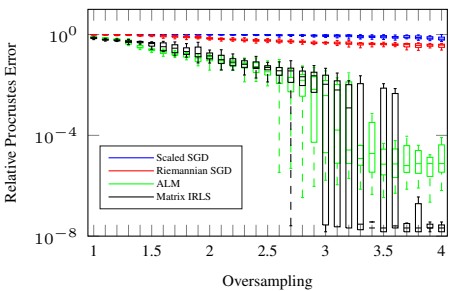
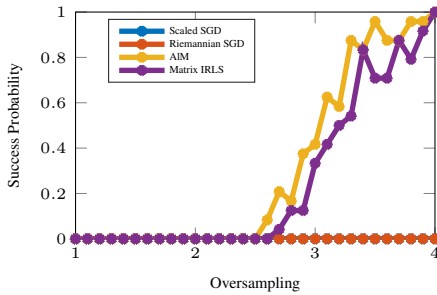

(a) The relative Procrustes error for all the algorithms for different oversampling rates for US cities data

(b) The probability of success for all the algorithms for different oversampling rates for US cities

Figure 8

We visualize the reconstruction of the US cities data as shown in Figure 9.

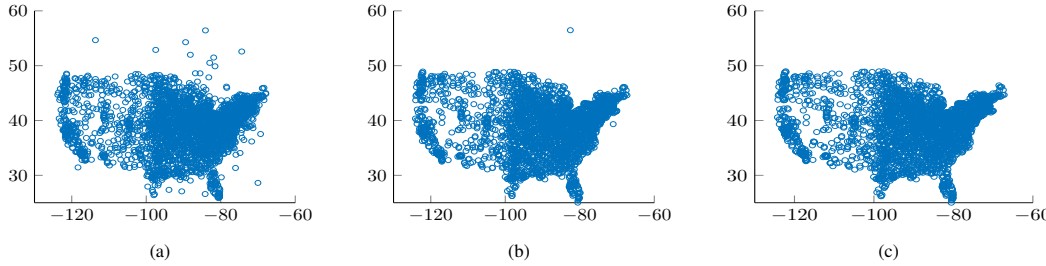

(a)           (b)           (c)

Figure 9: Visualizations of the recovery of US cities data by MatrixIRLS for oversampling rates $1.5, 2.5, 3.5$ in 9a,9b,9c respectively

In this paper, we have adapted the authors' codes of ScaledSGD [ZCZ22], ALM [TL18] and MatrixIRLS [KV21] from their respective githubs. The code of RieEDG [SLCT23][KV21], has been obtained from the author through personal communication.

### E.2 Choice of parameters

- **MatrixIRLS**: The algorithm is configured with the following options:

  The input parameter for MatrixIRLS includes the number of outer IRLS iterations $N^0$, the number of inner iterations $N^0_{inner}$, the tolerance, which is the stopping criteria for the algorithm $tol^0_{inner}$. For large datasets like the UScities data and Protein data $N = 400$, although the algorithm converges within 120 iterations. We run the experiments with $N^0_{inner} = 2000$ and $tol_{inner} = 10^{-10}$. In a smaller setup of the synthetic data we perform the experiments with $n = 500$ points with same parameters. although convergence of MatrixIRLS is observed much faster. To put more emphasis on the per iteration error, we study the experiment on per-iterate analysis with more iterations.

- **Augmented Lagrangian Method**: For the Augmented Lagrangian method(ALM) we have same number iterations which is parametrized as $N^0_{firstorder}$ for the Barzilai-Borwein gradient method in the code. There are 3 stopping criteria for the BB gradient method which are parametrised by $xtol, ftol$ and $gtol$ on the iterate, functional value of the iterate and the gradient of the function respectively. As mentioned in [TL18], the relative change in

*Energy* is the stopping criterion for the algorithm and the tolerance for that measure is set at $10^{-10}$ for our experiments. Similar to the above setup, for larger dataset, more iterations are observed so that we can achieve a fair comparison between all the algorithms.

- **Scaled SGD**: This method uses the learning rate and the number of iterations $N^0_{firstorder}$ as the parameters. For all the experiments the value of $N^0_{firstorder}$ are kept constant across the methods. however, based on the dimension of the input which is $n$, the iterations are modified to achieve a fair comparison. The learning rate is set at $0.2$, which is based on the respective paper [ZCZ22].

- **Riemmanian Gardient**: This method has parameters, number of iterations which is same $N^0_{firstorder}$, the thresholding tolerance which has same value as the $tol_{inner}$.

### E.3 Distance metric

The success of proabbility in the experiments of Section 5, is with respect to the Procrustes distance. The Procrustes analysis uses similarity transformations like scaling, rotation and maps into a common reference frame [ADSVF23]. If we consider our problem, this distance is considered with the ground truth as the reference frame and is defined as

$$\min_{\mathbf{Q} \in \mathbb{R}^{r \times r}, \mathbf{t} \in \mathbb{R}^r} \|\mathbf{Q} \cdot (\mathbf{P_0} + \mathbf{t}\mathbf{1}^\top) - \mathbf{P}_{rec}\|_F \text{ subject to } \mathbf{Q}^\top \mathbf{Q} = \mathbf{I}$$

### E.4 Degrees of freedom

We count the degrees of freedom in the spectral decomposition of a rank-r symmetric matrix: $\mathbf{A} = \mathbf{U}\mathbf{D}\mathbf{U}^\top$ where $\mathbf{U}$ is an orthogonal matrix and $\mathbf{D}$ is a diagonal matrix consisting of the eigenvalues of $\mathbf{A}$ on the diagonal. In $\mathbf{U}$ there are total $r$ unit norm constraints and there are total $\frac{r(r-1)}{2}$ constraints to the orthogonality of the columns vectors of $\mathbf{U}$ which follow from $\langle u_i, u_j \rangle = 0$ for $i \neq j$ and $i \in \{1, 2, ..., r\}$ and $j \in \{1, 2, ..., r\}$. Therefore, the total number of constraints is $\frac{r(r-1)}{2} + r = \frac{r(r+1)}{2}$. The total degrees of freedom in $\mathbf{D}$ is $r$. Hence, the total degrees of freedom of $\mathbf{A}$ is:

$$nr + r - (r + \frac{r(r+1)}{2}) = nr - \frac{r(r-1)}{2}.$$

## F    Computational Complexity

For a symmetric matrix $\mathbf{X} \in S_n$, we write its eigendecomposition (with in magnitude decaying eigenvalues) such that

$$\mathbf{X} = [\mathbf{U} \quad \mathbf{U}_\perp] \begin{bmatrix} \boldsymbol{\Lambda} & 0 \\ 0 & \boldsymbol{\Lambda}_\perp \end{bmatrix} \begin{bmatrix} \mathbf{U}^\top \\ \mathbf{U}_\perp^\top \end{bmatrix}, \tag{53}$$

where $\mathbf{U} \in \mathbb{R}^{n \times r}$, $\mathbf{U}_\perp \in \mathbb{R}^{n \times (n-r)}$ are matrices with orthonormal columns and $\boldsymbol{\Lambda} = \text{diag}(\gamma\sigma_1, \ldots \gamma\sigma_r)$ and $\boldsymbol{\Lambda}_\perp := \text{dg}(\gamma\sigma_{r+1}, \ldots \gamma\sigma_d)$ diagonal matrices with entries $\lambda_i = \sigma_i\gamma_i$ where $\sigma_i$ is the $i$-th singular value of $\mathbf{X}$ and $\gamma_i \in \{\pm 1\}$ a sign. Furthermore, we denote by $\mathcal{T}_r(\mathbf{X})$ the *best rank-r approximation* of $\mathbf{X}$, which can be written such that

$$\mathcal{T}_r(\mathbf{X}) := \underset{\mathbf{Z}:\text{rank}(\mathbf{Z}) \leq r}{\arg\min} \|\mathbf{Z} - \mathbf{X}\| = \mathbf{U}\boldsymbol{\Lambda}\mathbf{U}^\top, \tag{54}$$

where $\|\cdot\|$ can be any unitarily invariant norm, due to the Eckardt-Young-Mirsky theorem [Mir60], using the established notation.

The computational complexity of Algorithm 1 is dominated by the solution of the weighted least squares problem (5) and the computational of spectral information used in the update of the smoothing parameter $\epsilon_k$ of (7) and the weight operator update (7) (see also Definition 3.1).

For solving (5), we consider detailed in Appendix F.1, based on solving a linear system with the dimensionality of tangent space $T_{\mathcal{T}_{r_k}(\mathbf{X}^{(k)})}\mathcal{M}_{r_k}$. The tangent space formulation can be derived from (6) via the Sherman-Morrison-Woodbury [Woo50] formula using the weight operator structure.

In order to update the smoothing parameter and the weight operator we compute the SVD of the iterate by approximating top singular vectors and corresponding values following Randomized Block Krylov Methods [MM15]. This computation takes $O(((m + nr)r \log n)$.

### F.1 Tangent Space Implementation

Let $\mathcal{M}_r := \{\mathbf{X} \in S_n : \text{rank}(\mathbf{X}) = r\}$ the manifold of symmetric rank-$r$ matrices and let $\mathbf{X} \in S_n$ be as in (53). In this case, given $r \in \mathbb{N}$, we can write the tangent space of $\mathcal{M}_r$ at $\mathcal{T}_r(\mathbf{X})$ as

$$
\begin{aligned}
T := T_{\mathcal{T}_r(\mathbf{X})}\mathcal{M}_r &:= \left\{ [\mathbf{U}\,\mathbf{U}_\perp] \begin{bmatrix} \mathbb{R}^{r\times r} & \mathbb{R}^{r\times(n-r)} \\ \mathbb{R}^{(n-r)\times r} & \mathbf{0} \end{bmatrix} [\mathbf{U}\,\mathbf{U}_\perp]^* \right\} \\
&= \left\{ [\mathbf{U}\mathbf{U}_\perp] \begin{bmatrix} \mathbf{M}_1 & \mathbf{M}_2^\top \\ \mathbf{M}_2 & \mathbf{0} \end{bmatrix} [\mathbf{U}\mathbf{U}_\perp]^* : \mathbf{M}_1 \in S_r, \mathbf{M}_2 \in \mathbb{R}^{r\times(n-r)},\ \text{arbitrary} \right\} \\
&= \left\{ \mathbf{U}\Gamma_1\mathbf{U}^\top + \mathbf{U}\Gamma_2^\top\left(\mathbf{I} - \mathbf{U}\mathbf{U}^\top\right) + (\mathbf{I} - \mathbf{U}\mathbf{U}^\top)\Gamma_2\mathbf{U}^\top : \Gamma_1 \in S_r, \Gamma_2 \in \mathbb{R}^{n\times r} \right\} \\
&= \left\{ \mathbf{U}\Gamma_1\mathbf{U}^\top + \mathbf{U}\Gamma_2^\top + \Gamma_2\mathbf{U}^\top : \Gamma_1 \in S_r, \Gamma_2 \in \mathbb{R}^{n\times r}, \mathbf{U}^\top\Gamma_2 = \mathbf{0} \right\},
\end{aligned}
\tag{55}
$$

see also [Van13], [Bou20, Chapter 7.5].

If $\mathcal{P}_T : S_n \to S_n$ is the orthogonal projection operator that projects symmetric matrices onto $T$ as used in Theorem 4.5, we note that

$$
\mathcal{P}_T(\mathbf{X}) = \mathbf{U}\mathbf{U}^\top\mathbf{X}\mathbf{U}\mathbf{U}^\top + \mathbf{U}\mathbf{U}^\top\mathbf{X}(\mathbf{I} - \mathbf{U}\mathbf{U}^\top) + (\mathbf{I} - \mathbf{U}\mathbf{U}^\top)\mathbf{X}\mathbf{U}\mathbf{U}^\top,
$$

which can be decomposed such that

$$
\mathcal{P}_T(\mathbf{X}) = P_T P_T^*(\mathbf{X}),
$$

where the action of $P_T : \mathbb{R}^{r(n+r)} \to S_n$ can be described as

$$
P_T(\gamma) := \mathbf{U}\Gamma_1\mathbf{U}^\top + \mathbf{U}\Gamma_2^\top + \Gamma_2\mathbf{U}^\top
$$

with $\Gamma_1 \in S_r$ being the result of an $(r^2 \times 1)$ to $(r \times r)$ reshaping of the first $r^2$ coordinates of $\gamma$ and $\Gamma_2 \in \mathbb{R}^{n\times r}$ the result of an $(rn \times 1)$ to $(n \times r)$ reshaping of the remaining coordinates of $\gamma$. Further, we used the notation of the adjoint operator $P_T^* : S_n \to \mathbb{R}^{r(n+r)}$ of $P_T$ which maps $\mathbf{X} \in S_n$ to the vectorization $\gamma$ of $[\Gamma_1, \Gamma_2] := \left\{ \mathbf{U}^\top\mathbf{X}\mathbf{U}, (\mathbf{I} - \mathbf{U}\mathbf{U}^\top)\mathbf{X}\mathbf{U} \right\}$.

Due to the fact that the weight operator $W_k = W_{\mathbf{X}^{(k)}, \epsilon_k}$ of Definition 3.1 associated to the iterate $\mathbf{X}^{(k)} \in S_n$ and the smoothing parameter $\epsilon_k > 0$ is self-adjoint and invertible, we can write its inverse as

$$
W_k^{-1} = P_{T_k}\mathbf{D}_k^{-1}P_{T_k}^* + \epsilon_k^2(\mathbf{I} - P_{T_k}P_{T_k}^*),
\tag{56}
$$

where $\mathbf{D}_k^{-1} \in \mathbb{R}^{r_k(n+r_k)\times r_k(n+r_k)}$ is a diagonal matrix with entries $\max(\sigma_i(\mathbf{X}^{(k)}), \epsilon_k)\max(\sigma_j(\mathbf{X}^{(k)}), \epsilon_k)$, where either $i$ or $j$ are smaller or equal than $r_k$, see also [KV21, Eq. (12)], and $T_k = T_{\mathcal{T}_{r_k}(\mathbf{X}^{(k)})}\mathcal{M}_{r_k}$ is the tangent space at $\mathcal{T}_{r_k}(\mathbf{X}^{(k)})$ onto the rank-$r_k$ manifold.

Recall from (7) that the the defining equation for the next iterate $\mathbf{X}^{(k+1)}$ given $W_k$ is

$$
\mathbf{X}^{(k+1)} = W_k^{-1}\mathcal{A}^*\left(\mathcal{A}W_k^{-1}\mathcal{A}^*\right)^{-1}\mathbf{y}.
\tag{57}
$$

Using (56), we see that

$$
\begin{aligned}
\left(\mathcal{A}W_k^{-1}\mathcal{A}^*\right) &= \mathcal{A}\left(P_{T_k}\mathbf{D}_{S_k}^{-1}P_{T_k}^* + \epsilon_k^2\left(\mathbf{I} - P_{T_k}P_{T_k}^*\right)\right)\mathcal{A}^* \\
&= \mathcal{A}\left(P_{T_k}\left(\mathbf{D}_{S_k}^{-1} - \epsilon_k^2\mathbf{I}_{S_k}\right)P_{T_k}^*\right)\mathcal{A}^* + \epsilon_k^2\mathcal{A}\mathcal{A}^*
\end{aligned}
\tag{58}
$$

and, using the Sherman-Morrison-Woodbury [Woo50] formula, we have that

$$
\begin{aligned}
\left(\mathcal{A}(W^{(k)})^{-1}\mathcal{A}^*\right)^{-1} &= \epsilon_k^{-2}(\mathcal{A}\mathcal{A}^*)^{-1} \\
&\quad - \epsilon_k^{-2}(\mathcal{A}\mathcal{A}^*)^{-1}\mathcal{A}P_{T_k}\left(\epsilon_k^2\mathbf{C}^{-1} + P_{T_k}^*\mathcal{A}^*(\mathcal{A}\mathcal{A}^*)^{-1}\mathcal{A}P_{T_k}\right)^{-1}P_{T_k}^*\mathcal{A}^*(\mathcal{A}\mathcal{A}^*)^{-1} \\
&= \epsilon_k^{-2}(\mathcal{A}\mathcal{A}^*)^{-1} - \epsilon_k^{-2}(\mathcal{A}\mathcal{A}^*)^{-1}\mathcal{A}P_{T_k}\mathbf{M}^{-1}P_{T_k}^*\mathcal{A}^*(\mathcal{A}\mathcal{A}^*)^{-1}
\end{aligned}
\tag{59}
$$

with linear system matrix $\mathbf{M} := \left(\epsilon_k^2\mathbf{C}^{-1} + P_{T_k}^*\mathcal{A}^*(\mathcal{A}\mathcal{A}^*)^{-1}\mathcal{A}P_{T_k}\right)$, noting that $\mathbf{C} := \left(\mathbf{D}_k^{-1} - \epsilon_k^2\mathbf{I}\right)$ is invertible since $(\mathbf{D}_k^{-1})_{ii} > \epsilon_k^2$ for all diagonal indices. The observation of (59) can be turned an efficient implementation of the weighted least squares computing $\mathbf{X}^{(k+1)}$, which is presented in Algorithm 2. It can be shown that Algorithm 2 indeed computes $\mathbf{X}^{(k+1)}$ implicitly, cf. Lemma F.1. We omit its proof, which follows the proof of [KV21, Lemma A.1].

---

**Algorithm 2** Tangent space implementation of weighted least squares step in Algorithm 1

---

**Input:** Set $\Omega$, observation distances $\mathbf{D}_\Omega = (d_{ij})_{(i,j)\in\Omega}$, singular vectors $\mathbf{U} \in \mathbb{R}^{d_1\times r_k}$, $\mathbf{V}^{(k)} \in \mathbb{R}^{n\times r_k}$ and singular values $\sigma_1^{(k)}, \ldots, \sigma_{r_k}^{(k)}$, smoothing parameter $\epsilon_k$, projection $\gamma_k^{(0)} = P_{T_k}^* P_{T_{k-1}}(\gamma_{k-1}) \in \mathbb{R}^{r_k(n+r_k)}$ of solution $\gamma_{k-1} \in \mathbb{R}^{r_{k-1}(n+r_{k-1})}$ of linear system (60) for previous iteration $k-1$.

1: Compute $\mathbf{h}_k^0 := P_{T_k}^* \mathcal{A}^* (\mathcal{A}\mathcal{A}^*)^{-1} \mathbf{y} - \left( \epsilon_k^2 \left( \mathbf{D}_k^{-1} - \epsilon_k^2\mathbf{I} \right)^{-1} + P_{T_k}^* \mathcal{A}^* (\mathcal{A}\mathcal{A}^*)^{-1} \mathcal{A}P_{T_k} \right) \gamma_k^{(0)} \in \mathbb{R}^{r_k(n+r_k)}$.

2: Solve
$$\mathbf{M}\Delta\gamma_k = \mathbf{h}_k^0 \tag{60}$$
for $\Delta\gamma_k \in S_k$ by the *conjugate gradient* method [HS52, Meu06], where
$$\mathbf{M} = \epsilon_k^2 \left( \mathbf{D}_k^{-1} - \epsilon_k^2\mathbf{I} \right)^{-1} + P_{T_k}^* \mathcal{A}^* (\mathcal{A}\mathcal{A}^*)^{-1} \mathcal{A}P_{T_k}.$$

3: Compute $\gamma_k = \gamma_k^{(0)} + \Delta\gamma_k$.

4: Compute residual $\mathbf{r}_{k+1} := (\mathcal{A}\mathcal{A}^*)^{-1} (\mathbf{y} - \mathcal{A}P_{T_k}(\gamma_k)) \in \mathbb{R}^m$.

**Output:** $\mathbf{r}_{k+1} \in \mathbb{R}^m$ and $\gamma_k \in \mathbb{R}^{r_k(n+r_k)}$.

---

**Lemma F.1.** *If $\mathbf{r}_{k+1} \in \mathbb{R}^m$ and $\gamma_k \in \mathbb{R}^{r_k(n+r_k)}$ is the output of Algorithm 2, then $\mathbf{X}^{(k+1)}$ as in step* (6) *of Algorithm 1 fulfills*
$$\mathbf{X}^{(k+1)} = \mathcal{A}^*(\mathbf{r}_{k+1}) + \mathcal{P}_{T_k}(\gamma_k)$$
$$= \mathcal{A}^*(\mathbf{r}_{k+1}) + \mathbf{U}\Gamma_1\mathbf{U}^\top + \mathbf{U}\Gamma_2^\top + \Gamma_2\mathbf{U}^\top,$$
*where $\Gamma_1 \in \mathbb{R}^{r_k\times r_k}$ is the matricization of the first $r_k^2$ elements of $\gamma_k$ and $\Gamma_2 \in \mathbb{R}^{n\times r_k}$ the matricization of the reamining elements.*

---

**Algorithm 3** Implementation of $\mathbf{U}^\top\mathcal{A}^* : \mathbb{R}^{m+n} \to \mathbb{R}^{r\times n}$

---

**Input:** Input vector $\mathbf{y} = [y_1, y_2, \ldots, y_{m+n}]$, $\Omega = (i_\ell, j_\ell)_{\ell=1}^m \subset \mathbf{I}$ be a multiset of double indices.

$\mathbf{S_1} = \sum_{\ell=1:(i_\ell,j)\in\Omega \text{ for some } j}^m \mathbf{y}_\ell \mathbf{e}_{i_\ell} \mathbf{e}_{i_\ell}^T$

$\mathbf{S_2} = \sum_{\ell=1:(i,j_\ell)\in\Omega \text{ for some } i}^m \mathbf{y}_\ell \mathbf{e}_{j_\ell} \mathbf{e}_{j_\ell}^T$

$\mathbf{S_3} = \sum_{\ell=1:(i_\ell,j_\ell)\in\Omega}^m \mathbf{y}_\ell \mathbf{e}_{i_\ell} \mathbf{e}_{j_\ell}^T$

$\mathbf{S} = \mathbf{S}_1 + \mathbf{S}_2 - \mathbf{S}_3 - \mathbf{S}_3^T$.      saved as a sparse matrix.

$\mathbf{V}_1 = \mathbf{U}^\top\mathbf{S}$      ▷ $4rm$ flops.

$\mathbf{V}_2 = \frac{1}{2}\mathbf{U}^\top \cdot [y_{m+1}, \ldots, y_{m+n}]^\top \cdot \mathbf{1}^\top$      ▷ $rn$ flops.

$\mathbf{V}_3 = \frac{1}{2}\mathbf{U}^\top \cdot \mathbf{1} \cdot [y_{m+1}, \ldots, y_{m+n}]$      ▷ $2rn$ flops

$\mathbf{V} = \mathbf{V}_1 + \mathbf{V}_2 + \mathbf{V}_3$

**Output:** $\mathbf{V}$      Total of $rm + 3rn$ flops.

---

**Algorithm 4** Implementation of $\mathcal{A}\mathcal{A}^* : \mathbb{R}^{m+n} \to \mathbb{R}^{m+n}$

---

**Input:** $\Omega = (i_\ell, j_\ell)_{\ell=1}^m \subset \mathbf{I}$ be a multiset of double indices fulfilling $m < L = n(n-1)/2$ that are sampled independently with replacement, the vector $\mathbf{y} = [y_1, y_2, \ldots, y_{m+n}]$.

Load $\mathbf{S}$ from Algorithm 3 which computes $\mathcal{A}^*(\mathbf{y})$.

$\mathbf{T_1} = (\mathbf{e}_{i_\ell}^\top \mathbf{S} \mathbf{e}_{i_\ell})_{\ell=1:(i_\ell,j)\in\Omega \text{ for some } j}^m$

$\mathbf{T_2} = (\mathbf{e}_{j_\ell}^\top \mathbf{S} \mathbf{e}_{j_\ell})_{\ell=1:(i,j_\ell)\in\Omega \text{ for some } i}^m \cdot$

$\mathbf{T_3} = (\mathbf{e}_{i_\ell}^\top \mathbf{S} \mathbf{e}_{j_\ell})_{\ell=1:(i_\ell,j_\ell)\in\Omega}^m \cdot$

$\mathbf{T_5} = \mathbf{T}_1 + \mathbf{T}_2 - \mathbf{T}_3 - \mathbf{T}_3^\top$.      ▷ $4m + 2n$ flops

$\text{avg} = \frac{1}{n}\sum_{i=m+1}^{m+n}\mathbf{y}_i \cdot$      ▷ $n$ flops

$\mathbf{T_6} = [y_{m+1} + \text{avg}, \ldots, y_{m+n} + \text{avg}]$      ▷ $n$ flops

**Output:** $[\mathbf{T}_5; \mathbf{T}_6]$      Total of $4m + 4n$ flops

---

---

**Algorithm 5** Implementation of $P_T^* \mathcal{A}^* : \mathbb{R}^{m+n} \to \mathbb{R}^{r(n+r)}$

---

**Input:** Input vector $\mathbf{y} = [y_1, y_2, \ldots, y_{m+n}] \in \mathbb{R}^{m+n}$, index set $\Omega$, left singular vector matrix $\mathbf{U} \in \mathbb{R}^{n \times r}$.

$\mathbf{A}_1 = \mathbf{V} \in \mathbb{R}^{r \times n}$ from Algorithm 3. $\hfill \triangleright O(r(m+n))$ flops

$\Gamma_1 = \mathbf{A}_1 \mathbf{U} \in \mathbb{R}^{r \times r}$. $\hfill \triangleright O(r^2 n)$ flops

$\mathbf{M} = \Gamma_1 \mathbf{U}^\top \in \mathbb{R}^{r \times n}$. $\hfill \triangleright O(r^2 n)$ flops

$\Gamma_2 = (\mathbf{A}_1^\top - \mathbf{M}^\top)$

**Output:** $\{\Gamma_1, \Gamma_2\}$.

---

Lemma F.1 shows that an iterate $\mathbf{X}^{(k+1)}$ can be represented via only $m + r_k(n + r_k)$ parameters. In the remainder of this discussion, we will assume that $r_k = r$, which is the case in most cases in practice if the rank estimate $\widetilde{r}$ of Algorithm 1 is chosen as $\widetilde{r} = r$.

To quantify the computational cost of Algorithm 2, we assume a fixed number $\mathrm{N}_{inner}^0$ of CG iterations solving (60). When applying the system matrix $\mathbf{M}$ via matrix-vector multiplication, we observe that its first summand $\epsilon_k^2 \left( \mathbf{D}_k^{-1} - \epsilon_k^2 \mathbf{I} \right)^{-1}$ is diagonal and thus results in $O(r(n+r))$ flops per CG iteration. To quantify the matrix-vector multiplication cost of its second summand $P_{T_k}^* \mathcal{A}^* (\mathcal{A}\mathcal{A}^*)^{-1} \mathcal{A} P_{T_k}$, we define below algorithms that efficiently implement the application of the operators $P_T^* \mathcal{A}^* : \mathbb{R}^{m+n} \to \mathbb{R}^{r(n+r)}$ (Algorithm 5), $\mathcal{A}\mathcal{A}^* : \mathbb{R}^{m+n} \to \mathbb{R}^{m+n}$ (Algorithm 4) and $\mathcal{A}P_T : \mathbb{R}^{r(n+r)} \to \mathbb{R}^{m+n}$ (Algorithm 6). As an auxiliary function, we also provide an implementation of $\mathbf{U}^\top \mathcal{A}^* : \mathbb{R}^{m+n} \to \mathbb{R}^{r \times n}$ (Algorithm 3) below.

The application of $(\mathcal{A}\mathcal{A}^*)^{-1} : \mathbb{R}^{m+n} \to \mathbb{R}^{m+n}$ can be achieved by an inexact iterative solver that applies Algorithm 4 a fixed number $\tilde{N}$ of iterations, which costs $O((m+n)\tilde{N})$. Since the time complexity of Algorithm 5 and Algorithm 6 are $O(r^2 n + rm)$, we obtain one matrix vector multiplication with $\mathbf{M}$ from Algorithm 2 in a time complexity of $O(r^2 n + rm + (m+n)\tilde{N})$.

Since $\mathrm{N}_{inner}^0$ CG iterations are used, we obtain a total time complexity of $O(\mathrm{N}_{inner}^0(r^2 n + rm + (m+n)\tilde{N}))$ for Algorithm 2. In our experiments, we observe that a small number $\tilde{N} = 10$ of iterations for solving the system associated to $(\mathcal{A}\mathcal{A}^*)^{-1}$ is sufficient to obtain high-accuracy solutions.

This breakdown of the computational costs of each algorithm is shown in Algorithm 3 to Algorithm 6.

---

**Algorithm 6** Implementation of $\mathcal{A}P_T : \mathbb{R}^{r(n+r)} \to \mathbb{R}^{m+n}$

---

**Input:** $\{\Gamma_1, \Gamma_2\}$, index set $\Omega$, left singular vectors $\mathbf{U} \in \mathbb{R}^{n \times r}$.

$\mathbf{M} = \Gamma_1 \mathbf{U}^\top \in \mathbb{R}^{r \times n}$. $\hfill \triangleright O(r^2 n)$ flops from Algorithm 5

$\mathbf{z} = (\sum_{k=1}^r \mathbf{U}_{(j_\ell, k)} \mathbf{M}_{(k, j_\ell)})_{(i, j_\ell) \in \Omega \text{ for some } i}$ $\hfill \triangleright O(mr)$ flops

$\mathbf{z} = \mathbf{z} + (\sum_{k=1}^r \mathbf{U}_{(i_\ell, k)} \mathbf{M}_{(k, i_\ell)})_{(i_\ell, j) \in \Omega \text{ for some } j}$ $\hfill \triangleright O(mr)$ flops

$\mathbf{z} = \mathbf{z} - (\sum_{k=1}^r \mathbf{U}_{(j_\ell, k)} \mathbf{M}_{(k, i_\ell)})_{(i_\ell, j_\ell) \in \Omega}$ $\hfill \triangleright O(mr)$ flops

$\mathbf{z} = \mathbf{z} - (\sum_{k=1}^r \mathbf{U}_{(i_\ell, k)} (\mathbf{M}_{(k, j_\ell)}))_{(i_\ell, j_\ell) \in \Omega}$ $\hfill \triangleright O(mr)$ flops

$\alpha = (\sum_{k=1}^r \mathbf{U}_{(i_\ell, k)} (\Gamma_2^\top)_{(k, i_\ell)})_{(i_\ell, j) \in \Omega \text{ for some } j}$ $\hfill \triangleright O(mr)$ flops

$\alpha = \alpha + (\sum_{k=1}^r \mathbf{U}_{(j_\ell, k)} (\Gamma_2^\top)_{(k, j_\ell)})_{(i, j_\ell) \in \Omega \text{ for some } i}$ $\hfill \triangleright O(mr)$ flops

$\alpha = \alpha - (\sum_{k=1}^r \mathbf{U}_{(j_\ell, k)} (\Gamma_2^\top)_{(k, i_\ell)})_{(i_\ell, j_\ell) \in \Omega}$ $\hfill \triangleright O(mr)$ flops

$\alpha = \alpha - (\sum_{k=1}^r (\mathbf{U})_{(i_\ell, k)} (\Gamma_2^\top)_{k, j_\ell})_{(i_\ell, j_\ell) \in \Omega}$ $\hfill \triangleright O(mr)$ flops

$\zeta_1 = \mathbf{z} + 2\alpha$.

$\mathbf{c}_1 = (\sum_{i=1}^n \mathbf{U}_{(i, k)})_{k=1}^r$ $\hfill \triangleright O(nr)$ flops

$\gamma_1 = \mathbf{c}_1 \mathbf{M}$ $\hfill \triangleright O(mr)$ flops

$\gamma_2 = \mathbf{c}_1 \Gamma_2^\top$ $\hfill \triangleright O(mr)$ flops

$\mathbf{c}_2 = (\sum_{i=1}^n (\Gamma_2)_{(i_\ell, k)})_{k=1}^r$ $\hfill \triangleright O(nr)$ flops

$\gamma_3 = \mathbf{c}_2 \mathbf{U}^\top$ $\hfill \triangleright O(mr)$ flops

$\zeta_2 = \gamma_1 + \gamma_2 + \gamma_3$

**Output:** $[\zeta_1; \zeta_2]$

---

