# OpenReview forum: "Sample-Efficient Geometry Reconstruction from Euclidean Distances using Non-Convex Optimization"
_NeurIPS.cc/2024/Conference — NeurIPS 2024 poster_

### Official Review · Reviewer_fWZw · 2024-07-06

**Soundness:** 4
**Presentation:** 3
**Contribution:** 4
**Rating:** 6
**Confidence:** 3

**Summary:**

This paper proposed a method called MatrixIRLS to solve the EDG problem, an important problem in various fields, especially when pairwise distance is partially observed. The proposed method improves the number of random sampled distances to guarantee local convergence to the ground truth.

**Strengths:**

The related work is discussed comprehensively. The method and theory seem solid. Experiments are convincing.

**Weaknesses:**

1.	There are quite a few typos or undefined notation that make some of the paragraphs hard to read. For example:

  a.	Line 66, $\nu$ is not defined. Although it’s briefly mentioned in line 80, its rigorous definition is on Page 5. I suggest the following change to line 66: $O()$ (where $\nu$ is the coherence factor, see \cref{def:4.1} for details).

  b.	Line 158, $\sigma_i(\cdot)$ is not defined, tho I can guess the meaning.

  c.	Line 165: $X^{(k)}$ should be $X$.

  d.	Eq (5) gives the update from k to k+1, while Eq (6) gives the rule from k-1 to k.

  e.	Eq(9), at the end, there is a missing comma.

2.	Figure 2 looks great, but it will be even better if we can observe the claimed rate $O(\nu rn\log n)$ empirically.

**Questions:**

1.	There are some vertical space issues, like Line 286 for Section 5.
2.	Can the authors briefly discuss the main challenges for global convergence?

**Limitations:**

Discussed in the supplementary material. I suspect the file in the supplementary zip file should be the main submitted file.

---

> ### Author Rebuttal · Authors · 2024-08-07
>
> Thank you for your careful feedback.
>
> We appreciate your pointers regarding typos and your presentation suggestions. We will refer to the definition of the coherence $\nu$ at the first occurrence in the paper.
>
> > Figure 2 looks great, but it will be even better if we can observe the claimed rate $O(\nu r n \log(n))$  empirically.
>
> Thank you for the feedback. We would like to mention that even for related, simpler problems such as the low-rank matrix completion problem, there is little empirical exploration of a related sample complexity phase transition available for any algorithm (only for the more generic Gaussian measurements, such an analysis was done in [Donoho, Gavish, Montanari, PNAS 2013]. The sample complexity order of $O(\nu r n \log(n))$ should be seen as a sharp worst-case bound, but both in practice as well as for our data generation protocols (Gaussian or ill-conditioned data), it is unclear if this dependence on factors $\log(n)$ or coherence parameter $\nu$ is visible in reconstruction experiments. Since we defined the oversampling factor $\rho = m / (n r - r*(r-1)/2)$ as quotient between sample number $m$ and the degrees of freedom of a rank-r p.s.d. matrix, a success rate of $100%$ for $n=500$, $r=5$ at $\rho = 2$ for MatrixIRLS (see Figure 1) shows that recovery is well possible at $m = 4980 < 15537 \approx n r \log(n) \leq \nu n r \log(n)$ (it always holds that $\nu \geq 1$) in many cases.
>
> > Can the authors briefly discuss the main challenges for global convergence?
>
> A major challenge for the global convergence theory of the proposed method is that main theorem, Thm. 4.3, applies only if the method is initialized close enough to the ground truth. The theory of similar IRLS algorithms for other problems (see papers [DDFG10,MF10,KV21]) shares this limitation, and therefore is not unique to our method. The challenge for applying classical optimization theory tools to the proposed scheme is rooted in the non-convexity of the (smoothed) log-det objective as well as in the non-constant smoothing: While MatrixIRLS can be considered as a descent method due to a majorization property of the quadratic function implicitly defined by the weight operator with respect to the $\epsilon$-smoothed log-det objective, the $\epsilon$-smoothing _changes_ at each iteration (cf. Algorithm 1), which slightly changes the underlying functional to be minimized at each iteration. Obtaining a global convergence statement for a scheme like Algorithm 1 would therefore constitute solving a major open problem in the theory of "non-convex" IRLS algorithms.
>
>
> If we were able to adequately address your concerns, we kindly would ask you to consider adjusting your rating accordingly. Thank you!

---

> > ### Comment · Reviewer_fWZw · 2024-08-11
> >
> > Thanks for the response.
> >
> > Your explanation to the rate makes sense to me. It will be very helpful if the authors can discuss this in section 5, to call back the claimed rate in Section 1.1
> >
> > Similarly, discussing the challenges for global convergence will make the story more complete.
> >
> > I raised the score to 6, mainly because the rate is explained clearly to me, which is part of the main contribution claimed in Section 1.1.

---

### Official Review · Reviewer_Jqjh · 2024-07-12

**Soundness:** 2
**Presentation:** 2
**Contribution:** 2
**Rating:** 4
**Confidence:** 4

**Summary:**

The paper studies the under-constrained problem to reconstruct n ordered points in Euclidean space R^r, which are given by fewer than all n(n-1)/2 pairwise distances.

**Strengths:**

The authors should be highly praised for clearly stating main Problem 1 on page 1 and for formulating main Theorems 4.2 and 4.5 almost rigorously.

Problem 1 is well-motivated by the AlphaFold approach to protein folding prediction when a protein backbone in 3D is reconstructed form partially known distances between alpha-carbon atoms.

**Weaknesses:**

Different from the initial Problem 1, the actual considered problem is the rank minimization in (1) after line 54. Does the paper explain anywhere why it makes sense to minimize this integer rank of a Gram matrix even if we forget about initial points. Another weakness is the practical impossibility of solving this rank minimization, which is claimed in line 54 to be NP-hard. It would be useful to re-phrase a problem in another way that allows a polynomial solution.

More importantly, the rank minimization in (1) and algorithm MatrixILRS appeared under the same name but with slightly different notations in the earlier paper [KV21]. So the problem and algorithm in the submitted paper look incremental updates of the past work, which even uses the same names, includes similar theorems, example structures and pseudo-codes.

**Questions:**

For a simple example, if n=4 points in the plane are given by 4 distances that are edge lengths of a quadrilateral on these 4 points, then what's the meaning of this rank minimization in this case? Can Problem 1 be explicitly solved in some partial cases, e.g. for 5 distances on 4 points in the plane?

In Def 3.2, is w_alpha a matrix n x n? In lines 177-178, how is measurement operator <w_alpha_l,X> computed? Are angle brackets missed for l>m in line 178?

The concept of a nu-incoherent matrix appears in Theorems 4.2, 4.5 without earlier explanations. What are these matrices?

The words "coherence factor" appear for the first time in line 80 without defining this factor nu. Also, nu appears earlier in line 66, still without explanations. Is it the same "coherence" appearing much later in Definition 4.1?

The word "probability" appears for the first time in Theorem 4.2, later again in Theorem 4.5. How is the probability space defined to make the convergence with "high probability" meaningful?

In Theorem 4.3, can mu be large enough so that the resulting inequality becomes equality and hence no convergence is guaranteed? What are the values of mu in the experiments form section 5?

Lines 251-252 say that "an adaptation of the proof of [KV21] to the EDG setting is not possible due to non-orthogonality of the basis {w_alpha}_alpha of Definition 3.2". What specific step in the past proof requires this orthogonality?

Lines 857-858 say that "The Procrustes analysis uses similarity transformations like scaling, rotation and maps into a common reference frame [ADSVF23]". However the minimization after line 859 includes only translations and orthogonal maps. Was uniform scaling missed?  Why is the minimum achieved over infinitely many possible translations? Does this Procrustes distance satisfy the metric axioms and what is its computational complexity in the number of points?

The biggest concern is line 82 saying that "from which the geometry of the points P is trivially recovered". Main Algorithm 1 (MatrixIRLS) outputs a matrix X^(k) through some iterative optimization. Why is this final n x n matrix realizable by n points in a Euclidean space R^r? For instance, we cannot arbitrarily choose 6 pairwise distances between 4 points in the plane. Even if these distances are positive and satisfy triangle inequalities, they should also satisfy a complicated polynomial equation saying that the tetrahedron on these 4 points has volume 0. For n>4 points, there are many more such equations, so a random symmetric matrix of positive numbers is realized by points with probability 0.

**Limitations:**

This submission seems an incremental improvement of the past work [KV21] with little comparison between the two papers.

---

> ### Author Rebuttal · Authors · 2024-08-07
>
> Thank you for your thoughts about our submission. Please consider our responses regarding your  assessment:
>
> > (Meaningfulness of rank min. for Problem 1)
>
> Assuming the points matrix $P$ has rank $r$, it follows that the Gram matrix also has rank $r$. When $n \gg r$, this implies that the rank of the Gram matrix is small. This observation is the motivation behind minimizing the rank. As you noted, directly solving (1) is challenging.  Instead of solving (1) directly, we discuss convex approaches that use a convex surrogate for rank, which are more suitable for optimization. However, these polynomial-time methods are not scalable for large problems. Our proposed method avoids explicitly minimizing the rank and, under the assumptions of our theory, recovers the optimal solution of (1).
> Finally, we would like to mention that a number of prior works derive effective algorithms based on other relaxations of rank minimization.
>
> > "The rank minimization in (1) and algorithm MatrixILRS appeared under the same name but with slightly different notations in the earlier paper [KV21]."
>
> In our manuscript, we clearly mention that the proof strategy is inspired by the paper [KV21], in which a similar algorithm was proposed for low-rank matrix completion (LRMC). However, the considered EDG setup is more challenging due to the non-orthogonality of the sampling basis, which we overcome via Thm. 4.5, which is one reason why guarantees for any non-convex method have not been previously available in the literature.
>
>
> > (Motivation of rank-min for P1)
>
> Consider a quadrilateral with vertices located at $p_1=(-1,1)$, $p_2=(1,2)$, $p_3=(1,-2)$, and $p_4=(-1,-1)$. We know the lengths of the sides but not the length of the diagonals. Ignoring rank minimization, this problem can be formulated as an SDP feasibility problem using the relations $X_{i,i}+X_{j,j}-X_{i,j}=D_{i,j}$ where {(i,j)} are the indices corresponding to the available entries of the distance matrix. Solving this using CVX yields the rank $4$ matrix
> $$
> X=
> \begin{pmatrix}
>  1.431  &  0.431 &   -1.294 &   -0.569\\\\
>  0.431 &   4.431 &   -3.569 &   -1.294\\\\
>  -1.294 &  -3.569 &    4.431 &  0.431\\\\
>  -0.569 &  -1.294 &    0.431 &   1.431\\\\
> \end{pmatrix}
> $$
> This example illustrates that, when distances are missing, the feasibility problem can admit higher rank solutions i.e., it is "easier" to satisfy the distance constraints in higher dimensions. To find low-rank solutions, we can employ approaches that are based on factorizing the Gram matrix as $PP^T$. Doing so for this problem, with rank=2, yields
> $$
> \begin{pmatrix}
>  1 &  0 &   0 &  -1\\\\
>  0 &  4 &  -4 &   0\\\\
>  0 & -4 &   4 &   0\\\\
> -1 &  0 &   0 &   1
> \end{pmatrix}
> $$
> This not the true Gram matrix, but it satisfies the given distance constraints, indicating that no unique solution to P1 exists. However, if we assume the distance between \(p_1\) and \(p_3\) is known and restrict the rank to 2, then among various initializations, the one with the lowest objective function accurately identifies the underlying quadrilateral up to rigid motion and translation.
> > "In Def $3.2$..."
>
> Yes, $w_{\alpha}$ is a matrix of size $n\times n$. The measurement operator is based on the trace inner product
> $
> \langle w_{\alpha_l}, X \rangle =  \sum_{i,j}    w_{\alpha_l}(i,j) X(i,j)
> $.
> > (coherence $\nu$)
>
> We have introduced the notion of coherence in the Definition 4.1, the $\nu$-incoherent matrices in Theorem 4.2 and 4.5 refer to those matrices which have a coherence of $\nu$ (Definition 4.1). We will add an appropriate reference to earlier occurrences of this notion.
> > (Prob. space)
>
> We use the probability space with the discrete sample space of all subsets of cardinality m of the set if integers between $1$ and $n(n-1)/2$, equipped with the uniform distribution (sampling without replacement). The probability lower bound for the statement is $1-\frac{2}{n^{2}}$ (see proof of Thm. 4.3).
> >"In Theorem 4.3, can mu be large enough..."
>
> No, the initialization condition on the $k$-th iterate $X^{(k)}$ is formulated such that in the statement of Thm. 4.3, the inequality you are referring to will be strict (cf. the proof in Appendix D), given the specific value of $\mu$. For the Gaussian experiment for Figure 5, in the case of $r=5$ and $\rho=2$, for example, the resulting value will be $\mu \approx 2.56\cdot 10^{6}$. This leads to a relatively small theoretically certified "basin of quadratic convergence" if one wants to apply Thm 4.3. with the constants provided. However, we note that, first, Thm 4.3 is unique in the literature for this problem as it applies for the minimal sample complexity of $O(\nu n r log(n))$.
> >"Lines 251-252 say that "an adaptation ..."
>
> Consider the expansion of a low-rank matrix \(X\) in an orthogonal basis \(\{w_{\alpha}\}\):
> $$
> X = \sum_{\alpha} \langle X, w_{\alpha} \rangle w_{\alpha}.
> $$
> The standard low-rank matrix recovery analysis can be reformulated as the problem of finding a low-rank matrix given a few of the expansion coefficients $\{\langle X, w_{\alpha} \rangle\}$. In contrast, in the EDG setting, the natural basis considered is no longer orthogonal, making the above expansion invalid. The proof approach relies on a dual basis expansion, and the specific challenging steps involve showing bounds related to a modified sampling operator. These bounds depend on the linear-algebraic properties of the basis and the dual basis.
> > "The biggest concern is line 82 saying..."
>
> As the reviewer points out, not every random symmetric matrix corresponds to a set of points; positive definiteness is required to realize such points. Under the assumptions of our main theorem, the proposed algorithm converges to the true Gram matrix of rank
> $r$, which is both symmetric and positive definite. We then obtain the configuration of the points through an r-truncated eigendecomposition.
>
> We would appreciate if you consider raising your rating if we resolved your concerns. Thank you!

---

> > ### Comment · Reviewer_Jqjh · 2024-08-10
> >
> > Thank you for the detailed reply.
> >
> > >As the reviewer points out, not every random symmetric matrix corresponds to a set of points; positive definiteness is required to realize such points. Under the assumptions of our main theorem, the proposed algorithm converges to the true Gram matrix of rank $r$, which is both symmetric and positive definite. We then obtain the configuration of the points through an r-truncated eigendecomposition.
> >
> > Is it possible to specify the line numbers where the final matrix is proved to be symmetric and positive definite?

---

> > > ### Author Response · Authors · 2024-08-10
> > >
> > > Regarding symmetry, we want to note that all iterates $\mathbf{X}^{(k)}$ of Algorithm 1 are symmetric by construction: By (6), the solution of the equality-constrained weighted least squares problem (5) (with $W_{k-1}$ instead of $W_{k}$) can be written as
> > > $$
> > > \mathbf{X}^{(k)} = W_{k-1}^{-1}\mathcal{A}^*(z_{k-1}),
> > > $$
> > > where $z_{k-1}:= (\mathcal{A}W_{k-1}^{-1} \mathcal{A}^*)^{-1}(\mathbf{y})$ is the solution of the linear system $\mathcal{A}W_{k-1}^{-1} \mathcal{A}^* \mathbf{z} = \mathbf{y}$. From the definition of the measurement operator $\mathcal{A}$ (see line 177-178), it becomes clear that the adjoint operator $\mathcal{A}^*$ of $\mathcal{A}$ maps a vector $z \in \mathbb{R}^{m + n}$ to $\mathcal{A}^*(z) = \sum_{\alpha_{\ell} \in \Omega} z_\ell w_{\alpha_\ell} + \sum_{i=1}^n z_{m+i} w_{(i-m,i-m)}$, where the $w_{\alpha}$ are as in (4). Since all $w_{\alpha}$ are symmetric, it holds therefore that also $\mathcal{A}^*(z_{k-1})$ is symmetric as a sum of symmetric matrices. Based on the definition $W_{k-1}: = W_{\mathbf{X}^{(k-1)},\epsilon_{k-1}}$ of $W_{k-1}$ and Definition 3.1, it can be seen that $W_{k-1}^{-1}$ maps a matrix $\mathbf{Z}$ to $\mathbf{U}\Sigma_{\epsilon_{k-1}}\mathbf{U}^*\mathbf{Z}\mathbf{U}\Sigma_{\epsilon_{k-1}}\mathbf{U}^*$, where $\mathbf{U}$ is a left singular vector matrix of $\mathbf{X}^{({k-1})}$, and $\Sigma_{\epsilon_{k-1}}  = \operatorname{diag}(\max(\epsilon_{k-1},\sigma_i(\mathbf{X}^{(k-1)}))$, which is itself symmetric if $\mathbf{Z}$ is symmetric. Thus, each iterate $\mathbf{X}^{(k)}$ of Algorithm 1 is symmetric.
> > >
> > > Regarding positive semidefiniteness, it is true in general, iterates $\mathbf{X}^{(k)}$ can have negative eigenvalues. However, we prove in Theorem 4.3 that if Algorithm 1 is initialized close enough to a rank-$r$ ground truth matrix $\mathbf{X}^0$ and if the sample complexity satisfies the bound of line (229) under the mentioned random sampling model, with high probability, $\mathbf{X}^{(k)}$ converges to $\mathbf{X}^0$ (with quadratic rate). For the problems considered, the ground truth $\mathbf{X}^0 \in S_n$ is a realizable rank-$r$ matrix and thus p.s.d., so the iterates $\mathbf{X}^{(k)}$ converge to a psd matrix under these conditions. The proof of this theorem can be found in lines 786 to 809 (but uses also other intermediate lemmas, cf. the supplementary material).

---

> > > > ### Comment · Reviewer_Jqjh · 2024-08-12
> > > >
> > > > Thank you for your clarifications.
> > > > To summarize: symmetry holds, but positive-definiteness of the result is true with high probability only if there exists a ground truth matrix which is reconstructed, but there is no proof that the optimization result is indeed positive-definite in all reconstruction scenarios.
> > > > This coincides with my initial understanding.

---

> > > > > ### Author Response · Authors · 2024-08-13
> > > > >
> > > > > > To summarize: symmetry holds, but positive-definiteness of the result is true with high probability only if there exists a ground truth matrix which is reconstructed, but there is no proof that the optimization result is indeed positive-definite in all reconstruction scenarios.
> > > > >
> > > > > That is correct. As previously mentioned, in such cases, one can obtain a realizable point configuration by applying a truncated eigendecomposition. This point configuration will not satisfy the distance constraints exactly anymore, but approximately. Comparing such an outcome with the results of reference algorithms such as ALM, RieEDG and ScaledSGD, we would like to point out that the iterates of these algorithms do not satisfy the distance constraints in the first place, unless their objectives converge to 0, which they do not for many challenging problem instances. On the other hand, the satisfaction of the distance constraints by all iterates is built into MatrixIRLS. Thus, applying a truncated eigendecomposition to the outcome of MatrixIRLS is not less justified than using any of the other algorithms.

---

> > > > > > ### Author Response · Authors · 2024-08-13
> > > > > >
> > > > > > If you feel that we have addressed your questions and concerns adequately, we would appreciate if you can consider updating your score. We warmly welcome any further questions that you may have. Thank you!

---

### Official Review · Reviewer_wbRb · 2024-07-13

**Soundness:** 3
**Presentation:** 3
**Contribution:** 3
**Rating:** 6
**Confidence:** 3

**Summary:**

This paper tackles the problem of reconstructing geometric configurations from minimal Euclidean distance samples. The authors propose an innovative algorithm based on iteratively reweighted least squares (IRLS) within a non-convex optimization framework. They provide a local convergence guarantee for the method, ensuring accurate geometry recovery from a minimal number of random distance samples. Additionally, the paper establishes a restricted isometry property (RIP) relevant to other non-convex approaches. Numerical experiments demonstrate the algorithm’s efficiency and robustness compared to state-of-the-art methods.

**Strengths:**

1. The paper introduces a novel application of IRLS for non-convex optimization in Euclidean Distance Geometry (EDG) problems. Establishing the local convergence guarantee and RIP is a significant theoretical contribution.

2. The paper is well-researched, with rigorous theoretical analysis and proofs supporting the proposed method’s convergence and efficiency. The empirical validation through both synthetic and real-world datasets adds robustness to the claims.

3. The paper is well-structured and clearly written, making complex mathematical concepts accessible. The detailed algorithmic steps enhance understanding.

4. The proposed method addresses a fundamental problem in various applications, such as molecular conformation and sensor network localization. The ability to achieve accurate reconstruction with fewer samples has significant implications for data efficiency in these fields.

**Weaknesses:**

1. The paper compares the proposed method with a few state-of-the-art algorithms. Including more diverse comparisons, particularly with newer methods covered in the existing literature, would provide a more comprehensive evaluation.

2. The experiments could be enhanced by adding more parameter analysis or robustness analysis. For example, testing the performance under different parameters (n, r, k) on the ill-conditioned data will further validate the effectiveness of the proposed method.

3. The paper provides a local convergence guarantee, but a more detailed discussion on the global convergence properties of the algorithm would be beneficial.

**Questions:**

1.How sensitive is the proposed algorithm to the choice of parameters, such as rank estimate in the IRLS framework? Providing guidelines or heuristics for parameter selection would be helpful.

2.In the additional experiments presented in Appendix E.1, why does the MatrixIRLS method exhibit a large standard deviation when rho is in the range of 2.5 to 3.5? This suggests that the method is not very stable or robust across all settings. Considering the success probability, ALM performs better than MatrixIRLS. Could you explain the potential reasons for this discrepancy?

3.The time complexity of the MatrixIRLS algorithm appears to be missing a component in Step 7, which involves singular value decomposition with a complexity of O(n^3). Given this, what is the scalability of the MatrixIRLS method, such as the maximum problem size that can be solved within one hour?

4.Will the results of the MatrixIRLS method converge to the same solution as MDS when there is no incomplete information?

*Small suggestions:*

1.The linewidth and point size is too large in Figs. 2, 6(b), and 7(b).

2.No captions in Figs. 6 and 7.

3.Mixed use of fig. and Figure.

4.Some typos:

Line 52: should satisfy also satsify

Line 813: figure Figure

**Limitations:**

It would be beneficial to include the scalability limitations, such as the maximum problem size that can be solved.

---

> ### Author Rebuttal · Authors · 2024-08-07
>
> We appreciate your constructive and very detailed feedback to our submission. A point-by-point response and clarification regarding your comments follows below.
>
> **Weaknesses**
>
> > 1. The paper compares the proposed method with a few state-of-the-art algorithms. Including more diverse comparisons, particularly with newer methods covered in the existing literature, would provide a more comprehensive evaluation.
>
> We tried to be as comprehensive as reasonably possible in our experimental evaluation, including relevant methods ScaledSGD [ZCZ22] and RieEDG [SLCT23], the latter of which was very recently presented at a NeurIPS 2023 workshop. Another Riemannian method, which comes with an interesting theoretical analysis, was publicly available only in March 2024 [LS24]. Unfortunately, the authors of [LS24] do not provide a publicly available implementation, which made it not possible for us to reimplement their method and include fair algorithmic comparisons at short notice. For this reason, we contend that our evaluations are fair and comprehensive. We are open to suggestions for further methods to compare with.
>
> > 2. The experiments could be enhanced by adding more parameter analysis or robustness analysis. For example, testing the performance under different parameters (n, r, k) on the ill-conditioned data will further validate the effectiveness of the proposed method.
>
> We conducted additional experiments exploring the empirical performance of the considered algorithms in the case of ill-conditioned ground truth Gram matrices with $\kappa = 10^5$, considering now also $n=500$ and different choices of $r \in \{2,3,4,5\}$ as in the setting of Figure 1 of the submitted manuscript (Gaussian ground truth) and different oversampling factors $\rho$. The results can be found in Figure 1 of the attached PDF. It can be observed that RieEDG and ScaledSGD struggle considerably in this setting. On the other hand, the significant advantage of MatrixIRLS compared to ALM (reconstruction already with $\rho=1.5$ vs. with $\rho=2$ for MatrixIRLS vs. ALM) remains consistent for these different parameter choices. We also refer to our reply to your Question 1 for another robustness experiment.
>
> > 3. The paper provides a local convergence guarantee, but a more detailed discussion on the global convergence properties of the algorithm would be beneficial.
>
> We refer to our reply to a similar question of Reviewer fWZw below.
>
> **Questions**
>
> > 1. How sensitive is the proposed algorithm to the choice of parameters, such as rank estimate in the IRLS framework? Guidelines or heuristics [...] for parameter selection?
>
> Thank you for suggesting this insightful experiment. In order to compare the robustness with respect to the choice of the rank estimate (called $\tilde{r}$ for MatrixIRLS) for each of the four considered algorithms, we ran experiments over $8$ independent instances where the underlying rank estimate provided in the algorithm ($\tilde{r}$) is different from the actual rank ($r$) of the problem. For $n = 500$ and $r=5$ in a setup as in Figure 1 of the submission, we report in Figure 2 of the attached PDF that the median recovery error of MatrixIRLS only slightly deteriorates for $\tilde{r} \in \{6,7, 10\}$, and has a slight robustness advantage compared to ALM, whereas RieEDG performs significantly worse. None of the algorithms succeeds if $\tilde{r}< r$ is chosen.
>
> In many geometry reconstruction problems, however, there is a natural rank estimate choice based on the dimension of the geometry (e.g., $r=3$ for 3D problems), which makes $\tilde{r}$ easy to choose. If the dimensionality of the points to be embedded is not known, our experience suggests that when in doubt, a slightly higher $\tilde{r}$ than the true dimension (up to factor $2$ overestimation) will still lead to good results.
>
> > 2. In [...] Appendix E.1, why does the MatrixIRLS method exhibit a large standard deviation when rho is in the range of 2.5 to 3.5? [...] Considering the success probability, ALM performs better than MatrixIRLS. Could you explain the potential reasons for this discrepancy?
>
> The large standard deviation of MatrixIRLS in that experiment for $\rho \in [2.5,3]$ is due to the phenomenon that due to the randomness in the sampling pattern, MatrixIRLS sometimes does not converge to a meaningful ground truth, whereas it converges to very high precision in other cases. ALM exhibits a similar phenomenon, albeit already for smaller $\rho$ and which less pronounced standard deviation due to lower precision solutions. We do not have a final explanation of why ALM performs slightly better for smaller $\rho$ in this setting. One possible reason could be the large coherence $\nu$ of the ground truth in this real data setting considered, in which case Theorem 4.3 would be compatible with the need of more samples for successful reconstruction.
>
> > 3. Time complexity of the MatrixIRLS  $O(n^3)$ due to SVD? Maximal problem size to be solved via MatrixIRLS within an hour?
>
> Thank you for pointing out the omission of the computational discussion of Step 7. We would like to stress that it is _not_ true that we need full SVDs for the weight operator update step, but only the $r_k$ singular triplets. Based on the update rule for the smoothing $\epsilon$, it holds that $r_k \geq \tilde{r}$, but in many cases, empirically, $r_k$ coincides with $\tilde{r}$ throughout the algorithm. For this reason, we used a fast partial SVD method based on a randomized block Krylov iteration whose runtime scale as $O(C r_k log(n))$ [Musco & Musco 2015], where $C$ is the cost of a matrix-vector multiplication involving the matrix to be decomposed. Since we can use a "tangent space + sparse" representation of each iterate $X^{(k)}$ to which we apply this, which comes with a matrix-vector multiplication cost of $C= O(r_k n+ r_k^2 + m r_k)$, the cost of Step 7 is essentially linear in $n$. We refer to our global response for the maximal problem size question.

---

> > ### Comment · Reviewer_wbRb · 2024-08-14
> >
> > Thank you for your detailed clarification. As my concerns have been addressed, I will maintain my positive score. By the way, I would like to note that the time complexity you mentioned is not linear in $n$, but rather $O(n \log n)$.

---

### Official Review · Reviewer_rCFC · 2024-07-13

**Soundness:** 4
**Presentation:** 4
**Contribution:** 3
**Rating:** 7
**Confidence:** 2

**Summary:**

The paper examines the problem of recovering point locations based on incomplete pairwise Euclidean distance information. The problem is cast as a low-rank matrix recovery problem, and an algorithm based on iteratively-reweighted least squares is proposed. The authors derive a lower bound for the minimum number of required samples for reconstruction, and show that their algorithm has quadratic convergence. Experimental results show that their method has comparable results to other state-of-the-art methods for most data, but is also uniquely robust to ill-conditioned cases where other methods fail to converge.

**Strengths:**

* Provides the first theoretical convergence guarantee at the optimal amount of samples for the studied pairwise distance problem under non-convex optimization.
* Performance comparable to state of the art methods are demonstrated on
* The method converges quickly and effectively even for ill-conditioned matrices, where previous state of the art methods fail.
* Detailed theoretical derivations and proofs.
* An complete anonymized code repository is provided for reproducibility.

**Weaknesses:**

* Example data is somewhat small-scale and synthetic, but appears sufficient for comparison with other recovery methods.
* While this is not a weakness of the paper, my own theoretical background is insufficient to verify some of the proofs, such as the construction of the dual basis followed by the establishment of the restricted isometry property condition.

**Questions:**

* While number of steps/time to convergence is shown for ill-conditioned matrix experiments, they are not shown for real data experiments. How much faster does the proposed MatrixIRLS derivative converge compared to ALM? Are they comparable?
* Are there applicable downstream machine learning/deep learning applications that require this kind of point recovery, where improved results could be further demonstrated?

**Limitations:**

The authors address the limitations of their work in detail in the paper, and there are no societal impacts to be addressed.

---

> ### Author Rebuttal · Authors · 2024-08-07
>
> We appreciate your constructive reviews our submission. We address your questions below.
>
> > While number of steps/time to convergence is shown for ill-conditioned matrix experiments, they are not shown for real data experiments. How much faster does the proposed MatrixIRLS derivative converge compared to ALM? Are they comparable?
>
> We did not include runtime comparisons for this molecular confirmation problem in the submission as the respective protein geometries can be executed in an offline manner. However, we agree that despite of this, it is important that the algorithmic runtime does not become prohibitive for relevant problem instances. We have provided in Table 1 in the attached PDF the reported reconstruction times for each of the four algorithms applied to the 1BPM Protein data of Subsection 5.3 (with $n=3672$ datapoints) in a low-data regime with oversampling factor of $\rho=3$.
>
> It can be seen that MatrixIRLS is with $7.08$ minutes around $3$ times faster than ALM, which needed $23.4$ minutes until convergence. While these times certainly depend on the precise choice of stopping criteria for each algorithm (we used the ones indicated in Appendix E.2), this shows that the proposed method is competitive in terms of clock time (we also note that we obtain a considerably higher accuracy solution in this time than ALM).
> We will include such a comparison in the final version of the paper.
>
> > Are there applicable downstream machine learning/deep learning applications that require this kind of point recovery, where improved results could be further demonstrated?
>
> In the introduction of the manuscript, we have outlined a possible pathway for including Euclidean geometry reconstruction algorithms such the method presented into a protein structure prediction pipeline as used by AlphaFold 1-3. In particular, it could be possible to obtain a subset of high-confidence distance predictions by, e.g., a Transformer architecture, which could be combined with a MatrixIRLS-type geometry reconstruction to improve overall performance. Testing this was beyond the scope of this work due to the complexity of the overall pipeline.
> Furthermore, related reconstruction problems can be used to obtain faster word embeddings in natural language processing, see for example the paper ["Distance geometry for word representations and applications" by Khalife et al., Journal of Computational Mathematics and Data Science 6 (2023)]. Improvements could be demonstrated if favorable tradeoffs between computational complexity and downstream accuracy of MatrixIRLS-based tokenizers could be observed compared to non-distance geometry based tokenizers.

---

> ### Comment · Reviewer_rCFC · 2024-08-13
> **Thanks**
>
> I thank the authors for their work on the rebuttal. Based on their answers to my review, which are further positive points for the paper, as well as reading and considering the other reviews and their rebuttals, I see no reason to change my positive rating of the paper.

---

### Author Rebuttal · Authors · 2024-08-07

We appreciate the reviewers' constructive and very detailed feedback to our submission. A point-by-point response and clarification regarding their comments have been addressed separately to each reviewer, however, we highlight some key points below, many of which are supported by additional experiments whose results can be seen in the attached PDF.

**Runtime on Real Data and Scalability**

Certainly, it is important that the algorithmic runtime does not become prohibitive for relevant problem instances. We have provided in Table 1 in the attached PDF the reported reconstruction times for each of the four algorithms applied to the 1BPM Protein data of Subsection 5.3 (with $n=3672$ datapoints) in a low-data regime with oversampling factor of $\rho=3$.

It can be seen that MatrixIRLS is with $7.08$ minutes around $3$ times faster than ALM, which needed $23.4$ minutes until convergence. While these times certainly depend on the precise choice of stopping criteria for each algorithm (we used the ones indicated in Appendix E.2), this shows that the proposed method is competitive in terms of clock time.
In order to understand the scalability of the proposed algorithm, we have provided the time of completion of MatrixIRLS in the table Table 2. The first two rows shows that if we fixed the oversampling factor($\rho$) as $3$, then we are able to recover the points in the magnitude of $10^4$, is just $13.7$ minutes. So to further test the scalability, we also looked at the time to completion when less number of samples are provided ($\rho = 2.5$). In that setup $n = 10000$ takes $57.5$ minutes to recover with high precision. We will include such a comparison in the final version of the paper.

**Robustness to Parameter Choices**

 In order to compare the robustness with respect to the choice of the rank estimate (called $\tilde{r}$ for MatrixIRLS) for each of the four considered algorithms, we ran experiments over $8$ independent instances where the underlying rank estimate provided in the algorithm ($\tilde{r}$) is different from the actual rank ($r$) of the problem. For $n = 500$ and $r=5$ in a setup as in Figure 1 of the submission, we report in Figure 2 of the attached PDF that the median recovery error of MatrixIRLS only slightly deteriorates for $\tilde{r} \in \{6,7, 10\}$, and has a slight robustness advantage compared to ALM, whereas RieEDG performs significantly worse. None of the algorithms succeeds if $\tilde{r}< r$ is chosen.

In many geometry reconstruction problems, however, there is a natural rank estimate choice based on the dimension of the geometry ($r=2$ for 2D problems, $r=3$ for 3D problems), which makes $\tilde{r}$ easy to choose. If the dimensionality of the points to be embedded is not known, our experience suggests that when in doubt, a slightly higher $\tilde{r}$ than the true dimension (up to factor $2$ overestimation) will still lead to good results.

We conducted additional experiments exploring the empirical performance of the considered algorithms in the case of ill-conditioned ground truth Gram matrices with $\kappa = 10^5$, considering now also $n=500$ and different choices of $r \in \{2,3,4,5\}$ as in the setting of Figure 1 of the submitted manuscript (Gaussian ground truth) and different oversampling factors $\rho$. The results can be found in Figure 1 of the attached PDF. It can be observed that RieEDG and ScaledSGD struggle considerably in this setting, necessitating more than $\rho= 3.5$ for reasonable reconstruction rates. On the other hand, the significant advantage of MatrixIRLS compared to ALM (reconstruction already with $\rho=1.5$ vs. with $\rho=2$ for MatrixIRLS vs. ALM) remains consistent for these different parameter choices.

**Challenges for Global Convergence**

A major challenge for the global convergence theory of the proposed method is that main theorem, Thm. 4.3, applies only if the method is initialized close enough to the ground truth. The theory of similar IRLS algorithms for other problems (see papers [DDFG10,MF10,KV21]) shares this limitation, and therefore is not unique to our method. The challenge for applying classical optimization theory tools to the proposed scheme is rooted in the non-convexity of the (smoothed) log-det objective as well as in the non-constant smoothing (see reply to Reviewer fWZw for more details, as well as the discussion about the choice of $\mu$ in reply to Reviewer Jqjh).

--------
Due to lack of space, we provide below also an answer to one of the questions of Reviewer Jqjh:
> "Lines 857-858 say that "The Procrustes analysis uses similarity transformations like scaling, rotation and maps into a common reference frame [ADSVF23]". However the minimization after line 859 includes only translations and orthogonal maps. Was uniform scaling missed? Why is the minimum achieved over infinitely many possible translations? Does this Procrustes distance satisfy the metric axioms and what is its computational complexity in the number of points?"

Thank you for pointing this out. Rigid motions and translation preserve the distance matrix of an underlying set of points. In contrast, scaling does not. Given that, we only consider the Procrustes analysis without uniform scaling to remain consistent with the problem formulation. If we differentiate the objective with respect to the translation and find the closed form of the optimal translation as a function of the rotation, it can be shown that this optimal objective, for any choice of rotation, is equivalent to an objective whereby we first center the points at the origin. Specifically, we can equivalently consider the objective
$$
\underset{Q}{\min} ||QX_c-Y_c||_F,
 $$
where $X_c$ and $Y_c$ are centered at the origin. In terms of computation, the Procrustes analysis involves the SVD of a matrix of size $r \times r$ that costs $O(r^3)$, and matrix multiplication that costs $O(nr^2)$. Hence, the complexity is linear in the number of points.

---

### Decision · Program_Chairs · 2024-09-25

**Decision:**

Accept (poster)

**Comment:**

The submission studies the recovery of a low-rank Euclidean distance matrix from a subset of its entries. This problem has applications in sensor positioning, protein folding, and related areas. The paper studies an iterative reweighed least squares algorithm, which solves a sequence of weighted least squares completion problems. It proves that this method converges locally, with a linear rate. In particular, if the observed distances are sampled uniformly at random, and the number of observations is at least n r log n times the coherence of the target matrix, then IRLS converges linearly within a certain schatten-infinity neighborhood of the truth. In experiments, the IRLS approach shows advantages in handling ill-conditioned data — a challenging case for nonconvex factorization-type methods.

Reviewers produced a generally positive evaluation of the paper: while there has been quite a bit of theoretical work on nonconvex low-rank recovery, including analyses of both factorization and iterative least squares methods, the paper has novelty within the literature on nonconvex *euclidean distance matrix* reconstruction. The paper is crisply written. The main limitation of the theory is that it does not comprise a complete recovery result, since convergence is local in nature. One device for achieving a recovery result is to couple the local analysis with spectral initialization; doing so would help to contextualize the convergence guarantee (i.e., is the radius large enough to ensure near optimal sample complexity for this hybrid method). Overall, the problem makes a solid contribution to the literature on nonconvex low-rank recovery.